# Contextual Online Pricing with (Biased) Offline Data

**Yixuan Zhang**
Department of Industrial & Systems Engineering
University of Wisconsin-Madison
yzhang2554@wisc.edu

**Ruihao Zhu**
SC Johnson College of Business
Cornell University
ruihao.zhu@cornell.edu

**Qiaomin Xie**
Department of Industrial & Systems Engineering
University of Wisconsin-Madison
qiaomin.xie@wisc.edu

## Abstract

We study contextual online pricing with biased offline data. For the scalar price elasticity case, we identify the instance-dependent quantity $\delta^2$ that measures how far the offline data lies from the (unknown) online optimum. We show that the time length $T$, bias bound $V$, size $N$ and dispersion $\lambda_{\min}(\hat{\Sigma})$ of the offline data, and $\delta^2$ jointly determine the statistical complexity. An Optimism-in-the-Face-of-Uncertainty (OFU) policy achieves a minimax-optimal, instance-dependent regret bound $\tilde{\mathcal{O}}\big(d\sqrt{T} \wedge (V^2T + \frac{dT}{\lambda_{\min}(\hat{\Sigma})+(N\wedge T)\delta^2})\big)$. For general price elasticity, we establish a worst-case, minimax-optimal rate $\tilde{\mathcal{O}}\big(d\sqrt{T} \wedge (V^2T + \frac{dT}{\lambda_{\min}(\hat{\Sigma})})\big)$ and provide a generalized OFU algorithm that attains it. When the bias bound $V$ is unknown, we design a robust variant that always guarantees sub-linear regret and strictly improves on purely online methods whenever the exact bias is small. These results deliver the first tight regret guarantees for contextual pricing in the presence of biased offline data. Our techniques also transfer verbatim to stochastic linear bandits with biased offline data, yielding analogous bounds.

## 1 Introduction

Contextual online pricing [11, 2] models the real-world task in which a firm, upon observing customer-specific features, sets a price, observes the resulting demand, and then adjusts future prices to maximize long-term revenue. A central challenge here is to continuously balance exploitation—using the current estimated optimal pricing strategy to maximize immediate revenue—with exploration—testing alternative prices to improve those estimates. Importantly, most firms already maintain extensive historical pricing logs—data that are *free* to use and impose no opportunity cost on current revenue. Leveraging these logs can shorten the costly exploration phase, reduce the risk of customer churn from sub-optimal prices, and provide valuable information on rare or infrequent contexts. Motivated by these observations, recent work [31] introduced the framework of *Contextual Online Pricing with Offline Data* (C-OPOD) and showed that if the offline data are *unbiased*—that is, drawn from the same distribution as the forthcoming online data—incorporating them enables an online policy to outperform purely online learning approaches.

In practice, distributional shifts are ubiquitous. For instance, historical iPhone pricing data often differ from current patterns because of competitor moves, product upgrades, and evolving economic conditions, making the no-shift assumption unrealistic. Recent research therefore starts to explore the use of *biased* offline data. In the degenerate $K$-armed bandit setting (with no context and finite

actions), recent work [10] showed that, given offline data and information of the bias, one can design an algorithm that outperforms the canonical online method and attains matching upper and lower regret bounds.

However, the method and results of [10] do not translate directly to contextual online pricing, where contextual information, a continuous action space, and pricing-specific structure must be accommodated; a naive extension incurs sub-optimal regret (see discussions under Theorem 1). To fill this gap, in this work, we formulate and study the *Contextual Online Pricing with Biased Offline Data* (CB-OPOD) problem.

## 1.1 Main contributions

**Impossibility Result.** We first demonstrate in Corollary 1 that, with access to an offline pricing dataset only, no policy can uniformly outperform the contextual online pricing algorithm in [2] without further information on the discrepancy between the offline and online data distributions.

**Algorithm design and analysis.** To sidestep the impossibility result, we start by assuming the firm knows a *bias bound* $V$ on the true distributional shift between the offline and online data distributions. We develop algorithms and regret guarantees for two scenarios: 1. the *scalar price-elasticity* case that the market-baseline feature is $d_1$-dimensional, while the price-elasticity feature is scalar ($d_2 = 1$), and 2. the *general CB-OPOD* setting that further permits $d_2 > 1$.

1. For the *scalar price elasticity* setting, we identify the first instance-dependent quantity $\delta^2$ that measures how far the offline data lies from the (unknown) optimal price strategy and governs the statistical complexity of CB-OPOD. We propose the *Contextual Online–Offline Pricing with Optimism* (CO3) algorithm with a novel three-ellipsoid constructed confidence set. Our algorithm achieves a minimax-optimal, instance-dependent regret bound $\tilde{\mathcal{O}}\Big(d_1\sqrt{T} \wedge (V^2 T + \frac{d_1 T}{\lambda_{\min}(\hat{\Sigma}) + (N \wedge T)\delta^2})\Big)$, where $\lambda_{\min}(\hat{\Sigma})$ measures the dispersion of the offline data and $N$ denotes its sample size. Under certain conditions, this can be tightened to $\mathcal{O}(\delta^2 T)$. Our results also recover the OPOD regret in [7] when $V = 0$ while improving the bounds and relaxing the offline data assumptions in [31]. We provide a summary in Table 1 and detailed descriptions in Section 3.

2. For the *general CB-OPOD* setting, we propose the *General Contextual Online-Offline Pricing with Optimism* (GCO3) algorithm and establish a minimax-optimal, worst-case regret bound $\tilde{\mathcal{O}}\Big((d_1 + d_2)\sqrt{T} \wedge (V^2 T + \frac{(d_1+d_2)T}{\lambda_{\min}(\hat{\Sigma})})\Big)$. This result provides the first guarantee for general CB-OPOD with either biased or unbiased offline data. In addition, our techniques apply directly to the stochastic linear bandit setting, thereby subsuming the result for the $K$-arm bandit setting [10]. These results are summarized in Table 1 and discussed in details in Sections 4.1.

**Robustness result.** If the bias bound $V$ is unknown ahead, we propose the *Robust Contextual Online-Offline Pricing with Optimism* (RCO3) algorithm for the general CB-OPOD setting. By choosing a parameter $\alpha \in (0, \frac{1}{2})$, RCO3 can achieve regret no larger than $\tilde{\mathcal{O}}(T^{1-\alpha})$ and also $\tilde{\mathcal{O}}(T^{\alpha})$ regret when $V_{\text{true}}$ is relatively small. To the best of our knowledge, this is the first robust algorithm for CB-OPOD. These findings are summarized in Table 1 and explained in details in Section 4.2.

## 1.2 Related work

In this section, we provide a brief overview of closely related work and defer a comprehensive discussion to Appendix B.

**Online learning with external (biased) information.** Online learning with offline information attracts growing attention. [24, 19, 7, 29, 31] show that one can achieve improved regret when utilizing unbiased offline data appropriately. [23, 32, 30, 10, 14] further consider the case where offline data can be biased. Meanwhile, Bayesian methods such as Thompson sampling (TS) can leverage biased offline data to construct the prior for online learning, but misspecified priors may lead to regret bounds worse than purely online approaches [20, 16, 26], e.g., an additional $\mathcal{O}(\epsilon T^2)$ when the prior is off by $\epsilon$, which exceeds the bound in Table 1.

| Setting | Offline data | Bias bound $V$ | Regret |
|---------|-------------|---------------|--------|
| **N** [7] | **I** | $V = 0$ | $\tilde{\mathcal{O}}\big(\sqrt{T} \wedge \frac{T}{\lambda_{\min}(\hat{\Sigma}) + (N \wedge T)\delta^2}\big)$ |
| **S** [31] | **I, F** | $V = 0$ | $\tilde{\mathcal{O}}\big(d_1\sqrt{T} \wedge \frac{d_1^2 T}{(N \wedge T)\delta^2}\big)$ |
| **S** (Theorem 1) | **I** | $V$ is known | $\tilde{\mathcal{O}}\big(d_1\sqrt{T} \wedge (V^2 T + \frac{d_1 T}{\lambda_{\min}(\hat{\Sigma}) + (N \wedge T)\delta^2}))$ |
| **G** (Theorem 3) | **I** | $V$ is known | $\tilde{\mathcal{O}}\big((d_1 + d_2)\sqrt{T} \wedge (V^2 T + \frac{(d_1 + d_2) T}{\lambda_{\min}(\hat{\Sigma})}))$ |
| **G** (Theorem 5) | **I** | $V$ is unknown | $\tilde{\mathcal{O}}\big(T^\alpha + V_{\text{true}}^2 T\big)$ if $V_{\text{true}}^2 \lesssim T^{-\alpha}$ |

Table 1: Summary of our results and the most related work on online pricing with offline data. Here, **N** denotes **N**on-contextual online pricing, **S** denotes **S**calar price elasticity, **G** denotes **G**eneral contextual online pricing, **I** denotes **I**.i.d. demand fluctuations and **F** denotes **F**ixed pricing policy.

**Online pricing.** Online pricing has also garnered significant interest. Under linear demand function, [15] and [2] establish $\tilde{\Theta}(\sqrt{T})$ minimax regret bound in the *non-contextual* and *contextual* settings, respectively. Built on these, [7] study the impact of unbiased offline data in the non-contextual setting and [31] extend this to the contextual setting with a fixed policy that collects offline data. It is worth noting that in [31], due to the *fixed policy* used to collect offline data, the minimax regret is worsened, and cannot recover the non-contextual result in [7]. This is because the fixed policy may limit the offline data dispersion and the regret incorporates the randomness in the offline data. In contrast, we make no such assumptions on the offline data, which allows a tighter minimax regret over all possible offline data, and it also recovers the non-contextual setting [7] when offline data is unbiased.

### 1.3 Notations

Throughout this paper, we use $\|\cdot\|$ to denote the Euclidean norm. We use $\mathcal{O}(\cdot)$, $\Theta(\cdot)$, and $\Omega(\cdot)$ to denote upper, tight, and lower bounds on growth rates, respectively; analogously, $\tilde{\mathcal{O}}(\cdot)$, $\tilde{\Theta}(\cdot)$, and $\tilde{\Omega}(\cdot)$ further hides the logarithmic factors. We also use $A \lesssim B$, $A \gtrsim B$ and $A \asymp B$ to indicate $A \in \mathcal{O}(B)$, $A \in \Omega(B)$ and $A \in \Theta(B)$, respectively. For any $a, b \in \mathbb{R}$, we denote $a \wedge b = \min\{a, b\}$. Given a matrix $M$, we let $\lambda_{\min}(M)$ and $\lambda_{\max}(M)$ represent the smallest and largest eigenvalues of $M$, respectively. We denote $\text{Proj}_{[a,b]}(c) := \arg\min_{x \in [a,b]} |x - c|$ for any $a, b, c \in \mathbb{R}$.

## 2 Problem setup and preliminaries

In this section, we introduce the model of the CB-OPOD problem. We also present an impossibility result that emphasizes the importance of information on the offline data.

**Online model.** Consider a firm that sells products over a horizon of $T$ periods. In each period $t = 1, 2, \ldots, T$, the firm observes contextual information (e.g., category, brand, origin, and other attributes) about the incoming product. We denote by $x_t \in \mathbb{R}^{d_1}$ the feature vector affecting baseline demand, and by $y_t \in \mathbb{R}^{d_2}$ the vector governing price elasticity. We assume that the online feature sequence $\{(x_1, y_1), (x_2, y_2), \ldots, (x_T, y_T)\}$ is independently and identically distributed (i.i.d.) with support in a set $\mathcal{X} \times \mathcal{Y} \subset \mathbb{R}^d$. In period $t$, the firm sets a price $p_t$ (potentially based on historical data), after which the random demand $D_t$ is observed. We adopt the following linear demand model [21, 2, 4]:

$$D_t = \alpha_*^\top x_t + \beta_*^\top y_t p_t + \epsilon_t, \quad \forall t \in [T], \tag{1}$$

where $\theta_* := (\alpha_*, \beta_*) \in \Theta^\dagger \subseteq \mathbb{R}^{d_1 + d_2}$ denotes the *unknown* demand parameter that lies in a set $\Theta^\dagger$, and $\{\epsilon_t\}_{t \geq 1}$ is an sequence of independent random demand fluctuations with zero mean and is $R$−subgaussian. We highlight that linear demand model is both classic [15, 5, 21] and actively studied [2, 4, 25, 18]. Moreover, [5] show that a semi-myopic policy based on a linear model converges to the optimal policy for an unknown (possibly nonlinear) true model under suitable regularity conditions and attains minimax-optimal regret. Therefore, in this work we focus on the linear demand model (1). Extending our results to general nonlinear models and analyzing model misspecification are left for future work.

For the demand (1), the first term $\alpha_*^\top x_t$ represents the baseline market size, and $\beta_*^\top y_t$ represents the price sensitivity. For a fixed parameter $\theta = (\alpha, \beta) \in \Theta^\dagger$ and context $(x, y) \in \mathcal{X} \times \mathcal{Y}$, the firm's expected revenue from charging price $p$ is given by $r_\theta(p, x, y) = p(\alpha^\top x + \beta^\top yp)$. Then, the firm's single-period optimal price and optimal expected revenue are defined as

$$p_\theta^*(x, y) = \arg\max_{p \geq 0} r_\theta(p, x, y) \quad \text{and} \quad r_\theta^*(x, y) = \max_{p \geq 0} r_\theta(p, x, y). \tag{2}$$

Next, we introduce the following assumption on the online model.

**Assumption 1.** *For online model, we assume*

1. $\Theta^\dagger$ *and* $\mathcal{X} \times \mathcal{Y}$ *are compact sets and there exist positive constants* $\alpha_{\max}, \beta_{\max}, x_{\max}$ *and* $y_{\max}$ *such that* $\|\alpha\| \leq \alpha_{\max}, \|\beta\| \leq \beta_{\max}, \|x\| \leq x_{\max}$ *and* $\|y\| \leq y_{\max}$ *for all* $(\alpha, \beta) \in \Theta^\dagger$ *and* $(x, y) \in \mathcal{X} \times \mathcal{Y}$.

2. $\mathbb{E}[x_1 x_1^\top]$ *and* $\mathbb{E}[y_1 y_1^\top]$ *are positive definite.*

3. *There exist positive constants* $l_\alpha, u_\alpha, l_\beta$ *and* $u_\beta$ *such that* $l_\alpha \leq \alpha^\top x \leq u_\alpha$ *and* $l_\beta \leq -\beta^\top y \leq u_\beta$ *for all* $(\alpha, \beta) \in \Theta^\dagger, (x, y) \in \mathcal{X} \times \mathcal{Y}$. *Consequently, the optimal price satisfies* $p_\theta^*(x, y) = -\frac{\alpha^\top x}{2\beta^\top y} \in [l, u]$ *for any* $(\alpha, \beta) \in \Theta^\dagger$, *where* $l = \frac{l_\alpha}{2u_\beta}$ *and* $u = \frac{u_\alpha}{2l_\beta}$.

Assumption 1 is a standard regularity condition in contextual-pricing studies [2, 4, 18]. In the case $d_2 = 1$, Assumption 1 further guarantees a constant $y_{\min} > 0$ such that $|y| \geq y_{\min}$ for every $y \in \mathcal{Y}$.

**Offline data model.** In practice, the firm does not know the exact values of $(\alpha_*, \beta_*)$, but has access to a pre-existing offline dataset prior to the online learning process. Suppose this dataset consists of $N$ samples $\{(\hat{x}_n, \hat{y}_n, \hat{p}_n, \hat{D}_n)\}_{n \in [N]}$, where $(\hat{x}_n, \hat{y}_n) \in \mathcal{X} \times \mathcal{Y}$ for all $n \in [N]$. For each $n \in [N]$, the demand realization $\hat{D}_n$ under the historical price $\hat{p}_n$ is generated according to the linear model:

$$\hat{D}_n = \alpha_*'^\top \hat{x}_n + \beta_*'^\top \hat{y}_n \hat{p}_n + \hat{\epsilon}_n,$$

where $\theta_*' := (\alpha_*', \beta_*') \in \Theta^\dagger$ are the *unknown* offline demand parameters. The fluctuations $\{\hat{\epsilon}_n\}_{n \in [N]}$, independent of features and prices $\{(\hat{x}_n, \hat{y}_n, \hat{p}_n)\}_{n \in [N]}$, form a sequence of independent, zero-mean $R$-subgaussian random variables, and are the only source of randomness in the offline dataset. We use $\hat{\Sigma} = \begin{bmatrix} \hat{\Sigma}_{x,x} & \hat{\Sigma}_{x,y} \\ \hat{\Sigma}_{y,x} & \hat{\Sigma}_{y,y} \end{bmatrix} = \sum_{n=1}^N \begin{bmatrix} \hat{x}_n \hat{x}_n^\top & \hat{x}_n \hat{p}_n \hat{y}_n^\top \\ \hat{y}_n \hat{p}_n \hat{x}_n^\top & \hat{y}_n \hat{p}_n^2 \hat{y}_n^\top \end{bmatrix}$ to denote the offline Gram matrix.

**Pricing policies and performance metrics.** We consider the design and analysis of pricing policies for a firm that does not know the true $\theta_*$ nor the distribution of the i.i.d. online feature $\{(x_t, y_t)\}_{t \in [T]}$. At the time $t$, the firm proposes the price $p_t$ as an output of a policy function $\pi_t$ that takes all the historical information by time $t - 1$ and the current feature $(x_t, y_t)$ as input arguments. That is,

$$p_t = \pi_t(\{(\hat{x}_n, \hat{y}_n, \hat{p}_n, \hat{D}_n)\}_{n \in [N]}, \{(x_s, y_s, p_s, D_s)\}_{s \in [t-1]}, x_t, y_t).$$

We denote $\Pi$ as the set of all such policies $\pi = (\pi_1, \pi_2, \dots)$. The set $\Pi$ includes all policies that are feasible for the firm to execute. For any policy $\pi \in \Pi$, the regret of $\pi$, denoted by $R_{\theta_*', \theta_*}^\pi(T)$, is defined as the difference between the optimal expected revenue generated by the clairvoyant policy that knows the exact value of $\theta_*$ and the expected revenue generated by pricing policy $\pi$, i.e.,

$$R_{\theta_*', \theta_*}^\pi(T) = \mathbb{E}\Big[\sum_{t=1}^T r_{\theta_*}^*(x_t, y_t) - r_{\theta_*}(p_t, x_t, y_t)\Big].$$

The expectation is taken with respect to two sources of randomness: 1) the randomness in online features $\{(x_t, y_t)\}_{t \in [T]}$ and 2) the randomness from both offline and online fluctuations $\{\hat{\epsilon}_n\}_{n \in [N]}$ and $\{\epsilon_t\}_{t \in [T]}$. We treat the offline feature–price tuples $\{(\hat{x}_n, \hat{y}_n, \hat{p}_n)\}_{n \in [N]}$ as *deterministic*, imposing no distributional assumptions. Consequently, our regret is defined *conditional* on the realized offline data. An unconditional bound can be easily obtained by taking an additional expectation over the offline feature–price tuples and applying standard concentration inequalities to the data-dependent terms in our regret bounds.

**An impossibility result.** We first present an impossibility result on the CB-OPOD problem without any information on the bias of the offline data.

**Corollary 1** (Impossibility Result). *Under Assumption 1, for any policy $\pi \in \Pi$ without the prior knowledge on the exact bias $V_{\text{true}} = \|\theta'_* - \theta_*\|$, we have $\sup_{(\theta'_*, \theta_*) \in \Theta^\dagger \times \Theta^\dagger} R^\pi_{\theta'_*, \theta_*}(T) \in \Omega(\sqrt{T})$.*

Corollary 1 states that, even with access to an offline dataset, any algorithm will face a worst-case scenario where it cannot outperform the purely online algorithm [2] without additional information or constraints on the discrepancy between the offline and online models. In practice, a bias bound $V \geq \|\theta'_* - \theta_*\|$ can be estimated with robust ML techniques [6] or via cross-validation [9]. Hence the above lower bound is conservative, motivating the study of more practically relevant settings that admit tighter regret guarantees. Corollary 1 follows directly from Theorem 4, so we omit the proof.

# 3 Scalar price elasticity

In this section, we assume that the firm has access to a bias bound $V$ prior to the online phase. We focus on the setting $d_2 = 1$, where the price elasticity is a scalar. For this setting, we make the following assumption on the offline data.

**Assumption 2.** *There exists a positive constant $c > 0$ such that $\lambda_{\min}(\hat{\Sigma}_{x,x}) \geq cN$.*

Assumption 2 implies that the offline market base-demand features are sufficiently well-covered. We remark that Assumption 2 is directly satisfied by choosing $c = 1$ for the OPOD problem [7]. Furthermore, [31] assumes that $\{\hat{x}_n\}_{n \in [N]}$ and $\{x_t\}_{t \in [T]}$ are i.i.d., which implies that Assumption 2 holds with high probability. Hence, our Assumption 2 is no stronger than those in [7, 31].

Importantly, we first define the offline empirical price strategy $\hat{p}(x, y)$ and introduce the generalized distance $\delta^2$ between $\hat{p}(x, y)$ and the true optimal price strategy $p^*_{\theta_*}(x, y)$ as follows:

$$\hat{p}(x, y) := \hat{A}^\top x / y, \quad \forall (x, y) \in \mathcal{X} \times \mathcal{Y} \quad \text{and} \quad \delta^2 := \mathbb{E}_{x,y}[(\hat{p}(x, y) - p^*_{\theta_*}(x, y))^2], \qquad (3)$$

where $\hat{A} := \hat{\Sigma}_{x,x}^{-1} \hat{\Sigma}_{x,y} = \arg\min_{\alpha \in \mathbb{R}^d} \sum_{n=1}^N (\alpha^\top \hat{x}_n - \hat{y}_n \hat{p}_n)^2$ represents the ordinary least–squares estimator that best fits the linear relation $\hat{A}^\top \hat{x}_n \approx \hat{y}_n \hat{p}_n$ in the $L_2$ sense. To the best of our knowledge, our definitions of $\hat{p}(x, y)$ and $\delta^2$ are the first to expose the intrinsic connection between contextual offline data and online pricing. In the special case where the problem reduces to OPOD (i.e., no context and all features equal to 1), $\hat{p}(x, y)$ simplifies to the classical average price $\hat{p} = N^{-1} \sum_{n=1}^N \hat{p}_n$ [7]. Choosing appropriate forms for $\hat{p}$ and $\delta^2$ is critical; Appendix C.1 provides a detailed discussion.

The quantity $\delta^2$ measures the deviation of the offline data from the (unknown) optimal pricing strategy, and thus plays a crucial role in guiding the algorithm's behavior: a small $\delta^2$ suggests the algorithm can primarily rely on the offline data for exploitation, whereas a large $\delta^2$ indicates the need for additional online exploration. Since $\delta^2$ is unknown, adapting to it introduces challenges in both the algorithm design and the regret analysis. We elaborate on these issues in the next subsection.

## 3.1 The CO3 algorithm: upper and lower regret bounds

Building on the above definitions, we propose the *Contextual Online–Offline Pricing with Optimism* (CO3) algorithm, which follows the celebrated Optimism in the Face of Uncertainty (OFU) principle [1]. In contrast to traditional OFU methods and online pricing approaches with *unbiased* offline data [7, 31], which typically rely on a single confidence ellipsoid, our CO3 algorithm incorporates the *biased* offline dataset by constructing a confidence set as the intersection of three ellipsoids at time $t$:

$$\mathcal{C}_t = \left\{ \theta \in \mathbb{R}^{d_1+d_2} : \|\theta - \hat{\theta}_{t,N}\|_{\Sigma_{t,N}} \leq w_{t,N}, \|\theta - \hat{\theta}_{t,N}\| \leq \hat{w}_{t,N}, \|\theta - \hat{\theta}_t\|_{\Sigma_t} \leq w_t \right\}, \qquad (4)$$

where $\Sigma_t$ is the *online* Gram matrix, $\Sigma_{t,N}$ the *combined* online–offline Gram matrix, and $\hat{\theta}_t$ and $\hat{\theta}_{t,N}$ are the corresponding least-squares estimators (see Appendix A). We choose the constants $(w_{t,N}, \hat{w}_{t,N}, w_t)$ properly to ensure that that $\theta_* \in \mathcal{C}_t$ with high probability (cf. Lemma 5). Then, the CO3 algorithm proceeds in two stages: (i) an offline test to decide whether to rely on the offline empirical pricing strategy $\hat{p}$ during the online phase; and (ii) otherwise, pricing optimistically with respect to $\mathcal{C}_t$.

---

**Algorithm 1** CO3 Algorithm

---

**Input:** Offline data $\{(\hat{x}_n, \hat{y}_n, \hat{p}_n, \hat{D}_n)\}_{n \in [N]}$, regularization parameter $\lambda$, $\{(w_{t,N}, \hat{w}_{t,N}, w_t)\}_{t \geq 0}$ defined in Appendix A with $\epsilon = 1/T^2$ and bias bound $V$.

1: **if** $\min_{\theta \in \mathcal{C}_0 \cap \Theta^\dagger} \sum_{n=1}^N (\hat{p}(\hat{x}_n, \hat{y}_n) - p_\theta^*(\hat{x}_n, \hat{y}_n))^2 \leq \frac{N x_{\max}^2 y_{\max}^2}{y_{\min}^2 \lambda_{\min}(\mathbb{E}[xx^\top])} \max\{V^2, \frac{1}{\lambda_{\min}(\hat{\Sigma})}\}$ and
    $\max\{V^2, \frac{1}{\lambda_{\min}(\hat{\Sigma})}\} \leq T^{-1/2}$ **then**               ▷ evaluating if $\hat{p}$ by itself is enough
2:    Charge $p_t = \mathrm{Proj}_{[l,u]}(\hat{p}(x_t, y_t))$ for $t \in [T]$.
3: **else**
4:    **for** $t = 1, 2, \ldots, T$ **do**
5:        Observe context vector $(x_t, y_t)$;
6:        **if** $\mathcal{C}_{t-1} \cap \Theta^\dagger \neq \emptyset$ **then**
7:            Compute $(p_t, \tilde{\theta}_t) = \arg\max_{p \in [l,u], \theta \in \mathcal{C}_{t-1} \cap \Theta^\dagger} p \cdot (\alpha^\top x_t + \beta y_t p)$;
8:            Charge price $p_t$;                    ▷ optimism with respect to $\mathcal{C}_t$
9:        **else**  Charge $p_t = l$;
10:       **end if**  Observe $D_t$.
11:   **end for**
12: **end if**

---

We highlight that the three-ellipsoid confidence set in Algorithm 1 is designed to capture the best of three worlds. (1) *Online safety.* The constraint $\|\theta - \hat{\theta}_t\|_{\Sigma_t} \leq w_t$ follows as the classical purely online algorithm, thus guaranteeing regret no larger than $\tilde{\mathcal{O}}(\sqrt{T})$. (2) *Offline-boosted estimation.* The Euclidean condition $\|\theta - \hat{\theta}_{t,N}\| \leq \hat{w}_{t,N}$ leverages the offline data to sharpen the *estimate* of $\theta_*$. (3) *Aggressive exploitation.* Intuitively, a large $\delta^2$ implies that the offline data lie far from the true optimum, so pricing decisions close to $p^*$ simultaneously promote exploration and exploitation. The ellipsoid $\|\theta - \hat{\theta}_{t,N}\|_{\Sigma_{t,N}} \leq w_{t,N}$, paired with the UCB pricing rule (Line 7 of Algorithm 1), allows the algorithm to set prices well away from the offline estimate $\hat{p}$ (see Lemma 9), thereby exploiting the market more aggressively when $\delta^2$ is large. Earlier pricing work with *unbiased* offline data [7, 31] relies on a *single* ellipsoid $\|\theta - \hat{\theta}_{t,N}\|_{\Sigma_{t,N}} \leq w_{t,N}$, which can perform *worse* than purely online algorithm when the offline data is biased. The work on $K$-armed bandit with biased offline data [10] employs two confidence intervals; however, their technique does not extend to contextual pricing with infinitely many actions and fails to capture the dependence on the instance-dependent quantity $\delta^2$.

With *unbiased* offline data, [7] showed a sharp phase transition governed by $\delta^2$: (1) when the offline data is highly informative— specifically, $\delta^2 \lesssim 1/\lambda_{\min}(\hat{\Sigma}) \lesssim T^{-1/2}$— offline data alone suffice, so online exploration is unnecessary; (2) otherwise, a larger $\delta^2$ boosts both exploration and exploitation. We extend this principle to the *biased* setting: (i) if the offline data remain informative despite the shift, i.e. $\delta^2 \lesssim \max\{V^2, 1/\lambda_{\min}(\hat{\Sigma})\} \lesssim T^{-1/2}$, then simply deploying the empirical policy $\hat{p}$ achieves regret $\mathcal{O}(\delta^2 T)$; (ii) otherwise, a larger $\delta^2$ accelerates both exploitation and exploration, leading to a lower overall regret. Since $\delta^2$ is *unknown*, in order to adapt to the two regimes, we introduce an *offline testing phase* (Line 1 of Algorithm 1)—new for contextual pricing with biased offline data—to determine whether employing $\hat{p}$ is sufficient. The following theorem provides an upper bound on the regret of Algorithm 1.

**Theorem 1.** *Let $\pi$ be Algorithm 1. Under Assumptions 1 and 2, for any possible $(\theta'_*, \theta_*) \in \Theta^\dagger \times \Theta^\dagger$ such that $\|\theta'_* - \theta_*\| \leq V$ and for any $T \geq 1$,*

$$R_{\theta'_*, \theta_*}^\pi(T) \in \begin{cases} \mathcal{O}\left(\delta^2 T\right), \text{ if } \delta^2 \lesssim \max\{V^2, \frac{1}{\lambda_{\min}(\hat{\Sigma})}\} \lesssim T^{-1/2}; \\ \mathcal{O}\left(d_1 \sqrt{T} \log T \wedge (V^2 T + \frac{d_1 T \log T}{\lambda_{\min}(\hat{\Sigma})}) \wedge \frac{\lambda_{\max}(\hat{\Sigma}) V^2 T \log T + d_1 T \log^2 T}{\lambda_{\min}(\hat{\Sigma}) + (N \wedge T) \delta^2}\right), \text{otherwise.} \end{cases}$$

We first remark that the regret of Algorithm 1 always satisfies $R_{\theta'_*, \theta_*}^\pi(T) \in \tilde{\mathcal{O}}(\sqrt{T})$, ensuring that the algorithm is *never worse* than a purely online strategy. Theorem 1 refines the regret guarantee in line with above insights (i)–(ii). If the offline data is further well conditioned—specifically when $\lambda_{\min}(\hat{\Sigma}) \asymp \lambda_{\max}(\hat{\Sigma})$, a condition commonly satisfied in price experiments [2, Lemma 1]—the regret scales as $\tilde{\mathcal{O}}\left(d_1 \sqrt{T} \wedge (V^2 T + \frac{d_1 T}{\lambda_{\min}(\hat{\Sigma}) + (N \wedge T) \delta^2})\right)$ and improves further to $\mathcal{O}(\delta^2 T)$ in a special corner regime. Crucially, *smaller* bias bounds $V$ and *larger* dispersion $\lambda_{\min}(\hat{\Sigma})$ of offline data yield lower

regret. In particular, if the bias bound is small with $V^2 \in \mathcal{O}(T^{-1/2})$, the regret becomes strictly smaller than $\tilde{\mathcal{O}}(\sqrt{T})$ whenever either of the following holds: $\lambda_{\min}(\hat{\Sigma}) \in \Omega(\sqrt{T})$, indicating strong dispersion that sharpens the estimate of $\theta_*$; or $(N \wedge T)\delta^2 \in \Omega(\sqrt{T})$, meaning the offline data is sufficient and far from the optimum with a large distance, thus accelerating both exploration and exploitation. Conversely, when the bias bound is large, i.e., $V^2 \in \Omega(T^{-1/2})$, Algorithm 1 cannot beat the baseline rate $\tilde{\mathcal{O}}(\sqrt{T})$.

When $V = 0$, Theorem 1 reproduces the OPOD bound of [7] and improves upon the C-OPOD bound of [31]: if $N, \lambda_{\min}(\hat{\Sigma}) \to \infty$, our theorem 1 implies *zero* regret, whereas [31] yields $\mathcal{O}(\log^2 T)$ at best. We remark that the techniques and results developed for the degenerate $K$-armed bandit [10] do not carry over. Applying their method directly here yields a regret term $VT$ (rather than the sharper $V^2 T$) and cannot capture the dependence on the key quantity $\delta^2$, as they fail to exploit the special structure of CB-OPOD. We summarize the main technical challenges and highlights in Appendix C.1 and provide the full proof in Appendix E.

**Lower bound.** To establish a lower bound, we first specify the *admissible policy class*

$$\Pi^\circ = \Big\{ \pi \in \Pi : \sup_{(\theta'_*, \theta_*) \in \Theta^\dagger \times \Theta^\dagger} R^\pi_{\theta'_*, \theta_*}(T) \leq K_0 \sqrt{T}(\log T)^{\lambda_0}, \text{ for some constant } K_0, \lambda_0 \Big\}.$$

$\Pi^\circ$ contains every policy whose regret is uniformly bounded by $\tilde{\mathcal{O}}(\sqrt{T})$ over all pairs of offline and online demand parameters. Given offline data, $V \geq 0$ and $\delta^2$, we define

$$\mathcal{J} := \Big\{ (\theta'_*, \theta_*) \in \Theta^\dagger \times \Theta^\dagger : \|\theta'_* - \theta_*\| \leq V, \; \mathbb{E}_{x,y}[(\hat{p}(x,y) - p^*_\theta(x,y))^2] \in [(1-\xi)\delta^2, (1+\xi)\delta^2] \Big\}.$$

This class contains problem instances with specified bias upper bound $V$ and (approximate) generalized distance $\delta^2$. The following theorem provides a lower bound on the regret for every $\pi \in \Pi^\circ$, which incurs on some instance in $\mathcal{J}$.

**Theorem 2.** *Under Assumptions 1 and 2, $\forall \pi \in \Pi^\circ$ and any $\xi \in (0,1)$,*

$$\sup_{(\theta'_*, \theta_*) \in \mathcal{J}} R^\pi_{\theta'_*, \theta_*}(T) \in \begin{cases} \Omega\left(\delta^2 T\right) & \text{if } \delta^2 \lesssim \max\{V^2, \frac{1}{\lambda_{\min}(\hat{\Sigma})}\} \lesssim T^{-1/2}; \\ \tilde{\Omega}\left(\sqrt{T} \wedge V^2 T + \frac{T}{\lambda_{\min}(\hat{\Sigma}) + (N \wedge T)\delta^2}\right) & \text{otherwise,} \end{cases}$$

As discussed under Theorem 1, when the offline data is well conditioned, the lower bound in Theorem 2 matches the upper bound of Theorem 1 up to a linear factor in $d_1$. Key technical challenges and highlights are summarised in Appendix C.2 and the full proof appears in Appendix E.

# 4 General price elasticity

In this section we extend both the algorithmic design and the regret analysis to the general CB-OPOD setting with price elasticity of arbitrary dimension $d_2 \in \mathbb{Z}_+$.

## 4.1 The GCO3 algorithm: upper and lower regret bounds

Building on the idea of CO3, we propose the *General Contextual Online–Offline Pricing with Optimism* (GCO3) algorithm. Unlike CO3, GCO3 incorporates the biased offline dataset by constructing the confidence set as the intersection of only two ellipsoids.

---

**Algorithm 2** GCO3 Algorithm

---

**Input:** Same input as Algorithm 1.
   **for** $t = 1, 2, \ldots, T$ **do**
      Same procedure as the **for**-loop of Algorithm 1, except for updating $\bar{\mathcal{C}}_t = \Big\{ \theta \in \mathbb{R}^{d_1 + d_2} : \|\theta - \hat{\theta}_{t,N}\| \leq \hat{w}_{t,N}, \|\theta - \hat{\theta}_t\|_{\Sigma_t} \leq w_t \Big\}$.
   **end for**

---

The confidence set $\bar{\mathcal{C}}_t$ keeps the regret at most $\tilde{\mathcal{O}}(\sqrt{T})$ while fully exploiting the offline data to refine the estimate of $\theta_*$, giving the optimal dependence on the bias bound $V$, the dispersion $\lambda_{\min}(\hat{\Sigma})$,

and the horizon $T$. For the general CB-OPOD problem, however, a clean analogue of the instance-dependent distance $\delta^2$ is still unknown; such a quantity may not exist in every online-with-offline setting. In the $K$-armed bandit case, for example, [10] show that the fundamental difficulty is governed solely by $V$ and a term of the same order as $\lambda_{\min}(\hat{\Sigma})$. Defining an appropriate distance metric for richer contextual environments remains an attractive open problem. The following theorem gives an upper bound on the regret of Algorithm 2.

**Theorem 3.** *Let $\pi$ be Algorithm 2. Under Assumption 1, for any possible $(\theta'_*, \theta_*) \in \Theta^\dagger \times \Theta^\dagger$ such that $\|\theta'_* - \theta_*\| \le V$ and for any $T \ge 1$, $R^\pi_{\theta'_*, \theta_*}(T) \in \mathcal{O}\Big( (d_1 + d_2)\sqrt{T}\log T \wedge (V^2 T + \frac{(d_1 + d_2)T\log T}{\lambda_{\min}(\hat{\Sigma})}) \Big)$.*

As with Algorithm 1, Algorithm 2 never performs worse than the baseline $\tilde{\mathcal{O}}(\sqrt{T})$ rate. Theorem 3 sharpens this statement. If the bias bound is small with $V^2 \in \mathcal{O}(T^{-1/2})$ and the dispersion is strong with $\lambda_{\min}(\hat{\Sigma}) \in \Omega(\sqrt{T})$, the offline data are informative and Algorithm 2 attains regret strictly below $\tilde{\mathcal{O}}(\sqrt{T})$. Conversely, when either the bias bound is large, i.e., $V^2 \in \Omega(T^{-1/2})$ or the dispersion is weak, i.e., $\lambda_{\min}(\hat{\Sigma}) = \mathcal{O}(\sqrt{T})$, the offline data add little value and the Algorithm 2 cannot improve on $\tilde{\mathcal{O}}(\sqrt{T})$. Theorem 3 is proved by adapting the argument for Theorem 1; see Appendices E.1.1 and E.1.2. Because the steps are nearly identical, the proof is omitted. Notably, the design of Algorithm 2 and its regret–upper-bound analysis extend seamlessly to the *stochastic linear bandit with biased offline data*, thereby recovering the $K$-armed bandit result of [10]. Further details appear in Appendix I.

**Lower bound.** Given $V \ge 0$, we define $\bar{\mathcal{J}} := \big\{ (\theta'_*, \theta_*) \in \Theta^\dagger \times \Theta^\dagger : \|\theta'_* - \theta_*\| \le V \big\}$. The next theorem provides a regret lower bound that every policy $\pi \in \Pi^\circ$ must incur on some instance in $\bar{\mathcal{J}}$.

**Theorem 4.** *Under Assumption 1, $\forall \pi \in \Pi$, $\sup_{(\theta'_*, \theta_*) \in \bar{\mathcal{J}}} R^\pi_{\theta'_*, \theta_*}(T) \in \Omega\Big( \sqrt{T} \wedge \big( V^2 T + \frac{T}{\lambda_{\min}(\hat{\Sigma})} \big) \Big)$.*

The regret upper bound of GCO3 in Theorem 3 matches the lower bound of Theorem 4 up to a linear factor in dimension $(d_1 + d_2)$. The full proof appears in Appendix G.

## 4.2 Robustness

We now consider the setting in which the firm *does not* know the bias bound $V$. The objective is to design a policy whose regret is sub-linear for every exact bias $V_{\text{true}}$, and that beats the $\tilde{\mathcal{O}}(\sqrt{T})$ benchmark whenever $V_{\text{true}}$ is small. When the offline dispersion is weak, i.e. $\lambda_{\min}(\hat{\Sigma}) = \mathcal{O}(\sqrt{T})$, Theorem 4 shows that such performance is impossible. We therefore focus on the well-conditioned regime in which $\lambda_{\min}(\hat{\Sigma}) = \Theta(T^\beta)$ for some $\beta > \frac{1}{2}$. We next introduce the *Robust Contextual Online–Offline Pricing with Optimism* (**RCO3**) algorithm and explain the intuition behind its design.

---

**Algorithm 3** RCO3 Algorithm

---

**Input:** Test length $T'$ and all inputs of Algorithm 1 except the bias bound $V$.
    **for** $t = 1, 2, \ldots, T'$ **do**
        Charge $p_t$ uniformly from $\{l, u\}$;
    **end for**
    Calculate $\hat{\theta}'_* = (\hat{\alpha}'_*, \hat{\beta}'_*) = \hat{\theta}_{0,N}$ and $\hat{\theta}_* = \hat{\theta}_{T'}$.
    **if** $\|\hat{\theta}'_* - \hat{\theta}_*\| \le 2f$, where $f$ is defined in Lemma 14 **then**
        Charge $p_t = \arg\max_{p \in [l,u]} p \cdot (\hat{\alpha}'^\top_* x_t + \hat{\beta}'^\top_* y_t p)$ for $t = T' + 1, \ldots, T$.
    **else**
        Run pure online algorithm [2] for $t = T' + 1, \ldots, T$.
    **end if**

---

Algorithm 3 starts with a *test phase* of length $T' = \Theta(T^\alpha)$ for some $\alpha \in (0, \frac{1}{2})$. Choosing $\alpha < \frac{1}{2}$ keeps the test regret $\mathcal{O}(T') = o(\sqrt{T})$, while $\alpha > 0$ prevents linear growth of total regret. Prices in this phase are sampled uniformly, producing an estimate of $\theta_*$. With this estimate we test whether the exact bias satisfies $V_{\text{true}} \gtrsim T^{-\alpha/2}$: if $V_{\text{true}} \in \Omega(T^{-\alpha/2})$, the algorithm switches to a pure-online policy; if $V_{\text{true}} \in \mathcal{O}(T^{-\alpha/2})$, it relies on the offline regression estimate $\hat{\theta}'_*$ and prices accordingly. The next theorem specifies the admissible choices of $T'$ and bounds the regret of Algorithm 3.

**Theorem 5.** *Given $\beta > 1/2$, the optimal choices of $\alpha$ is $\alpha \in (\max\{0, 1 - \beta\}, \frac{1}{2})$. Let $\pi$ be the Algorithm 3. Under Assumption 1, for any $T \geq 1$, and for any possible value of $(\theta'_*, \theta_*) \in \Theta^\dagger \times \Theta^\dagger$,*

$$R^\pi_{\theta'_*, \theta_*}(T) \in \begin{cases} \mathcal{O}\big((d_1 + d_2)\sqrt{T}\log T\big), & \text{if } V^2_{\text{true}} \gtrsim T^{-\alpha}; \\ \tilde{\mathcal{O}}\big(T^\alpha + V^2_{\text{true}}T\big), & \text{otherwise.} \end{cases}$$

Theorem 5 shows that the optimal test length, $T' = \Theta(T^\alpha)$, lies in the interval $\alpha \in \big(\max\{0, 1 - \beta\}, \frac{1}{2}\big)$. Within this range there is no dominant choice–ultimately it depends on risk tolerance. Choosing a smaller $\alpha$ reduces regret whenever $V^2_{\text{true}} \lesssim T^{\alpha-1}$, but raises the worst-case rate to $T^{1-\alpha}$. Fixing $\alpha$, the regret as a function of $V^2_{\text{true}}$ exhibits three distinct phases. *Phase 1:* when $V^2_{\text{true}} \lesssim T^{\alpha-1}$, the offline data is highly informative and the regret is dominated by the $T'$ test period. *Phase 2:* for $T^{\alpha-1} \lesssim V^2_{\text{true}} \lesssim T^\alpha$ the value of the offline data wanes; the regret rises and peaks around $V^2_{\text{true}} \asymp T^{-\alpha}$. *Phase 3:* once $V^2_{\text{true}} \gtrsim T^{-\alpha}$ the test phase correctly detects the shift and the algorithm reverts to pure-online behaviour, so the regret falls back to $\tilde{\mathcal{O}}(\sqrt{T})$. This phase transition is visualised in Figure 1. To the best of our knowledge, Theorem 5 provides the first robust guarantee for CB-OPOD. The proof appears in Appendix H.

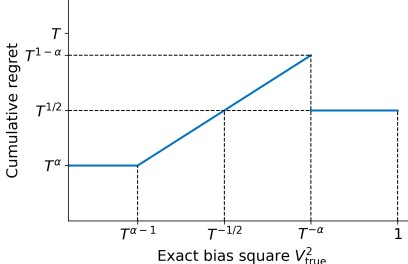

Figure 1: Piecewise regret bound as a function of exact bias square $V^2_{\text{true}}$

## 5 Numerical experiments

In this section, we conduct numerical experiments on synthetic data to assess our algorithms, leaving experiments on real data for future work. Specifically, we evaluate **CO3** (or **GCO3** when $d_2 > 1$) against four baselines: 1) **UCB**: a pure online UCB policy that ignores the offline data, 2) **UCB-Offline**: the UCB policy of [7, 31], which forms its single confidence ellipsoid from the combined offline and online data, 3) **TS**: a pure online Thompson-sampling policy and 4) **TS-Offline**: Thompson-sampling with a prior fitted to the offline data.

We randomly generate two online models: 1) a scalar price elasticity case with $d_2 = 1$ and 2) a general case with $d_2 = 5$. In both cases the offline data is drawn from a market with exact bias $V_{\text{true}} = \Theta(T^{-5/16})$ and dispersion $\lambda_{\min}(\hat{\Sigma}) = \Theta(T)$. We compare **CO3/GCO3** against four baselines under two bias-bound settings: a *tight* bound $V = 1.1\,V_{\text{true}}$ and a *loose* bound $V = 10\,V_{\text{true}}$. Every configuration is averaged over 20 independent trials with $T = 1000$ rounds; shaded bands indicate 2-sigma error bars. Figures 2(a)–(b) reveal several trends. First, **UCB-Offline** and **TS-Offline** rely uncritically on the biased offline data and accumulate regret faster than the pure-online baselines, illustrating the danger of ignoring distributional shift. Second, when the bias bound is tight $(V = 1.1\,V_{\text{true}})$, **CO3/GCO3** decisively outperform every baseline, in line with Theorems 1 and 3. Finally, even under a loose bound $(V = 10\,V_{\text{true}})$, **CO3/GCO3** track the performance of **UCB** and incur no additional regret, demonstrating the algorithms' "never-worse" safety property.

We next evaluate **RCO3** in the general setting with $d_2 = 5$. A single online model is randomly generated and fixed, and ten independent offline datasets are generated, each with dispersion $\lambda_{\min}(\hat{\Sigma}) = \Theta(T)$ but different exact biases $V^2_{\text{true}} \in \Theta(T^{-n/5})$ for $n = 0, \ldots, 9$. For every offline-online instance we run **RCO3** with a test phase of length $T' = \Theta(T^{1/4})$ $(\alpha = 1/4)$ and compare it to the pure-online **UCB** baseline, repeating each policy 20 times. Figure 2(c) reports the mean cumulative regret at $T = 5000$ with a 2-sigma error bar as a function of $V_{\text{true}}$. The empirical pattern matches Theorem 5. When $V^2_{\text{true}} \lesssim T^{-3/4}$ the offline data is highly informative and **RCO3**

outperforms **UCB**. As the bias increases to the intermediate range $T^{-3/4} \lesssim V_{\text{true}}^2 \lesssim T^{-1/4}$, the value of the offline data diminishes; the regret of **RCO3** rises and peaks near $V_{\text{true}}^2 \asymp T^{-1/4}$, where the test phase is just able to detect the shift. We highlight that the percentage drop in regret for small bias ($V_{\text{true}}^2 \lesssim T^{-3/4}$) is comparable to the percentage increase for large bias ($T^{-3/4} \lesssim V_{\text{true}}^2 \lesssim T^{-1/4}$), even though the absolute loss in the latter case is higher. For larger biases $\left(V_{\text{true}}^2 \gtrsim T^{-1/4}\right)$ the test correctly rejects the offline data and the regret of **RCO3** returns to the **UCB** level.

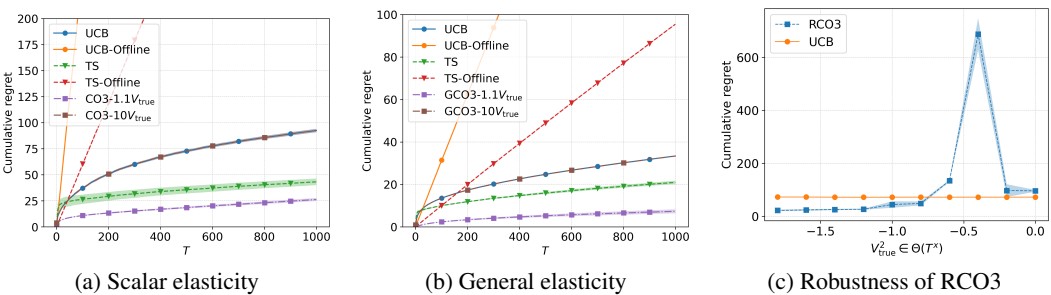

| (a) Scalar elasticity | (b) General elasticity | (c) Robustness of RCO3 |

Figure 2: Performances of CO3, GCO3, and RCO3 compared with baseline algorithms.

## 6 Conclusion

We study *contextual online pricing with biased offline data* (CB-OPOD). For the scalar price elasticity case, we introduce an instance-specific metric $\delta^2$ that quantifies the gap between the offline data and the (unknown) online optimal pricing strategy and an optimism-based policy attains the instance–optimal regret $\tilde{\mathcal{O}}\big(d_1\sqrt{T} \wedge (V^2T + \frac{d_1 T}{\lambda_{\min}(\hat{\Sigma})+(N \wedge T)\delta^2})\big)$. For general price elasticity, we show that the worst-case minimax rate reduces to $\tilde{\mathcal{O}}\big((d_1+d_2)\sqrt{T} \wedge (V^2T + \frac{(d_1+d_2)T}{\lambda_{\min}(\hat{\Sigma})})\big)$ and provide a matching algorithm. When bias bound $V$ is unknown, we introduce a robust variant that retains sub-linear regret and outperforms purely online policies whenever the true bias is small. Our analysis carries over verbatim to stochastic linear bandits with biased offline data.

Future work includes extending to general nonlinear demand models, studying model misspecification, sharpening the dimension dependence in the CB-OPOD lower bound and, more broadly, developing a suitable instance-specific distance metric for general online-with-offline learning. Applying our theoretical results to real-world datasets is also a promising avenue for future research.

## Acknowledgments and Disclosure of Funding

Q. Xie and Y. Zhang are supported in part by National Science Foundation (NSF) grants CNS-1955997, ECCS-2339794, and ECCS-2432546.

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

## A  Additional notations

This section summarizes the notation used for algorithmic construction and the theoretical analysis.

| **Scalars** | |
|---|---|
| $L$ | $\sqrt{x_{\max}^2 + y_{\max}^2 u^2}$ |
| **Gram Matrices** | |
| $\Sigma_t$ | $\lambda I + \sum_{s=1}^t \begin{bmatrix} x_s x_s^\top & x_s p_s y_s^\top \\ y_s p_s x_s^\top & y_s p_s^2 y_s^\top \end{bmatrix}$ |
| $\Sigma_{t,N}$ | $\Sigma_t + \hat{\Sigma}$ |
| **Estimators** | |
| $\hat{\theta}_t$ | $\arg\min_{\theta \in \mathbb{R}^{d_1+d_2}} \left\{ \lambda\|\theta\|^2 + \sum_{s=1}^t (D_t - (\alpha^\top x_t + \beta^\top y_t p_t))^2 \right\}$ |
| $\hat{\theta}_{t,N}$ | $\arg\min_{\theta \in \mathbb{R}^{d_1+d_2}} \left\{ \lambda\|\theta\|^2 + \sum_{n=1}^N (\hat{D}_n - (\alpha^\top \hat{x}_n + \beta^\top \hat{y}_n \hat{p}_n))^2 \right.$ $\left. + \sum_{s=1}^t (D_t - (\alpha^\top x_t + \beta^\top y_t p_t))^2 \right\}$ |
| **Confidence radius** | |
| $w_t$ | $\sqrt{\lambda} \cdot \sqrt{\alpha_{\max}^2 + \beta_{\max}^2} + \sqrt{2\log(3/\epsilon) + (d_1+d_2)\log\left(1 + \frac{tL^2}{(d_1+d_2)\lambda}\right)}$ |
| $w_{t,N}$ | $\frac{\lambda\sqrt{\alpha_{\max}^2+\beta_{\max}^2}}{\sqrt{\lambda+\lambda_{\min}(\hat{\Sigma})}} + \frac{\lambda_{\max}(\hat{\Sigma})V}{\sqrt{\lambda+\lambda_{\max}(\hat{\Sigma})}} + \sqrt{2\log(6/\epsilon) + (d_1+d_2)\log\left(1 + \frac{tL^2}{(d_1+d_2)\lambda}\right)}$ $+ R\sqrt{d_1+d_2} + R\sqrt{2\log(6/\epsilon)}$ |
| $\hat{w}_{t,N}$ | $\frac{\lambda\sqrt{\alpha_{\max}^2+\beta_{\max}^2}}{\lambda+\lambda_{\min}(\hat{\Sigma})} + V + \frac{\sqrt{2\log(6/\epsilon)+(d_1+d_2)\log\left(1+\frac{tL^2}{(d_1+d_2)\lambda}\right)}}{\sqrt{\lambda+\lambda_{\min}(\hat{\Sigma})}} + \frac{R\sqrt{d_1+d_2}+R\sqrt{2\log(6/\epsilon)}}{\sqrt{\lambda+\lambda_{\min}(\hat{\Sigma})}}$ |

## B  Additional related work

**Online learning with external (biased) informations**   Online learning problems have attracted significant attention in recent years, and a growing body of work explores how external information can improve online learning. One line of research assumes that the external information is *unbiased*, demonstrating that it can yield lower regret than purely online approaches [24, 19, 7, 29, 31]. Another line focuses on *potentially biased* external information, aiming to design algorithms that perform no worse than purely online methods, yet can outperform them when the external information closely aligns with the online model [23, 32, 30, 10].

Among this second line of work, [23, 30] study online learning with a (biased) loss predictor, assuming that for each time $t$, one has access to a loss predictor $m_t$ for the i.i.d. true loss $l_t$. They measure bias as $T\,\mathbb{E}[\|m_1 - l_1\|^2]$ and show that if this bias grows more slowly than $T$, the regret can be further reduced. However, with only offline data, it is impossible to generate a predictor whose bias grows more slowly than $T$, since offline data cannot forecast online fluctuations. Consequently, this approach does not guide us in designing an algorithm that meets the aforementioned goal.

Another line of research focuses on *hybrid transfer reinforcement learning*, including both empirical work [12] and theoretical work [8, 22]. In the theoretical results, [8] studies offline data containing only transitions (no rewards) from another Markov decision process (MDP), while [22] examines offline data from an MDP with the same reward function but different transitions. Neither scenario reduces to our setting. Moreover, these works typically focus on bounding $\mathbb{E}[\|q_k - q^*\|_\infty]$ or the sample complexity required to learn an $\epsilon$-optimal policy. In contrast, our work uses *regret* as the performance metric.

Bayesian policies assume a known prior distribution for unknown parameters and update their beliefs through online observations, making them popular for leveraging offline data in online learning.

Thompson sampling (TS) is a well-known method of this type. However, prior work [20, 16, 26] demonstrates that a misspecified prior can lead to higher regret bounds than those of purely online TS. In particular, [26] shows that TS with a misspecified prior of magnitude $\epsilon$ can incur an additional $\mathcal{O}(\epsilon T^2)$ regret. This exceeds our results in Table 1, where the additional regret is on the order of $\mathcal{O}\left(V^2 T + \frac{T}{\lambda_{\min}(\hat{\Sigma})}\right)$, and the total regret never grows faster than $\mathcal{O}(\sqrt{T})$.

[32] studies contextual bandits but does not improve on the standard online regret and require a restrictive offline log (see also [10, Appendix A.1]). By extending our Algorithm 2 and analysis to the stochastic linear bandit with biased offline data, we obtain regret upper bounds that subsume the $K$-armed result of [10]; see Appendix I for details.

**Online pricing**  Online pricing has also garnered significant interest in recent years. In this setting, a firm is initially uncertain about the parameters of the demand model and uses price experimentation to learn them through empirical market responses. [15] establish a $\tilde{\Theta}(\sqrt{T})$ minimax regret bound for the *non-contextual* online pricing problem, and [2] extend this result to the *contextual* setting. Building on [15], [7] study the regret when the firm also has access to an offline dataset from the same market. Subsequently, [31] integrate *unbiased* offline data with contextual online pricing but assume that the offline data are i.i.d. and generated by a fixed policy. This assumption can be unrealistic when the offline data come from an online pricing algorithm that dynamically adjusts its prices. Moreover, under a fixed pricing policy, they implicitly assume limited dispersion of offline prices and features, resulting in a larger minimax regret than ours and preventing them from recovering the non-contextual result of [7].

In contrast, our approach imposes no additional assumptions on the offline data and allows it to come from an entirely different market. We further achieve a tighter minimax regret bound, and by setting the valid bias bound $V = 0$, our framework recovers the non-contextual setting of [7].

## C   Technical challenges and highlights

In this section we offer a detailed technical comparison between our results and the existing analyses of online pricing and argue the technical challenges and highlights of our setting. The discussion is organised into two parts: 1) derivation of the regret upper bound and 2) derivation of the regret lower bound.

### C.1   Upper bound

A central difficulty in deriving a fine-grained regret upper bound that depends on the generalized distance $\delta^2$ is to identify a *valid empirical offline price policy*. In OPOD, [7] implicitly set this policy to the sample mean $\frac{1}{N} \sum_{n=1}^{N} \hat{p}_n$, but offered no intuition for its validity. In the CB-OPOD setting that policy is no longer appropriate. We show that a valid empirical policy naturally arises as the solution to

$$\hat{A} = \arg\min_{\alpha \in \mathbb{R}^d} \sum_{n=1}^{N} (\alpha^\top \hat{x}_n - \hat{y}_n \hat{p}_n)^2 = \hat{\Sigma}_{x,x} \hat{\Sigma}_{x,y}$$

and the valid empirical offline policy is defined as $\hat{p}(x, y) = \frac{\hat{A}^\top x}{y}$. When the problem reduces to OPOD (all features equal to 1), this policy simplifies to $\hat{p}(x, y) = \hat{p} = \frac{1}{N} \sum_{n=1}^{N} \hat{p}_n$, recovering the classical choice. Thus (3) reveals the more general structure underlying offline pricing data.

Although our formulation of the distance $\delta^2$ (3) aligns with that in [31], our choice of $\hat{p}$ differs. Specifically, [31] assumes that the offline data are i.i.d. under a *single* offline pricing policy $\hat{p}$, an assumption that can be unrealistic when data are generated by a dynamically changing online algorithm. In contrast, we define $\hat{p}$ *after* collecting the offline data, without imposing further restrictions on how the data are obtained.

The quantities $\hat{p}(x, y)$ and $\delta^2$ are central to linking contextual offline data to online pricing performance. In particular, Lemma 9 (Appendix D) shows how $\delta^2$ affects the accuracy of the online parameter estimate. While the statement of Lemma 9 resembles [7, Lemma 2], our proof is fundamentally different: it accounts for contextual covariates and explicitly captures the dependence on feature dimension.

Beyond this, although our proof strategy follows the general blueprint of [7], handling high-dimensional covariates together with *biased* offline data introduces substantial new technical challenges. While [31] also study a contextual setting, they impose restrictive assumptions on the offline data and derive a looser upper bound; consequently, most of their arguments do not extend to our more general framework.

## C.2 Lower bound

Previous lower-bound proofs for contextual online pricing [2, 31] use the multivariate van Trees inequality with a carefully chosen vector function $C(\theta)$. To apply the inequality, they inflate the offline feature dimension to $T$ (or $T + N$), destroying the original geometry of the offline Gram matrix and yielding bounds that depend on the dimension while obscuring the effect of offline dispersion, which is too conservative. We propose a new construction of $C(\theta)$ that works *without* feature augmentation, preserves the true dispersion of the offline Gram matrix, and produces sharper bounds for contextual pricing with *biased* offline data. How to simultaneously capture the explicit dependence on the ambient dimension remains open and is left for future work.

Furthermore, obtaining the *optimal* dependence on the bias bound $V$ simuteneously with the *optimal* dependence on the time $T$, dispersion $\lambda_{\min}(\hat{\Sigma})$, generalized distance $\delta^2$ and offline sample size $N$ is a distinctive contribution of this work and substantially complicates the analysis; see Appendices F and G for details.

# D  Preliminaries

This section collects several results that are useful throughout the proof.

## D.1  Preliminary linear algebra results

To begin with, in this subsection, we collect several linear algebra results.

**Lemma 1.** *Given a positive semi-definite matrix $M \in \mathbb{R}^{d \times d}$, a positive constant $\lambda > 0$ and an integer $k \in \mathbb{Z}$, the spectrum of $M(M + \lambda I_d)^k M$ is $\{\lambda_i^2(\lambda_i + \lambda)^k\}_{i \in [d]}$, where $\{\lambda_i\}_{i \in [n]}$ is the spectrum of M.*

*Proof.* Because $M$ is positive semi-definite, there exist an eigen-decomposition $M = QDQ^{-1}$. Therefore, we have

$$M(M + \lambda I)^k M = QDQ^{-1}(Q(D + \lambda I_d)Q^{-1})^k QDQ^{-1}$$
$$= QD(D + \lambda I_d)^k DQ^{-1}.$$

$D(D + \lambda I_d)^k D$ is a diagonal matrix whose diagonal entries are $\{\lambda_i^2(\lambda_i + \lambda)^k\}_{i \in [d]}$, thereby completing the proof of Lemma 1. □

**Lemma 2.** *Let $A, B : \mathbb{R}^d \to \mathbb{R}$ be two quadratic functions of the form*

$$A(x) = (x - u_A)^\top Q_A(x - u_A) + c_A, \quad B(x) = (x - u_B)^\top Q_B(x - u_B) + c_B,$$

*where $Q_A, Q_B$ are positive definite matrixes and $c_B \geq 0$. Define $x_A = \arg\min_x A(x)$ and $x_{A+B} = \arg\min_x \big(A(x) + B(x)\big)$. Then, we have*

$$A\big(x_{A+B}\big) + B\big(x_{A+B}\big) - A(x_A) \geq \frac{\lambda_{\min}(Q_B^{-1/2} Q_A Q_B^{-1/2})}{1 + \lambda_{\min}(Q_B^{-1/2} Q_A Q_B^{-1/2})} \cdot B(x_A).$$

*Proof.* Because $x_A = \arg\min_x A(x), \quad x_{A+B} = \arg\min_x \big(A(x) + B(x)\big)$, we have

$$x_A = u_A \quad \text{and} \quad Q_A(x_{A+B} - u_A) + Q_B(x_{A+B} - u_B) = 0.$$

Let $\delta = u_A - u_B$ and $\theta = x_{A+B} - u_B$. Then, the above equalities imply that $Q_A(\theta - \delta) + Q_B\theta = 0$, which further implies that $\theta = (Q_A + Q_B)^{-1} Q_A \delta$. Therefore, we have

$$A\big(x_{A+B}\big) + B\big(x_{A+B}\big) - A(x_A)$$

$$
\begin{aligned}
&=(x_{A+B} - u_A)^\top Q_A(x_{A+B} - u_A) + (x_{A+B} - u_B)^\top Q_B(x_{A+B} - u_B) + c_B \\
&=(\theta - \delta)^\top Q_A(\theta - \delta) + \theta^\top Q_B\theta + c_B \\
&=\delta^\top Q_B\theta + c_B \\
&=\delta^\top Q_B(Q_A + Q_B)^{-1}Q_A\delta + c_B.
\end{aligned}
$$

Next, we will find the the largest possible $c$ such that the following inequality holds:

$$
x^\top(Q_B(Q_A + Q_B)^{-1}Q_A - cQ_B)x \geq 0, \quad \forall x \in \mathbb{R}^d.
$$

We define $M = Q_B^{1/2}(Q_A + Q_B)^{-1}Q_AQ_B^{-1/2}$. For two matrices $M$ and $N$, we write $M \succeq N$ if $M - N$ is positive semidefinite. Then the condition $Q_B(Q_A + Q_B)^{-1}Q_A \succeq cQ_B$ is equivalent to $Q_B^{1/2}(Q_A + Q_B)^{-1}Q_AQ_B^{-1/2} \succeq cI$, which simplifies to $M \succeq cI$. Therefore, the largest possible largest $c$ is $\lambda_{\max}(M)$. Meanwhile, by defining $N = Q_B^{-1/2}Q_AQ_B^{-1/2}$, we have $Q_A = Q_B^{1/2}NQ_B^{1/2}$ and $Q_A + Q_B = Q_B^{1/2}(I + N)Q_B^{1/2}$. Therefore, we have

$$
M = (I + N)^{-1}N.
$$

Because $N$ commutes with $I + N$, i.e. $N(I + N) = (I + N)N$, $N$ and $I + N$ are simultaneously diagonalizable, which implies

$$
\lambda_i(M) = \frac{\lambda_i(N)}{1 + \lambda_i(N)}.
$$

Because $N$ is positive semi-definite, we have

$$
c = \lambda_{\min}(M) = \frac{\lambda_{\min}(N)}{1 + \lambda_{\min}(N)} = \frac{\lambda_{\min}(Q_B^{-1/2}Q_AQ_B^{-1/2})}{1 + \lambda_{\min}(Q_B^{-1/2}Q_AQ_B^{-1/2})}.
$$

Finally, because $c_B \geq 0$, we have

$$
\begin{aligned}
A(x_{A+B}) + B(x_{A+B}) - A(x_A) &= \delta^\top Q_B(Q_A + Q_B)^{-1}Q_A\delta + c_B \\
&\geq \frac{\lambda_{\min}(Q_B^{-1/2}Q_AQ_B^{-1/2})}{1 + \lambda_{\min}(Q_B^{-1/2}Q_AQ_B^{-1/2})}\delta^\top Q_B\delta + c_B \\
&\geq \frac{\lambda_{\min}(Q_B^{-1/2}Q_AQ_B^{-1/2})}{1 + \lambda_{\min}(Q_B^{-1/2}Q_AQ_B^{-1/2})}(\delta^\top Q_B\delta + c_B) \\
&= \frac{\lambda_{\min}(Q_B^{-1/2}Q_AQ_B^{-1/2})}{1 + \lambda_{\min}(Q_B^{-1/2}Q_AQ_B^{-1/2})}B(x_A),
\end{aligned}
$$

which finishes the proof of Lemma 2. $\qquad\square$

**Lemma 3.** *Let $x \in \mathbb{R}^{d_1}$ and $y \in \mathbb{R}^{d_2}$ be random vectors with finite second moments. Assume that $\mathbb{E}[xx^\top]$ is invertible. Then*

$$
\mathbb{E}[yy^\top] - \mathbb{E}[yx^\top]\mathbb{E}[xx^\top]^{-1}\mathbb{E}[xy^\top] \succeq 0.
$$

*Proof.* Let $z = y - \mathbb{E}[yx^\top]\mathbb{E}[xx^\top]^{-1}x$. Notice that

$$
\begin{aligned}
\mathbb{E}[zz^\top] &= \mathbb{E}[(y - \mathbb{E}[yx^\top]\mathbb{E}[xx^\top]^{-1}x)(y - \mathbb{E}[yx^\top]\mathbb{E}[xx^\top]^{-1}x)^\top] \\
&= \mathbb{E}[yy^\top] - \mathbb{E}[yx^\top\mathbb{E}[xx^\top]^{-1}\mathbb{E}[xy^\top]] - \mathbb{E}[\mathbb{E}[yx^\top]\mathbb{E}[xx^\top]^{-1}xy^\top] \\
&\quad + \mathbb{E}[\mathbb{E}[yx^\top]\mathbb{E}[xx^\top]^{-1}xx^\top\mathbb{E}[xx^\top]^{-1}\mathbb{E}[xy^\top]] \\
&= \mathbb{E}[yy^\top] - \mathbb{E}[yx^\top]\mathbb{E}[xx^\top]^{-1}\mathbb{E}[xy^\top].
\end{aligned}
$$

Because $\mathbb{E}[zz^\top] \succeq 0$, we complete the proof of Lemma 3. $\qquad\square$

## D.2 Preliminary contextual online pricing results

In this subsection, we collect several preliminary lemmas related to online pricing with offline data. All proofs of these lemmas are postponed to the end of this subsection.

First, we present three lemmas that apply to the general contextual online pricing setting, i.e., $\mathcal{X} \times \mathcal{Y} \subseteq \mathbb{R}^{d_1} \times \mathbb{R}^{d_2}$ with $d_2 \in \mathbb{Z}^+$.

**Lemma 4.** *Under Assumption 1, for all $(x, y) \in \mathcal{X} \times \mathcal{Y}, \theta, \theta' \in \Theta^\dagger$, we have*

$$|p_\theta^*(x, y) - p_{\theta'}^*(x, y)| \leq \frac{\sqrt{y_{\max}^2 u_\alpha^2 + x_{\max}^2 u_\beta^2} \|\theta - \theta'\|}{2l_\beta^2}.$$

**Lemma 5.** *Recall the confidence set $\mathcal{C}_t$ defined in (4). Under Assumption 1, with probability at least $1 - \delta$ we have $\theta_* \in \mathcal{C}_t \cap \Theta^\dagger$ for all $t \in \mathbb{N}$.*

**Lemma 6.** *Consider Algorithm 1 and assume Assumption 1 holds. Then, for any $\xi \in (0, e^{-2})$, the following event occurs for every $t \in [T]$ with probability at least $1 - \xi$.*

$$\sup_{\theta \in \mathcal{C}_t \cap \Theta^\dagger} \mathbb{E}_x \left[ \sum_{n=1}^N \left( (\alpha - \alpha_*)^\top \hat{x}_n + (\beta - \beta_*)^\top \hat{y}_n \hat{p}_n \right)^2 + \sum_{s=1}^t \left( (\alpha - \alpha_*)^\top x + (\beta - \beta_*)^\top y p_s(x) \right)^2 \mid \mathcal{F}_{t-1} \right] \leq K\eta_t^2,$$

(5)

*where $p_s$ is any empirical pricing function in period $s \in [t]$ that is $(s-1)$–measurable, $K$ is an absolute constant, and $\eta_t^2 = w_{t,n}^2 + (d_1 + d_2) \log T + \log(t/\xi)$.*

The remaining lemmas specialize to the scalar price elasticity setting, i.e., $d_2 = 1$.

**Lemma 7.** *Under Assumption 1, and with $\delta^2$ defined in (3), we obtain $\delta^2 \leq \frac{2\beta_{\max}^2 x_{\max}^4 y_{\max}^2}{l_\beta^2 c^2} + 2u^2$.*

**Lemma 8.** *Given the offline data $\{(\hat{x}_n, \hat{y}_n, \hat{p}_n)\}_{n \in [N]}$ and any $\alpha \in \mathbb{R}^{d_1}$ and $\beta \in \mathbb{R}$, we have*

$$\sum_{n=1}^N (\alpha^\top \hat{x}_n + \beta \hat{y}_n \hat{p}_n)^2 \geq \sum_{n=1}^N (\alpha^\top \hat{x}_n + \beta \hat{A}^\top \hat{x}_n)^2,$$

*where $\hat{A}$ is defined in equation (3).*

**Lemma 9.** *Suppose $\delta^2 \geq \sqrt{\frac{\bar{C} K \eta_T^2}{N}}$ and $N\delta^2 \geq \lambda_{\min}(\hat{\Sigma})$, where $\bar{C} > 0$ is a constant. Under Assumptions 1 and 2, there exists a positive constant $C$ such that if $\theta_* \in \mathcal{C}_t$ for each $t \in [T]$, two sequences of events $\{U_{t,1} : t \in [T]\}$ and $\{U_{t,2} : t \in [T]\}$ hold, where*

$$U_{t,1} = \left\{ \mathbb{E}_{x_t, y_t} [(\hat{p}(x_t, y_t) - p_t(x_t, y_t))^2] \geq (\frac{1}{4} \wedge \frac{l_\beta l y_{\min}^2}{8 y_{\max}^2 u_\beta u}) \delta^2 \right\}$$

$$U_{t,2} = \left\{ \mathbb{E}_{x_t} [\|\tilde{\theta}_t - \theta_*\|^2] \leq \frac{C K \eta_{t-1}^2}{\lambda_{\min}(\hat{\Sigma}) + (N \wedge (t-1))\delta^2} \right\}$$

**Lemma 10.** *Given the offline dataset $\{(x_n, y_n, p_n)\}_{n=1}^N$, which satisfies Assumption 2, and under Assumption 1, we have*

$$\sum_{n=1}^N (\hat{p}(\hat{x}_n, \hat{y}_n) - p_{\theta_*}^*(\hat{x}_n, \hat{y}_n))^2 \leq \frac{N x_{\max}^2 y_{\max}^2}{y_{\min}^2 \lambda_{\min}(\mathbb{E}[xx^T])} \mathbb{E}_{x,y}[(\hat{p}(x, y) - p_{\theta_*}^*(x, y))^2] \quad \text{and}$$

$$\sum_{n=1}^N (\hat{p}(\hat{x}_n, \hat{y}_n) - \hat{p}_n)^2 \leq \frac{\max\{c, 2(x_{\max}^2 + u^2 y_{\max}^2)\}}{c y_{\min}^2} \cdot \lambda_{\min}(\hat{\Sigma}).$$

### D.2.1 Proof of Lemma 4

For any $x, y \in \mathcal{X} \times \mathcal{Y}$ and $\theta, \theta' \in \Theta^\dagger$, we have

$$|p_\theta^*(x, y) - p_{\theta'}^*(x, y)| = |\frac{\alpha^\top x}{-2\beta^\top y} - \frac{\alpha'^\top x}{-2\beta'^\top y}| = |\frac{\alpha'^\top x \beta^\top y - \alpha^\top x \beta'^\top y}{2\beta^\top y \beta'^\top y}|$$

$$\leq |\frac{\alpha'^\top x \beta^\top y - \alpha^\top x \beta'^\top y}{2l_\beta^2}|$$

$$\leq |\frac{\alpha'^\top x \beta^\top y - \alpha^\top x \beta^\top y + \alpha^\top x \beta^\top y - \alpha^\top x \beta'^\top y}{2l_\beta^2}|$$

$$\leq \frac{\|\alpha - \alpha'\| x_{\max} u_\beta + \|\beta - \beta'\| y_{\max} u_\alpha}{2l_\beta^2}$$

$$\leq \frac{\sqrt{y_{\max}^2 u_\alpha^2 + x_{\max}^2 u_\beta^2}\|\theta - \theta'\|}{2l_\beta^2},$$

thereby completing the proof of Lemma 4.

### D.2.2 Proof of Lemma 5

We prove Lemma 5 by showing that each of the following events holds with probability at least $1 - \epsilon/3$: (1) $\|\theta_* - \hat\theta_{t,N}\|_{\Sigma_{t,N}} \leq w_{t,N}$, (2) $\|\theta_* - \hat\theta_{t,N}\| \leq \hat w_{t,N}$ and (3) $\|\theta_* - \hat\theta_t\|_{\Sigma_t} \leq \hat w_t$. Since event (3) follows directly from [17, Theorem 20.5], we omit its proof and focus on proving events (1) and (2) in what follows.

**(1) $\|\theta_* - \hat\theta_{t,N}\|_{\Sigma_{t,N}} \leq w_{t,N}$ with probability at least $1 - \epsilon/3$.**

$$\theta_* - \hat\theta_{t,N} = \theta_* - \Sigma_{t,N}^{-1}\Big(\sum_{n=1}^N \begin{bmatrix} \hat x_n \\ \hat y_n \hat p_n \end{bmatrix} \hat D_i + \sum_{s=1}^t \begin{bmatrix} x_t \\ y_t p_t \end{bmatrix} D_i\Big)$$

$$= \theta_* - \Sigma_{t,N}^{-1}\Big(\sum_{n=1}^N \begin{bmatrix} \hat x_n \\ \hat y_n \hat p_n \end{bmatrix}\big([\; \hat x_n^\top \; \hat p_n \hat y_n^\top \;]\theta'_* + \hat\epsilon_n\big) + \sum_{s=1}^t \begin{bmatrix} x_t \\ y_t p_t \end{bmatrix}\big([\; x_t^\top \; p_t y_t^\top \;]\theta_* + \epsilon_t\big)\Big)$$

$$= \Sigma_{t,N}^{-1}\Big(\lambda\theta_* + \hat\Sigma(\theta_* - \theta'_*) + \sum_{n=1}^N \begin{bmatrix} \hat x_n \\ \hat y_n \hat p_n \end{bmatrix}\hat\epsilon_n + \sum_{s=1}^t \begin{bmatrix} x_t \\ y_t p_t \end{bmatrix}\epsilon_t\Big). \tag{6}$$

Then, we have

$$\|\theta_* - \hat\theta_{t,N}\|_{\Sigma_{t,N}} \leq \lambda\|\theta_*\|_{\Sigma_{t,N}^{-1}} + \|\hat\Sigma(\theta_* - \theta'_*)\|_{\Sigma_{t,N}^{-1}} + \|\sum_{n=1}^N \begin{bmatrix} \hat x_n \\ \hat y_n \hat p_n \end{bmatrix}\hat\epsilon_n + \sum_{s=1}^t \begin{bmatrix} x_t \\ y_t p_t \end{bmatrix}\epsilon_t\|_{\Sigma_{t,N}^{-1}}$$

$$\leq \lambda\|\theta_*\|_{V_{0,N}^{-1}} + \|\hat\Sigma(\theta_* - \theta'_*)\|_{V_{0,N}^{-1}} + \|\sum_{n=1}^N \begin{bmatrix} \hat x_n \\ \hat y_n \hat p_n \end{bmatrix}\hat\epsilon_n + \sum_{s=1}^t \begin{bmatrix} x_t \\ y_t p_t \end{bmatrix}\epsilon_t\|_{\Sigma_{t,N}^{-1}}$$

$$\leq \frac{\lambda\sqrt{\alpha_{\max}^2 + \beta_{\max}^2}}{\sqrt{\lambda + \lambda_{\min}(\hat\Sigma)}} + \frac{\lambda_{\max}(\hat\Sigma)V}{\sqrt{\lambda + \lambda_{\max}(\hat\Sigma)}} + \underbrace{\|\sum_{n=1}^N \begin{bmatrix} \hat x_n \\ \hat y_n \hat p_n \end{bmatrix}\hat\epsilon_n\|_{V_{0,N}^{-1}}}_{T_1} + \underbrace{\|\sum_{s=1}^t \begin{bmatrix} x_t \\ y_t p_t \end{bmatrix}\epsilon_t\|_{\Sigma_t^{-1}}}_{T_2},$$

where the last inequality holds by following Lemma 1 with $k = -1$. Then, by [17, Theorem 20.3], with probability at least $1 - \epsilon/6$, the term $T_2$ can be bounded as follows:

$$\|\sum_{s=1}^t \begin{bmatrix} x_t \\ y_t p_t \end{bmatrix}\epsilon_t\|_{\Sigma_t^{-1}} \leq \sqrt{2\log(6/\epsilon) + (d_1 + d_2)\log\Big(1 + \frac{tL^2}{(d_1 + d_2)\lambda}\Big)}. \tag{7}$$

Next, we use the following lemma to provide a high probability upper bound of term $T_1$.

**Lemma 11** (Theorem 1 in [13]). *Let $w = (w_1, \ldots, w_N)^T$ be a vector of independent, mean-zero, $R$-subgaussian random variables. Let $M \in \mathbb{R}^{N \times N}$ be a positive semi-definite matrix. For any $t \geq 0$,*

$$\mathbb{P}(w^\top M w > R^2(\text{Tr}(M) + 2\sqrt{\text{Tr}(M^2)\,t} + 2\|M\|_2 t)) \leq \exp(-t).$$

Then, let $w = (\hat\epsilon_1, \ldots, \hat\epsilon_N)^T$ and $M = A^\top(AA^\top + \lambda I)^{-1}A$, where $A \in \mathbb{R}^{(d_1+d_2)\times N}$ and $A_n = \begin{bmatrix} \hat x_n \\ \hat y_n \hat p_n \end{bmatrix}$ for every $n \in [N]$. Consider the singular value decomposition of $A$: $A = U\Sigma V^\top$. We

have

$$M = A^\top \left(AA^\top + \lambda I\right)^{-1} A = \left(V\Sigma^\top U^\top\right) \left[U\left(\Sigma\Sigma^\top + \lambda I\right)^{-1} U^\top\right] \left(U\Sigma V^\top\right)$$
$$= V\Sigma^\top \left(\Sigma\Sigma^\top + \lambda I\right)^{-1} \Sigma V^\top.$$

Therefore, the nonzero eigenvalues of $M$ are

$$\lambda_i(M) = \frac{\sigma_i^2(A)}{\sigma_i^2(A) + \lambda}, \quad 1 \le i \le r,$$

where $r \le d_1 + d_2$ denotes the rank of $A$ and $\{\sigma_i(A)\}_{i\in[r]}$ denotes the nonzero singular values of $A$. Therefore, we have $\|M\|_2 = \max_{i\in[r]} \frac{\sigma_i^2(A)}{\sigma_i^2(A)+\lambda} \le 1$, $\mathrm{Tr}(M) \le r \cdot 1 \le d_1 + d_2$ and $\mathrm{Tr}(M^2) \le d_1 + d_2$. Therefore, by Lemma 11 and choosing $t = \log(6/\epsilon)$, with probability at least $1 - \epsilon/6$, we have

$$w^\top M w \le R^2(\mathrm{Tr}(M) + 2\sqrt{\mathrm{Tr}\left(M^2\right)\log(6/\epsilon)} + 2\|M\|_2 \log(6/\epsilon))$$
$$\le R^2((d_1 + d_2) + 2\sqrt{(d_1 + d_2)\log(6/\epsilon)} + 2\log(6/\epsilon))$$
$$\le (R\sqrt{d_1 + d_2} + R\sqrt{2\log(6/\epsilon)})^2.$$

Therefore, we have proved that with probability at least $1 - \epsilon/3$,

$$\|\theta_* - \hat{\theta}_{t,N}\|_{\Sigma_{t,N}} \le \frac{\lambda\sqrt{\alpha_{\max}^2 + \beta_{\max}^2}}{\sqrt{\lambda + \lambda_{\min}(\hat{\Sigma})}} + \frac{\lambda_{\max}(\hat{\Sigma})V}{\sqrt{\lambda + \lambda_{\max}(\hat{\Sigma})}} + \sqrt{2\log(6/\epsilon) + (d_1 + d_2)\log\left(1 + \frac{tL^2}{(d_1 + d_2)\lambda}\right)}$$
$$+ R\sqrt{d_1 + d_2} + R\sqrt{2\log(6/\epsilon)} = w_{t,N}.$$

**(2) $\|\theta_* - \hat{\theta}_{t,N}\| \le \hat{w}_{t,N}$ with probability at least $1 - \epsilon/3$.** By equation (6), we have

$$\|\theta_* - \hat{\theta}_{t,N}\| \le \lambda\|\Sigma_{t,N}^{-1}\theta_*\| + \|\Sigma_{t,N}^{-1}\hat{\Sigma}(\theta_* - \theta'_*)\| + \|\Sigma_{t,N}^{-1} \sum_{n=1}^{N}\left[\begin{array}{c} \hat{x}_n \\ \hat{y}_n\hat{p}_n \end{array}\right]\hat{\epsilon}_n + \sum_{s=1}^{t}\left[\begin{array}{c} x_t \\ y_t p_t \end{array}\right]\epsilon_t\|$$

$$\le \lambda\|V_{0,N}^{-1}\theta_*\| + \|V_{0,N}^{-1}\hat{\Sigma}(\theta_* - \theta'_*)\| + \|\Sigma_{t,N}^{-1} \sum_{n=1}^{N}\left[\begin{array}{c} \hat{x}_n \\ \hat{y}_n\hat{p}_n \end{array}\right]\hat{\epsilon}_n + \sum_{s=1}^{t}\left[\begin{array}{c} x_t \\ y_t p_t \end{array}\right]\epsilon_t\|$$

$$\overset{(i)}{\le} \frac{\lambda\|\theta_*\|}{\lambda + \lambda_{\min}(\hat{\Sigma})} + \frac{\lambda_{\max}(\hat{\Sigma})V}{\lambda + \lambda_{\max}(\hat{\Sigma})} + \frac{1}{\sqrt{\lambda + \lambda_{\min}(\hat{\Sigma})}}\|\sum_{n=1}^{N}\left[\begin{array}{c} \hat{x}_n \\ \hat{y}_n\hat{p}_n \end{array}\right]\hat{\epsilon}_n + \sum_{s=1}^{t}\left[\begin{array}{c} x_t \\ y_t p_t \end{array}\right]\epsilon_t\|_{\Sigma_{t,N}^{-1}}$$

$$\le \frac{\lambda\sqrt{\alpha_{\max}^2 + \beta_{\max}^2}}{\lambda + \lambda_{\min}(\hat{\Sigma})} + V + \frac{1}{\sqrt{\lambda + \lambda_{\min}(\hat{\Sigma})}}\left(\|\sum_{n=1}^{N}\left[\begin{array}{c} \hat{x}_n \\ \hat{y}_n\hat{p}_n \end{array}\right]\hat{\epsilon}_n\|_{V_{0,N}^{-1}} + \|\sum_{s=1}^{t}\left[\begin{array}{c} x_t \\ y_t p_t \end{array}\right]\epsilon_t\|_{\Sigma_t^{-1}}\right),$$

where (i) holds by following Lemma 1 with $k = -2$. Therefore, combining the above upper bounds for the terms $T_1$ and $T_2$, with probability at least $1 - \epsilon/3$,

$$\|\theta_* - \hat{\theta}_{t,n}\| \le \frac{\lambda\sqrt{\alpha_{\max}^2 + \beta_{\max}^2}}{\lambda + \lambda_{\min}(\hat{\Sigma})} + V + \frac{\sqrt{2\log(6/\epsilon) + (d_1 + d_2)\log\left(1 + \frac{tL^2}{(d_1+d_2)\lambda}\right)}}{\sqrt{\lambda + \lambda_{\min}(\hat{\Sigma})}}$$
$$+ \frac{R\sqrt{d_1 + d_2} + R\sqrt{2\log(6/\epsilon)}}{\sqrt{\lambda + \lambda_{\min}(\hat{\Sigma})}} = \hat{w}_{t,N},$$

thereby completing the proof of Lemma 5.

### D.2.3 Proof of Lemma 6

The proof of Lemma 6 mirrors the argument in [31, Lemma 2.5] but under weaker assumptions. Whereas [31] requires all offline covariates $\{(\hat{x}_n, \hat{y}_n)\}_{n\in[N]}$ to be i.i.d. and the noise sequence

$\{\epsilon_j\}_{j \in [t]}$ to be almost surely bounded, our analysis imposes neither condition. Consequently, Lemma 6 holds in a strictly more general setting.

First, we restate the Freedman's inequality as follows:

**Lemma 12** (Freedman's inequality [3]). *Suppose $Z_1, Z_2, \ldots, Z_t$ is a martingale difference sequence with $|Z_i| \le B$ for all $i = 1, \ldots, t$. Then for any $\xi < 1/e^2$, with probability at least $1 - (\log_2 t)\xi$, we have*

$$\sum_{i=1}^t Z_i \le 4\sqrt{\sum_{i=1}^t \operatorname{Var}[Z_i \mid Z_1, \ldots, Z_{i-1}] \log(1/\xi)} + 2B \log(1/\xi).$$

Let $D_\theta(x, y, p) := \mathbb{E}[D_t \mid x_t = x, y_t = y, p_t = p; \theta_* = \theta] = \alpha^\top x + \beta^\top y \cdot p$. For any $(x, y) \in \mathcal{X} \times \mathcal{Y}, p \in [l, u]$ and $\theta, \theta' \in \Theta^\dagger$, we have

$$|D_\theta(x, y, p) - D_{\theta'}(x, y, p)| = |(\alpha - \alpha')^\top x + (\beta - \beta')^\top yp|$$
$$\le \|\theta - \theta'\| \cdot \sqrt{\|x\|_2^2 + \|y\|^2 p^2}$$
$$\le L \|\theta - \theta'\|,$$

where $L$ is defined in Appendix A. The diameter of the demand parameter set $\Theta^\dagger \in \mathbb{R}^{d_1 + d_2}$, $\operatorname{diam}(\Theta^\dagger)$, is definded as

$$\operatorname{diam}(\Theta^\dagger) = \sup_{\theta, \theta' \in \Theta^\dagger} \operatorname{diam}(\Theta^\dagger) \le 2\sqrt{\alpha_{\max}^2 + \beta_{\max}^2}.$$

By a standard covering-number result for $d$-dimensional balls [28], the minimum number of points needed to cover a $(2d)$-dimensional ball of radius $\frac{\operatorname{diam}(\Theta^\dagger)}{2}$ with balls of radius $\frac{1}{Lt}$ is at most $(1 + \operatorname{diam}(\Theta^\dagger)Lt)^{2d}$. Consequently, there exists a set $\Sigma_t$ of cardinality at most $(1 + \operatorname{diam}(\Theta^\dagger)Lt)^{2d}$ which satisfies

$$\|v\| \le \frac{\operatorname{diam}(\Theta^\dagger)}{2} + \frac{1}{Lt}, \quad \forall v \in \Sigma_t \quad \text{and} \quad \forall \theta \in \Theta^\dagger, \exists v \in \Sigma_t : \|\theta - v\| \le \frac{1}{Lt}.$$

For any $\theta \in \Theta^\dagger, (x, y) \in \mathcal{X} \times \mathcal{Y}, p \in [l, u]$, take $v$ to be the closest point to $\theta$ in $\Sigma_t$, we have

$$(D_\theta(x, y, p) - D_{\theta_*}(x, y, p))^2 = (D_\theta(x, y, p) - D_v(x, y, p) + D_v(x, y, p) - D_{\theta_*}(x, y, p))^2$$
$$\le 2(D_\theta(x, y, p) - D_v(x, y, p))^2 + 2(D_v(x, y, p) - D_{\theta_*}(x, y, p))^2$$
$$\le 2L^2 \|\theta - v\|_2^2 + 2(D_v(x, y, p) - D_{\theta_*}(x, y, p))^2$$
$$\le \frac{2}{t^2} + 2(D_v(x, y, p) - D_{\theta_*}(x, y, p))^2$$

For each fixed $v \in \Sigma_t$, let $Y_{v,t} = (D_v(x_t, y_t, p_t) - D_{\theta_*}(x_t, y_t, p_t))^2$. Since

$$\|Y_{v,t}\| \le L^2 \|v - \theta_*\|^2 \le L^2 (\operatorname{diam}(\Theta^\dagger) + \frac{2}{Lt})^2 \le (L\operatorname{diam}(\Theta^\dagger) + 2)^2 := B$$

for all $t \in [T]$, we apply Lemma 12. In particular, if $\frac{\xi_t}{|\Sigma_t|} \le \frac{1}{e^2}$, then with probability at least $1 - (\log_2 t)\frac{\xi_t}{|\Sigma_t|}$, the following holds:

$$\sum_{s=1}^t \mathbb{E}[Y_{v,s} \mid \mathcal{F}_{s-1}] - \sum_{s=1}^t Y_{v,s} \le 4\sqrt{\sum_{s=1}^t \operatorname{Var}[Y_{v,s} \mid \mathcal{F}_{s-1}] \log(|\Sigma_t|/\xi_t)} + 2B^2 \log(|\Sigma_t|/\xi_t).$$

By applying a union bound over all $v \in \Sigma_t$ in the inequality above, we conclude that, with probability at least $1 - \xi_t \log_2 t$, the following holds:

$$\sum_{s=1}^t \mathbb{E}[Y_{v,s} \mid \mathcal{F}_{s-1}] - \sum_{s=1}^t Y_{v,s} \le 4\sqrt{\sum_{s=1}^t \operatorname{Var}[Y_{v,s} \mid \mathcal{F}_{s-1}] \log(|\Sigma_t|/\xi_t)} + 2B^2 \log(|\Sigma_t|/\xi_t)$$

$$\leq 4\sqrt{\sum_{s=1}^{t} \mathbb{E}\left[Y_{v,s}^2 \mid \mathcal{F}_{s-1}\right] \log\left(\left|\Sigma_t\right|/\xi_t\right) + 2B^2 \log\left(\left|\Sigma_t\right|/\xi_t\right)}$$

$$\leq 8B\sqrt{\sum_{s=1}^{t} \mathbb{E}\left[Y_{v,s} \mid \mathcal{F}_{s-1}\right] \log\left(\left|\Sigma_t\right|/\xi_t\right) + 2B^2 \log\left(\left|\Sigma_t\right|/\xi_t\right)}, \quad \forall v \in \Sigma_t.$$

This implies

$$\left(\sqrt{\sum_{s=1}^{t} \mathbb{E}\left[Y_{v,s} \mid \mathcal{F}_{s-1}\right]} - 4B\sqrt{\log\left(\left|\Sigma_t\right|/\xi_t\right)}\right)^2 \leq 18B^2 \log\left(\left|\Sigma_t\right|/\xi_t\right) + \sum_{s=1}^{t} Y_{v,s},$$

which further implies

$$68B^2 \log\left(\left|\Sigma_t\right|/\xi_t\right) + 2\sum_{s=1}^{t} Y_{v,s} \geq \sum_{s=1}^{t} \mathbb{E}\left[Y_{v,s} \mid \mathcal{F}_{s-1}\right] = \sum_{s=1}^{t} \mathbb{E}\left[\left(D_v\left(x, y, p_s\right) - D_{\theta_*}\left(x, y, p_s\right)\right)^2 \mid \mathcal{F}_{s-1}\right],$$

Then, with probability at least $1 - \xi_t \log_2 t$, for any $\theta \in \Theta^\dagger$,

$$\sum_{s=1}^{t} \mathbb{E}\left[\left(D_\theta\left(x, y, p_s\right) - D_{\theta_*}\left(x, y, p_s\right)\right)^2 \mid \mathcal{F}_{s-1}\right] \leq 2\sum_{s=1}^{t} \mathbb{E}\left[\left(D_v\left(x, y, p_s\right) - D_{\theta_*}\left(x, y, p_s\right)\right)^2 \mid \mathcal{F}_{s-1}\right] + \frac{2}{t}$$

$$\leq 136B^2 \log\left(\left|\Sigma_t\right|/\xi_t\right) + 4\sum_{s=1}^{t} Y_{v,s} + \frac{2}{t}. \tag{8}$$

where $v$ is the closest point to $\theta$ in $\Sigma_t$. By the definition of $\mathcal{C}_t$, for all $\theta \in \mathcal{C}_t \cap \Theta^\dagger$, we have

$$\sum_{s=1}^{t} Y_{v,s} \leq 2\sum_{s=1}^{t} Y_{\theta,s} + 2\sum_{s=1}^{t} \left(D_v(x_t, y_t, p_t) - D_\theta(x_t, y_t, p_t)\right)^2$$

$$\leq 2\sum_{s=1}^{t} Y_{\theta,s} + \frac{2}{t}. \tag{9}$$

Then, by equations (8) and (9), with probability at least $1 - \xi_t \log_2 t$, for any $\theta \in \mathcal{C}_t \cap \Theta^\dagger$,

$$\mathbb{E}_x\left[\sum_{n=1}^{N} \left(\left(\alpha - \alpha_*\right)^\top \hat{x}_n + \left(\beta - \beta_*\right)^\top \hat{y}_n \hat{p}_n\right)^2 + \sum_{s=1}^{t} \left(\left(\alpha - \alpha_*\right)^\top x + \left(\beta - \beta_*\right)^\top y p_s(x, y)\right)^2 \mid \mathcal{F}_{t-1}\right]$$

$$= \sum_{n=1}^{N} \left(\left(\alpha - \alpha_*\right)^\top \hat{x}_n + \left(\beta - \beta_*\right)^\top \hat{y}_n \hat{p}_n\right)^2 + \sum_{s=1}^{t} \mathbb{E}\left[\left(D_\theta\left(x, y, p_s\right) - D_{\theta_*}\left(x, y, p_s\right)\right)^2 \mid \mathcal{F}_{s-1}\right]$$

$$\leq \sum_{n=1}^{N} \left(\left(\alpha - \alpha_*\right)^\top \hat{x}_n + \left(\beta - \beta_*\right)^\top \hat{y}_n \hat{p}_n\right)^2 + 136B^2 \log\left(\left|\Sigma_t\right|/\xi_t\right) + 8\sum_{s=1}^{t} Y_{\theta,s} + \frac{10}{t}$$

$$\leq 8\left(\sum_{n=1}^{N} \left(\left(\alpha - \alpha_*\right)^\top \hat{x}_n + \left(\beta - \beta_*\right)^\top \hat{y}_n \hat{p}_n\right)^2 + \sum_{s=1}^{t} Y_{\theta,s}\right) + 136B^2 \log\left(\left|\Sigma_t\right|/\xi_t\right) + \frac{10}{t}.$$

Then, by choosing $\eta_t = \frac{\xi}{t^3}$, we have

$$\sum_{t=1}^{\infty} \xi_t \log_2 t \leq \sum_{t=2}^{\infty} \frac{\xi}{t^2} \leq \xi.$$

Therefore, with the definition of $\mathcal{C}_t$ and Lemma 5, with probability at least $1 - \xi - \epsilon$, for any $\theta \in \mathcal{C}_t \cap \Theta^\dagger$,

$$\mathbb{E}_x\left[\sum_{n=1}^{N} \left(\left(\alpha - \alpha_*\right)^\top \hat{x}_n + \left(\beta - \beta_*\right)^\top \hat{y}_n \hat{p}_n\right)^2 + \sum_{s=1}^{t} \left(\left(\alpha - \alpha_*\right)^\top x + \left(\beta - \beta_*\right)^\top y p_s(x, y)\right)^2 \mid \mathcal{F}_{t-1}\right]$$

$$\leq 16 w_{t,n}^2 + 136 B^2 \log\left(|\Sigma_t|/\xi_t\right) + \frac{10}{t}$$

$$\leq 16 w_{t,n}^2 + 136 B^2 ((d_1 + d_2)\log(1 + \mathrm{diam}(\Theta^\dagger)Lt) + \log(t^3/\xi)) + \frac{10}{t}$$

$$\leq K(w_{t,n}^2 + (d_1 + d_2)\log T + \log(t/\xi)),$$

where $K$ is a uniform constant such that the last inequality holds. Therefore, we finish the proof of Lemma 6.

### D.2.4    Proof of Lemma 7

$$
\begin{aligned}
\delta^2 = \mathbb{E}_{x,y}[(\hat{p}(x,y) - p_{\theta_*}^*(x,y))^2] = \mathbb{E}_{x,y}[(\frac{\hat{\Sigma}_{y,x}\hat{\Sigma}_{x,x}^{-1}x}{y} - p_{\theta_*}^*(x,y))^2] \\
\leq 2\mathbb{E}_{x,y}[(\frac{\hat{\Sigma}_{y,x}\hat{\Sigma}_{x,x}^{-1}\beta x}{\beta y})^2] + 2u^2 \\
\leq \frac{2\beta_{\max}^2 x_{\max}^2 \|\hat{\Sigma}_{y,x}\hat{\Sigma}_{x,x}^{-1}\|^2}{l_\beta^2} + 2u^2 \\
\leq \frac{2\beta_{\max}^2 x_{\max}^4 y_{\max}^2 u^2}{l_\beta^2 c^2} + 2u^2,
\end{aligned}
$$

thereby completing the proof Lemma 7.

### D.2.5    Proof of Lemma 8

$$
\begin{aligned}
&\sum_{n=1}^N (\alpha^\top \hat{x}_n + \beta \hat{y}_n \hat{p}_n)^2 - \sum_{n=1}^N (\alpha \hat{x}_n + \beta \hat{A}^\top \hat{x}_n)^2 \\
=& 2\beta \alpha^\top \sum_{n=1}^N \hat{x}_n \left(\hat{y}_n \hat{p}_n - \hat{A}^\top \hat{x}_n\right) + \beta^2 \sum_{n=1}^N \left[(\hat{y}_n \hat{p}_n)^2 - \left(\hat{A}^\top \hat{x}_n\right)^2\right] \\
=& \beta^2 \sum_{n=1}^N \left[(\hat{y}_n \hat{p}_n)^2 - \left(\hat{A}^\top \hat{x}_n\right)^2\right],
\end{aligned}
\tag{10}
$$

where the last equality holds by the definition of $\hat{A}$. Let $v = (\hat{y}_1 \hat{p}_1, \ldots, \hat{y}_n \hat{p}_N)^\top \in \mathbb{R}^N$, $X = (\hat{x}_1, \ldots, \hat{x}_N)^\top \in \mathbb{R}^{N \times d_1}$ and $U$ be the subspace spanned by the columns of $X$, the term (10) can be reformulated as $\beta^2(\|v\|^2 - \|\Pi_U(v)\|^2)$. Because $\|v\| \geq \|\Pi_U(v)\|$, we have

$$\sum_{n=1}^N (\alpha^\top \hat{x}_n + \beta \hat{y}_n \hat{p}_n)^2 - \sum_{n=1}^N (\alpha^\top \hat{x}_n + \beta \hat{A}^\top \hat{x}_n)^2 \geq 0,$$

thereby completing the proof of Lemma 8.

### D.2.6    Proof of Lemma 9

Let $\Delta \alpha_t := \tilde{\alpha}_t - \alpha_*$ and $\Delta \beta_t := \tilde{\beta}_t - \beta_*$. When equation (5) holds, we have

$$\mathbb{E}_{x,y}\left[\sum_{n=1}^N \left(\Delta \alpha_t^\top \hat{x}_n + \Delta \beta_t \hat{y}_n \hat{p}_n\right)^2 + \sum_{s=1}^{t-1} \left(\Delta \alpha_t^\top x + \Delta \beta_t y p_s(x,y)\right)^2 \mid \mathcal{F}_{t-1}, x_t\right] \leq K \eta_{t-1}^2, \quad \forall t \in [T].$$

(11)

We now prove Lemma 9 by considering the following four cases.

**Case 1:** $\Delta \beta_t = 0$.  By equation (11), we have

$$K \eta_{t-1}^2 \geq \Delta \alpha_t^\top \hat{\Sigma}_{x,x} \Delta \alpha_t + (t-1)\Delta \alpha_t^\top \mathbb{E}[xx^\top]\Delta \alpha_t$$

$$\geq \left(\lambda_{\min}(\hat{\Sigma}_{x,x}) + (t-1)\lambda_{\min}(\mathbb{E}[xx^\top])\right)\|\Delta\alpha_t\|^2$$

$$\geq \left((cN \vee \lambda_{\min}(\hat{\Sigma})) + (t-1)\lambda_{\min}(\mathbb{E}[xx^\top])\right)\|\Delta\alpha_t\|^2,$$

where the last inequality holds by Assumption 2 and the fact that $\hat{\Sigma}_{x,x}$ is a principal submatrix of $\hat{\Sigma}$. Then, we have

$$\|\tilde{\theta}_t - \theta_*\|^2 = \|\Delta\alpha_t\|^2 \leq \frac{K\eta_{t-1}^2}{(cN \vee \lambda_{\min}(\hat{\Sigma})) + (t-1)\lambda_{\min}(\mathbb{E}[xx^\top])} \tag{12}$$

$$\leq \frac{K\eta_{t-1}^2}{\lambda_{\min}(\hat{\Sigma}) + (N \wedge (t-1))\lambda_{\min}(\mathbb{E}[xx^\top])}$$

$$\leq \frac{C_1 K\eta_{t-1}^2}{\lambda_{\min}(\hat{\Sigma}) + (N \wedge (t-1))\delta^2}, \tag{13}$$

where $C_1 = \max\{1, \frac{\frac{2\beta_{\max}^2 x_{\max}^4 y_{\max}^2 u^2}{l_\beta^2 c^2} + 2u^2}{\lambda_{\min}(\mathbb{E}[xx^\top])}\}$ and the last inequality holds by following Lemma 7. Then, by inequality (12), we have

$$\mathbb{E}_{x,y}[(\hat{p}(x,y) - p_t(x,y))^2] \geq \frac{1}{2}\mathbb{E}_{x,y}[(\hat{p}(x,y) - p^*(x,y))^2] - \mathbb{E}_{x,y}[(p^*(x,y) - p_t(x,y))^2]$$

$$\overset{(i)}{\geq} \frac{1}{2}\delta^2 - \frac{(y_{\max}^2 u_\alpha^2 + x_{\max}^2 u_\beta^2)\|\tilde{\theta}_t - \theta_*\|^2}{4l_\beta^4}$$

$$\geq \frac{1}{2}\delta^2 - \frac{(y_{\max}^2 u_\alpha^2 + x_{\max}^2 u_\beta^2)C_1 K\eta_{t-1}^2}{4l_\beta^4 cN} \overset{(ii)}{\geq} \frac{\delta^2}{4},$$

where (i) holds by Lemma 4 and (ii) holds since the assumption that $\delta^2 \geq \frac{(y_{\max}^2 u_\alpha^2 + x_{\max}^2 u_\beta^2)C_1 K\eta_T^2}{l_\beta^4 cN}$.

**Case 2:** $\Delta\beta_t \neq 0, \gamma_t \geq \sqrt{\frac{4x_{\max}^2 y_{\max}^2 u^2}{c^2}} \vee \sqrt{\frac{4y_{\max}^2 u^2}{\lambda_{\min}(\mathbb{E}[xx^\top])}}$. By equation (11) and Lemma 8, we have

$$K\eta_{t-1}^2 \geq \sum_{n=1}^{N}\left(\Delta\alpha_t^\top \hat{x}_n + \Delta\beta_t \hat{A}\hat{x}_n\right)^2 + \frac{1}{2}\mathbb{E}_{x,y}\left[\sum_{s=1}^{t-1}\left(\Delta\alpha_t^\top x\right)^2\right] - \mathbb{E}_{x,y}\left[\sum_{s=1}^{t-1}\left(\Delta\beta_t^\top yp_s(x,y)\right)^2\right]$$

$$\geq (cN \vee \lambda_{\min}(\hat{\Sigma}))\|\Delta\alpha_t + \hat{A}\Delta\beta_t\|^2 + \frac{(t-1)\lambda_{\min}(\mathbb{E}[xx^\top])\|\Delta\alpha_t\|^2}{2} - (t-1)y_{\max}^2 u^2\|\Delta\beta_t\|^2$$

$$\geq (cN \vee \lambda_{\min}(\hat{\Sigma}))\left(\frac{\|\Delta\alpha_t\|^2}{2} - \frac{x_{\max}^2 y_{\max}^2 u^2}{c^2}\|\Delta\beta_t\|^2\right) + \frac{(t-1)\lambda_{\min}(\mathbb{E}[xx^\top])\|\Delta\alpha_t\|^2}{2}$$

$$- (t-1)y_{\max}^2 u^2\|\Delta\beta_t\|^2$$

$$\geq \frac{(cN \vee \lambda_{\min}(\hat{\Sigma}))\|\Delta\alpha_t\|^2}{4} + \frac{(t-1)\lambda_{\min}(\mathbb{E}[xx^\top])\|\Delta\alpha_t\|^2}{4},$$

where the last inequality holds because $\gamma_t \geq \sqrt{\frac{4x_{\max}^2 y_{\max}^2 u^2}{c^2}} \vee \sqrt{\frac{4y_{\max}^2 u^2}{\lambda_{\min}(\mathbb{E}[xx^\top])}}$. Then, we have

$$\|\tilde{\theta}_t - \theta_*\|^2 = (1 + \frac{1}{\gamma_t^2})\|\Delta\alpha_t\|^2 \leq \frac{4K\eta_{t-1}^2(1 + \frac{1}{\gamma_t^2})}{(cN \vee \lambda_{\min}(\hat{\Sigma})) + (t-1)\lambda_{\min}\mathbb{E}[xx^\top]}$$

$$\leq \frac{4K\eta_{t-1}^2(1 + (\frac{c^2}{4x_{\max}^2 y_{\max}^2 u^2} \wedge \frac{\lambda_{\min}(\mathbb{E}[xx^\top])}{4y_{\max}^2 u^2}))}{(cN \vee \lambda_{\min}(\hat{\Sigma})) + (t-1)\lambda_{\min}\mathbb{E}[xx^\top]} \tag{14}$$

$$\leq \frac{4K\eta_{t-1}^2(1 + (\frac{c^2}{4x_{\max}^2 y_{\max}^2 u^2} \wedge \frac{\lambda_{\min}(\mathbb{E}[xx^\top])}{4y_{\max}^2 u^2}))}{\lambda_{\min}(\hat{\Sigma}) + (N \wedge (t-1))\lambda_{\min}(\mathbb{E}[xx^\top])}$$

$$\leq \frac{C_2 K\eta_{t-1}^2}{\lambda_{\min}(\hat{\Sigma}) + (N \wedge (t-1))\delta^2}, \tag{15}$$

where $C_2 = 4(1 + (\frac{c^2}{4x_{\max}^2 y_{\max}^2 u^2} \wedge \frac{\lambda_{\min}(\mathbb{E}[xx^\top])}{4y_{\max}^2 u^2}))C_1$. Then, by inequality (14), and proceeding as in Case 1, we have

$$\mathbb{E}_{x,y}[(\hat{p}(x,y) - p_t(x,y))^2] \geq \frac{1}{2}\delta^2 - \frac{(y_{\max}^2 u_\alpha^2 + x_{\max}^2 u_\beta^2)C_2 K\eta_{t-1}^2}{4l_\beta^4 cN} \geq \frac{\delta^2}{4},$$

where the last inequality holds since the assumption that $\delta^2 \geq \frac{(y_{\max}^2 u_\alpha^2 + x_{\max}^2 u_\beta^2)C_2 K\eta_T^2}{l_\beta^4 cN}$.

**Case 3:** $\Delta\beta_t \neq 0, \gamma_t \leq \sqrt{\frac{4x_{\max}^2 y_{\max}^2 u^2}{c^2}} \vee \sqrt{\frac{4y_{\max}^2 u^2}{\lambda_{\min}(\mathbb{E}[xx^\top])}}$ **and** $\|\Delta\alpha_t + \Delta\beta_t\hat{A}\|^2 \geq \frac{l_\beta l y_{\min}^2}{8u_\beta u((1+\frac{l_\beta l}{2u_\beta u}))x_{\max}^2} \cdot \delta^2 (\Delta\beta_t)^2$. By equation (11) and Lemma 8, we have

$$K\eta_{t-1}^2 \geq \sum_{n=1}^N \left(\Delta\alpha_t^\top \hat{x}_n + \Delta\beta_t\hat{A}\hat{x}_n\right)^2 \geq cN\|\Delta\alpha_t + \hat{A}\Delta\beta_t\|^2 \geq \frac{cl_\beta l y_{\min}^2}{8u_\beta u((1+\frac{l_\beta l}{2u_\beta u}))x_{\max}^2} \cdot N\delta^2(\Delta\beta_t)^2,$$

which implies

$$\|\tilde{\theta}_t - \theta_*\|^2 = (1 + \gamma_t^2)(\Delta\beta_t)^2 \leq \frac{C_3 K\eta_{t-1}^2}{N\delta^2} \tag{16}$$

$$\overset{(i)}{\leq} \frac{2C_3 K\eta_{t-1}^2}{\lambda_{\min}(\hat{\Sigma}) + N\delta^2} \leq \frac{2C_3 K\eta_{t-1}^2}{\lambda_{\min}(\hat{\Sigma}) + (N \wedge (t-1))\delta^2},$$

where $C_3 = \frac{8u_\beta u((1+\frac{l_\beta l}{2u_\beta u}))x_{\max}^2}{cl_\beta l y_{\min}^2}(1 + (\frac{4x_{\max}^2 y_{\max}^2 u^2}{c^2} \vee \frac{4y_{\max}^2 u^2}{\lambda_{\min}(\mathbb{E}[xx^\top])}))$ and (i) holds since the assumption that $N\delta^2 \geq \lambda_{\min}(\hat{\Sigma})$. Then, by inequality (16), and proceeding as in Case 1, we have

$$\mathbb{E}_{x,y}[(\hat{p}(x,y) - p_t(x,y))^2] \geq \frac{\delta^2}{2} - \frac{(y_{\max}^2 u_\alpha^2 + x_{\max}^2 u_\beta^2)C_3 K\eta_{t-1}^2}{4l_\beta^4 N\delta^2} \geq \frac{\delta^2}{4},$$

where the last inequality holds since the assumption that $\delta^2 \geq \sqrt{\frac{(y_{\max}^2 u_\alpha^2 + x_{\max}^2 u_\beta^2)C_3 K\eta_{t-1}^2}{l_\beta^4 N}}$.

**Case 4:** $\Delta\beta_t \neq 0, \gamma_t \leq \sqrt{\frac{4x_{\max}^2 y_{\max}^2 u^2}{c^2}} \vee \sqrt{\frac{4y_{\max}^2 u^2}{\lambda_{\min}(\mathbb{E}[xx^\top])}}$ **and** $\|\Delta\alpha_t + \Delta\beta_t\hat{A}\|^2 \leq \frac{l_\beta l y_{\min}^2}{8u_\beta u((1+\frac{l_\beta l}{2u_\beta u}))x_{\max}^2} \cdot \delta^2 (\Delta\beta_t)^2$. By optimizing the left hand of inequality (11) with respect to $\Delta\alpha_t$, we have

$$K\eta_{t-1}^2 \geq \min_{\alpha \in \mathbb{R}^{d_1}} \mathbb{E}_{x,y} \left[\sum_{n=1}^N \left(\alpha^\top \hat{x}_n + \Delta\beta_t\hat{y}_n\hat{p}_n\right)^2 + \sum_{s=1}^{t-1} \left(\alpha^\top x + \Delta\beta_t y p_s(x,y)\right)^2\right]$$

$$\overset{(i)}{\geq} \left(\min_{\alpha \in \mathbb{R}^{d_1}} \sum_{n=1}^N \left(\alpha^\top \hat{x}_i + \Delta\beta_t\hat{y}_i\hat{p}_i\right)^2\right)$$

$$\quad + \frac{\lambda_{\min}(\mathbb{E}[xx^T]^{-1/2}\hat{\Sigma}_{x,x}\mathbb{E}[xx^T]^{-1/2})}{(t-1) + \lambda_{\min}(\mathbb{E}[xx^T]^{-1/2}\hat{\Sigma}_{x,x}\mathbb{E}[xx^T]^{-1/2})}\mathbb{E}_{x,y}\left[\sum_{s=1}^{t-1}\left(\tilde{\alpha}^\top x + \Delta\beta_t y p_s(x,y)\right)^2\right]$$

$$= (\hat{\Sigma}_{y,y} - \hat{\Sigma}_{y,x}\hat{\Sigma}_{x,x}^{-1}\hat{\Sigma}_{x,y})(\Delta\beta_t)^2$$

$$\quad + \frac{\lambda_{\min}(\mathbb{E}[xx^T]^{-1/2}\hat{\Sigma}_{x,x}\mathbb{E}[xx^T]^{-1/2})}{(t-1) + \lambda_{\min}(\mathbb{E}[xx^T]^{-1/2}\hat{\Sigma}_{x,x}\mathbb{E}[xx^T]^{-1/2})}\mathbb{E}_{x,y}\left[\sum_{s=1}^{t-1}\left(\tilde{\alpha}^\top x + \Delta\beta_t y p_s(x,y)\right)^2\right]$$

$$\geq (\hat{\Sigma}_{y,y} - \hat{\Sigma}_{y,x}\hat{\Sigma}_{x,x}^{-1}\hat{\Sigma}_{x,y})(\Delta\beta_t)^2 + \frac{\frac{\lambda_{\min}(\hat{\Sigma})}{\lambda_{\max}(\mathbb{E}[xx^T])}(\Delta\beta_t)^2}{(t-1) + \frac{\lambda_{\min}(\hat{\Sigma})}{\lambda_{\max}(\mathbb{E}[xx^T])}}\mathbb{E}_{x,y}\left[\sum_{s=1}^{t-1}\left(-\hat{A}^\top x + y p_s(x,y)\right)^2\right]$$

$$\geq (\hat{\Sigma}_{y,y} - \hat{\Sigma}_{y,x}\hat{\Sigma}_{x,x}^{-1}\hat{\Sigma}_{x,y})(\Delta\beta_t)^2 + \frac{cN(\Delta\beta_t)^2}{(t-1)x_{\max}^2 + cN}\mathbb{E}_{x,y}\left[\sum_{s=1}^{t-1}\left(-\hat{A}^\top x + y p_s(x,y)\right)^2\right]$$

$$\geq (\hat{\Sigma}_{y,y} - \hat{\Sigma}_{y,x}\hat{\Sigma}_{x,x}^{-1}\hat{\Sigma}_{x,y})(\Delta\beta_t)^2 + \frac{cN(\Delta\beta_t)^2 y_{\min}^2}{(t-1)x_{\max}^2 + cN}\mathbb{E}_{x,y}\left[\sum_{s=1}^{t-1}(\hat{p}(x,y) - p_s(x,y))^2\right]$$

$$\overset{(ii)}{\geq}(\hat{\Sigma}_{y,y} - \hat{\Sigma}_{y,x}\hat{\Sigma}_{x,x}^{-1}\hat{\Sigma}_{x,y})(\Delta\beta_t)^2 + \frac{cN(t-1)\delta^2 y_{\min}^2(\Delta\beta_t)^2}{(t-1)x_{\max}^2 + cN}(\frac{1}{4}\wedge\frac{l_\beta l}{8u_\beta u})$$

$$\overset{(iii)}{\geq}\left(\lambda_{\min}(\hat{\Sigma}) + \frac{cN(t-1)\delta^2 y_{\min}^2}{(t-1)x_{\max}^2 + cN}(\frac{1}{4}\wedge\frac{l_\beta l}{8u_\beta u})\right)(\Delta\beta_t)^2,$$

where $\tilde{\alpha} := \arg\min_{\alpha\in\mathbb{R}^{d_1}}\sum_{i=1}^n\left(\alpha^\top\hat{x}_i + \Delta\beta_t\hat{y}_i\hat{p}_i\right)^2 = -\hat{A}\Delta\beta_t$. (i) follows from Lemma 2; (ii) is obtained by induction and (iii) holds because $\hat{\Sigma}_{y,y} - \hat{\Sigma}_{y,x}\hat{\Sigma}_{x,x}^{-1}\hat{\Sigma}_{x,y}$ is the the Schur complement of $\hat{\Sigma}_{x,x}$ in $\hat{\Sigma}$. Because $\frac{xy}{x+y}\geq\frac{x\wedge y}{2}$, there exists a positive constant $C_4$ such that

$$\|\tilde{\theta}_t - \theta_*\|^2 = (1 + \gamma_t^2)(\Delta\beta_t)^2 \leq \frac{C_4 K\eta_{t-1}^2}{\lambda_{\min}(\hat{\Sigma}) + (N\wedge(t-1))\delta^2},$$

To provide an upper bound on $\mathbb{E}_{x,y}[(\hat{p}(x,y) - p_t(x,y))^2]$, we have

$$(\Delta\beta_t)^2\mathbb{E}_{x,y}[(-\hat{A}^\top x + yp_t(x,y))^2]$$

$$=\mathbb{E}_{x,y}[(\hat{A}^\top x\Delta\beta_t + \Delta\beta_t yp_t(x,y))^2]$$

$$\geq\frac{1}{2}\mathbb{E}_{x,y}[(-\Delta\alpha_t^\top x + \Delta\beta_t yp_t(x,y))^2] - \mathbb{E}_{x,y}[(\Delta\alpha_t^\top x + \Delta\beta_t\hat{A}^\top x)^2]$$

$$\overset{(i)}{\geq}\frac{l_\beta l}{2u_\beta u}\mathbb{E}_{x,y}[(-\Delta\alpha_t^\top x + \Delta\beta_t yp^*(x,y))^2] - \mathbb{E}_{x,y}[(\Delta\alpha_t^\top x + \Delta\beta_t\hat{A}^\top x)^2]$$

$$\geq\frac{l_\beta l(\Delta\beta_t)^2}{4u_\beta u}\mathbb{E}_{x,y}[(-\hat{A}^\top x + yp^*(x,y))^2] - (1 + \frac{l_\beta l}{2u_\beta u})\mathbb{E}_{x,y}[(\Delta\alpha_t^\top x + \Delta\beta_t\hat{A}^\top x)^2]$$

$$\geq\frac{l_\beta l(\Delta\beta_t)^2 y_{\min}^2\delta^2}{4u_\beta u} - (1 + \frac{l_\beta l}{2u_\beta u})\mathbb{E}_{x,y}[(\Delta\alpha_t^\top x + \Delta\beta_t\hat{A}^\top x)^2],$$

where (i) holds by following the similar argument in [7, EC.12] and the above inequality implies

$$\mathbb{E}_{x,y}[(\hat{A}^\top x + yp_t(x,y))^2] \geq \frac{l_\beta ly_{\min}^2\delta^2}{4u_\beta u} - (1 + \frac{l_\beta l}{2u_\beta u})\frac{\mathbb{E}_{x,y}[(\Delta\alpha_t^\top x + \Delta\beta_t\hat{A}^\top x)^2]}{(\Delta\beta_t)^2}$$

$$\geq \frac{l_\beta ly_{\min}^2\delta^2}{4u_\beta u} - (1 + \frac{l_\beta l}{2u_\beta u})\frac{x_{\max}^2\|\Delta\alpha_t + \Delta\beta_t\hat{A}\|^2}{(\Delta\beta_t)^2} \geq \frac{l_\beta ly_{\min}^2\delta^2}{8u_\beta u},$$

where the last ineqaulity holds because $\|\Delta\alpha_t + \Delta\beta_t\hat{A}\|^2 \leq \frac{l_\beta ly_{\min}^2}{8u_\beta u((1+\frac{l_\beta l}{2u_\beta u}))x_{\max}^2}\cdot\delta^2(\Delta\beta_t)^2$. Therefore, we have

$$\mathbb{E}_{x,y}[(\hat{p}(x,y) - p_t(x,y))^2] \geq \frac{1}{y_{\max}^2}\mathbb{E}_{x,y}[(\hat{A}^\top x + yp_t(x,y))^2] \geq \frac{l_\beta ly_{\min}^2\delta^2}{8y_{\max}^2 u_\beta u},$$

thereby finishing the proof of Lemma 9.

### D.2.7 Proof of Lemma 10

By the definition of $\hat{p}(\cdot,\cdot)$, we have

$$\sum_{n=1}^N(\hat{p}(\hat{x}_n,\hat{y}_n) - p_{\theta_*}^*(\hat{x}_n,\hat{y}_n))^2 = \sum_{n=1}^N(\frac{\hat{A}^\top\hat{x}_n}{\hat{y}_n} - \frac{\alpha_*^\top\hat{x}_n}{2\beta_*\hat{y}_n})^2$$

$$\leq \frac{1}{y_{\min}^2}(\hat{A} - \frac{\alpha_*}{2\beta_*})^\top\left(\sum_{n=1}^N\hat{x}_n\hat{x}_n^\top\right)(\hat{A} - \frac{\alpha_*}{2\beta_*})$$

$$\leq \frac{Nx_{\max}^2}{y_{\min}^2}\|\hat{A} - \frac{\alpha}{2\beta}\|^2$$

$$\leq \frac{Nx_{\max}^2}{y_{\min}^2 \lambda_{\min}(\mathbb{E}[xx^T])}(\hat{A} - \frac{\alpha_*}{2\beta_*})^\top \mathbb{E}[xx^T](\hat{A} - \frac{\alpha}{2\beta})$$

$$\leq \frac{Nx_{\max}^2}{y_{\min}^2 \lambda_{\min}(\mathbb{E}[xx^T])}\mathbb{E}[(\hat{A}^\top x - \frac{\alpha_*^\top x}{2\beta_*})^2]$$

$$\leq \frac{Nx_{\max}^2 y_{\max}^2}{y_{\min}^2 \lambda_{\min}(\mathbb{E}[xx^T])}\mathbb{E}_{x,y}[(\hat{p}(x,y) - p_{\theta_*}^*(x,y))^2].$$

Meanwhile, we have

$$\sum_{n=1}^N (\hat{p}(\hat{x}_n, \hat{y}_n) - \hat{p}_n)^2 = \sum_{n=1}^N (\frac{\hat{A}^\top \hat{x}_n}{\hat{y}_n} - \hat{p}_n)^2 \leq \frac{1}{y_{\min}^2}\sum_{n=1}^N(-\hat{A}^\top \hat{x}_n + \hat{y}_n \hat{p}_n)^2$$

$$= \frac{1}{y_{\min}^2}\min_{v \in \mathbb{R}^{d_1}}\sum_{n=1}^N (u^\top \hat{x}_n + \hat{y}_n \hat{p}_n)^2.$$

Recall that $\hat{\Sigma} = \sum_{i=1}^N \begin{bmatrix} \hat{x}_n \hat{x}_n^\top & \hat{x}_n \hat{p}_n \hat{y}_n^\top \\ \hat{y}_n \hat{p}_n \hat{x}_n^\top & \hat{p}_n^2 \hat{y}_n^2 \end{bmatrix} = \begin{bmatrix} \hat{\Sigma}_{x,x} & \hat{\Sigma}_{x,y} \\ \hat{\Sigma}_{y,x} & \hat{\Sigma}_{y,y} \end{bmatrix}$. On one hand, by Assumption 2, we have

$$\min_{v \in \mathbb{R}^{d_1}}\sum_{n=1}^N (u^\top \hat{x}_n + \hat{y}_n \hat{p}_n)^2 \leq 2N(x_{\max}^2 + u^2 y_{\max}^2) \leq \frac{2(x_{\max}^2 + u^2 y_{\max}^2)}{c} \cdot \lambda_{\min}(\Sigma_{x,x}).$$

On the other hand, we have

$$\min_{v \in \mathbb{R}^{d_1}}\sum_{n=1}^N (u^\top \hat{x}_n + \hat{y}_n \hat{p}_n)^2 = \hat{\Sigma}_{y,y} - \hat{\Sigma}_{y,x}\hat{\Sigma}_{x,x}^{-1}\hat{\Sigma}_{x,y} = \lambda_{\min}(\hat{\Sigma}_{y,y} - \hat{\Sigma}_{y,x}\hat{\Sigma}_{x,x}^{-1}\hat{\Sigma}_{x,y}).$$

Therefore, we have

$$\sum_{n=1}^N (\hat{p}(\hat{x}_n, \hat{y}_n) - \hat{p}_n)^2 \leq \frac{\max\{1, \frac{2(x_{\max}^2 + u^2 y_{\max}^2)}{c}\}}{y_{\min}^2} \cdot \min\{\lambda_{\min}(\Sigma_{x,x}), \lambda_{\min}(\hat{\Sigma}_{y,y} - \hat{\Sigma}_{y,x}\hat{\Sigma}_{x,x}^{-1}\hat{\Sigma}_{x,y})\}$$

$$= \frac{\max\{c, 2(x_{\max}^2 + u^2 y_{\max}^2)\}}{cy_{\min}^2} \cdot \lambda_{\min}(\hat{\Sigma}),$$

where the last equality holds because $\hat{\Sigma}_{y,y} - \hat{\Sigma}_{y,x}\hat{\Sigma}_{x,x}^{-1}\hat{\Sigma}_{x,y}$ is the Schur complement of $\hat{\Sigma}_{x,x}$ in the matrix $\hat{\Sigma}$. Therefore, we complete the proof of Lemma 10.

# E    Proof of Theorem 1

We first present the Lemma 13 that establishes an upper bound for Algorithm 1 without checking the condition

$$\min_{\theta \in \mathcal{C}_0}\sum_{n=1}^N (\hat{p}(\hat{x}_n, \hat{y}_n) - p_\theta^*(\hat{x}_n, \hat{y}_n))^2 \leq \frac{Nx_{\max}^2 y_{\max}^2}{y_{\min}^2 \lambda_{\min}(\mathbb{E}[xx^T])}\max\{V^2, \frac{1}{\lambda_{\min}(\hat{\Sigma})}\} \quad \text{and} \quad \max\{V^2, \frac{1}{\lambda_{\min}(\hat{\Sigma})}\} \leq T^{-1/2} \tag{17}$$

and defer the proof of Lemma 13 to the end of this section.

**Lemma 13.** *Let $\pi$ be the Algorithm 1 without checking the condition* (17)*, for any $(\theta'_*, \theta_*) \in \{(\theta', \theta) \in \Theta^\dagger \times \Theta^\dagger : \|\theta' - \theta\| \leq V\}$, we have*

$$R_{\theta'_*, \theta_*}^\pi(T) \in \mathcal{O}\left(d_1\sqrt{T}\log T \wedge (V^2 T + \frac{d_1 T \log T}{\lambda_{\min}(\hat{\Sigma})}) \wedge \frac{\lambda_{\max}(\hat{\Sigma})V^2 T \log T + d_1 T \log^2 T}{\lambda_{\min}(\hat{\Sigma}) + (N \wedge T)\delta^2}\right).$$

By Lemma 5, if $\theta_* \in \mathcal{C}_0$, for any $\theta \in \mathcal{C}_0$, we have

$$\|\theta - \theta_*\| \leq 2w_{0,N} = \frac{\lambda\sqrt{\alpha_{\max}^2 + \beta_{\max}^2}}{\lambda + \lambda_{\min}(\hat{\Sigma})} + V + \frac{\sqrt{2\log(6T^2)}}{\sqrt{\lambda + \lambda_{\min}(\hat{\Sigma})}} + \frac{R\sqrt{d_1 + 1} + R\sqrt{2\log(6T^2)}}{\sqrt{\lambda + \lambda_{\min}(\hat{\Sigma})}}.$$

which implies that if $\lambda_{\min}(\hat{\Sigma}) \geq \sqrt{T}$, there exists constant $L_0 > 0$ such that

$$\|\theta - \theta_*\|^2 \leq L_0(V^2 + \frac{d_1 + \log T}{\lambda_{\min}(\hat{\Sigma})}), \quad \forall \theta \in \mathcal{C}_0. \tag{18}$$

Let A be the event $\{\min_{\theta \in \mathcal{C}_0} \sum_{n=1}^{N}(\hat{p}(\hat{x}_i, \hat{y}_i) - p_\theta^*(\hat{x}_i, \hat{y}_i))^2 \leq \frac{N x_{\max}^2 y_{\max}^2}{y_{\min}^2 \lambda_{\min}(\mathbb{E}[xx^T])} \max\{V^2, \frac{1}{\lambda_{\min}(\hat{\Sigma})}\}\}$.
We now prove Theorem 1 by consider the following four cases.

**Case 1:** $\max\{V^2, \frac{1}{\lambda_{\min}(\hat{\Sigma})}\} \geq \sqrt{T}$**.** In this case, the condition (17) does not hold. Then, by Lemma 13, the regret is bounded by

$$\mathcal{O}\left(d\sqrt{T}\log(T) \wedge (V^2 T + \frac{dT\log(T)}{\lambda_{\min}(\hat{\Sigma})}) \wedge \frac{\lambda_{\max}(\hat{\Sigma})V^2 T \log T + d^2 T \log^2(T)}{\lambda_{\min}(\hat{\Sigma}) + (N \wedge T)\delta^2}\right).$$

**Case 2:** $\delta^2 \leq \max\{V^2, \frac{1}{\lambda_{\min}(\hat{\Sigma})}\} \leq \sqrt{T}$**.** In this case, if $\theta_* \in \mathcal{C}_0$, we have

$$\min_{\theta \in \mathcal{C}_0} \sum_{n=1}^{N}(\hat{p}(\hat{x}_n, \hat{y}_n) - p_\theta^*(\hat{x}_n, \hat{y}_n))^2 \leq \sum_{n=1}^{N}(\hat{p}(\hat{x}_n, \hat{y}_n) - p_{\theta_*}^*(\hat{x}_n, \hat{y}_n))^2$$

$$\leq \frac{N x_{\max}^2 y_{\max}^2}{y_{\min}^2 \lambda_{\min}(\mathbb{E}[xx^T])} \mathbb{E}_{x,y}[(\hat{p}(x,y) - p^*(x,y))^2]$$

$$\leq \frac{N x_{\max}^2 y_{\max}^2}{y_{\min}^2 \lambda_{\min}(\mathbb{E}[xx^T])} \max\{V^2, \frac{1}{\lambda_{\min}(\hat{\Sigma})}\},$$

where the second inequality holds by following Lemma 10. Therefore, if $\theta_* \in \mathcal{C}_0$, event $A$ happens. Then, we have

$$R_{\theta'_*, \theta_*}^\pi(T) = \mathbb{P}(A) \cdot \sum_{t=1}^{T} \mathbb{E}\left[r_{\theta_*}^*(x_t, y_t) - r_{\theta_*}(x_t, y_t, p_t) \mid A\right] + \mathbb{P}(A^\complement) \cdot \sum_{t=1}^{T} \mathbb{E}\left[r_{\theta_*}^*(x_t, y_t) - r_{\theta_*}(x_t, y_t, p_t) \mid A^\complement\right]$$

$$\lesssim T\delta^2 + 1.$$

**Case 3:** $\max\{V^2, \frac{1}{\lambda_{\min}(\hat{\Sigma})}\} \leq \sqrt{T}$ **and** $\delta^2 \geq \frac{KN x_{\max}^2 y_{\max}^2}{y_{\min}^2 \lambda_{\min}(\mathbb{E}[xx^T])}(V^2 + \frac{d_1 + \log T}{\lambda_{\min}(\hat{\Sigma})})$**.** The constant $K$ will be specified later. In this case, if $\theta_* \in \mathcal{C}_0$, there exists $\tilde{\theta} \in \mathcal{C}_0$ such that

$$\min_{\theta \in \mathcal{C}_0} \sum_{n=1}^{N}(\hat{p}(\hat{x}_n, \hat{y}_n) - p_\theta^*(\hat{x}_n, \hat{y}_n))^2$$

$$\geq \frac{1}{2}\sum_{n=1}^{N}(\hat{p}(\hat{x}_n, \hat{y}_n) - p_{\theta_*}^*(\hat{x}_n, \hat{y}_n))^2 - \sum_{n=1}^{N}(p_{\tilde{\theta}}^*(\hat{x}_n, \hat{y}_n) - p_{\theta_*}^*(\hat{x}_n, \hat{y}_n))^2$$

$$\overset{(i)}{\geq} \frac{KN x_{\max}^2 y_{\max}^2}{2y_{\min}^2 \lambda_{\min}(\mathbb{E}[xx^T])}(V^2 + \frac{d_1 + \log T}{\lambda_{\min}(\hat{\Sigma})}) - \frac{N(y_{\max}^2 u_\alpha^2 + x_{\max}^2 u_\beta^2)\|\tilde{\theta} - \theta_*\|^2}{4l_\beta^4}$$

$$\overset{(ii)}{\geq} \frac{KN x_{\max}^2 y_{\max}^2}{2y_{\min}^2 \lambda_{\min}(\mathbb{E}[xx^T])}(V^2 + \frac{d_1 + \log T}{\lambda_{\min}(\hat{\Sigma})}) - \frac{NL_0(y_{\max}^2 u_\alpha^2 + x_{\max}^2 u_\beta^2)}{4l_\beta^4}(V^2 + \frac{d_1 + \log T}{\lambda_{\min}(\hat{\Sigma})})$$

$$\geq \frac{KN x_{\max}^2 y_{\max}^2}{4y_{\min}^2 \lambda_{\min}(\mathbb{E}[xx^T])}(V^2 + \frac{d_1 + \log T}{\lambda_{\min}(\hat{\Sigma})})$$

$$> \frac{N x_{\max}^2 y_{\max}^2}{4y_{\min}^2 \lambda_{\min}(\mathbb{E}[xx^T])} \max\{V^2, \frac{1}{\lambda_{\min}(\hat{\Sigma})}\}.$$

where (i) holds by following Lemma 4, (ii) holds because of ineqaulity (18) and the last two inequalities holds because we choose $K = \max\{5, \frac{L_0(y_{\max}^2 u_\alpha^2 + x_{\max}^2 u_\beta^2)y_{\min}^2 \lambda_{\min}(\mathbb{E}[xx^T])}{4l_\beta^4 x_{\max}^2 y_{\max}^2}\}$. The above inequality implies event $A$ does not happen. Therefore, we have

$$R_{\theta'_*, \theta_*}^\pi(T) = \mathbb{P}(A) \cdot \sum_{t=1}^{T} \mathbb{E}\left[r_{\theta_*}^*(x_t, y_t) - r_{\theta_*}(x_t, y_t, p_t) \mid A\right] + \mathbb{P}(A^\complement) \cdot \sum_{t=1}^{T} \mathbb{E}\left[r_{\theta_*}^*(x_t, y_t) - r_{\theta_*}(x_t, y_t, p_t) \mid A^\complement\right]$$

$$\lesssim \epsilon T + \mathcal{O}\left(d_1\sqrt{T}\log T \wedge (V^2 T + \frac{d_1 T\log T}{\lambda_{\min}(\hat\Sigma)}) \wedge \frac{\lambda_{\max}(\hat\Sigma)V^2 T\log T + d_1 T\log^2 T}{\lambda_{\min}(\hat\Sigma) + (N\wedge T)\delta^2}\right)$$

$$\in \mathcal{O}\left(d_1\sqrt{T}\log T \wedge (V^2 T + \frac{d_1 T\log T}{\lambda_{\min}(\hat\Sigma)}) \wedge \frac{\lambda_{\max}(\hat\Sigma)V^2 T\log T + d_1 T\log^2 T}{\lambda_{\min}(\hat\Sigma) + (N\wedge T)\delta^2}\right).$$

**Case 4:** $\max\{V^2, \frac{1}{\lambda_{\min}(\hat\Sigma)}\} \le \sqrt{T}$ **and** $\max\{V^2, \frac{1}{\lambda_{\min}(\hat\Sigma)}\} \le \delta^2 \le \frac{KNx_{\max}^2 y_{\max}^2}{y_{\min}^2 \lambda_{\min}(\mathbb{E}[xx^T])}(V^2 + \frac{d_1+\log T}{\lambda_{\min}(\hat\Sigma)})$. In this case, we have

$$\delta^2 T \lesssim V^2 T + \frac{d_1 T + T\log T}{\lambda_{\min}(\hat\Sigma)}$$

$$\lesssim d_1\sqrt{T}\log T \wedge (V^2 T + \frac{d_1 T\log T}{\lambda_{\min}(\hat\Sigma)}) \wedge \frac{\lambda_{\max}(\hat\Sigma)V^2 T\log T + d_1 T\log^2 T}{\lambda_{\min}(\hat\Sigma) + (N\wedge T)\delta^2}.$$

Therefore, no matter if event A holds or not, we have

$$R_{\theta'_*,\theta_*}^{\pi}(T) \in \mathcal{O}\left(d_1\sqrt{T}\log T \wedge (V^2 T + \frac{d_1 T\log T}{\lambda_{\min}(\hat\Sigma)}) \wedge \frac{\lambda_{\max}(\hat\Sigma)V^2 T\log T + d_1 T\log^2 T}{\lambda_{\min}(\hat\Sigma) + (N\wedge T)\delta^2}\right),$$

thereby completing the proof of Theorem 1.

### E.1 Proof of Lemma 13

### E.1.1 Regret is $\mathcal{O}(d_1\sqrt{T}\log T)$

For any $t \ge 1$, suppose $\theta_* \in \mathcal{C}_{t-1}$, then from the definition of $(p_t, \tilde\theta_t)$, we have

$$
\begin{aligned}
r_{\theta_*}^*(x_t, y_t) - r_{\theta_*}(x_t, y_t, p_t) &= p_{\theta_*}^*(x_t, y_t)(\alpha_*^\top x_t + \beta_* y_t p_{\theta_*}^*(x_t, y_t)) - p_t(\alpha_*^\top x_t + \beta_* y_t p_t) \\
&\le p_t(\tilde\alpha_t^\top x_t + \tilde\beta_t y_t p_t) - p_t(\alpha_*^\top x_t + \beta_* y_t p_t) \\
&\le u\|A_t\|_{V_{t-1}^{-1}}\|\tilde\theta_t - \theta_*\|_{V_{t-1}} \\
&\le 2uw_{t-1}\|A_t\|_{V_{t-1}^{-1}} \\
&\le \max\{2u, 2u(u_\alpha + u_\beta u)\}w_T\left(\|A_t\|_{V_{t-1}^{-1}} \wedge 1\right),
\end{aligned}
$$

where we define $A_t = \left[x_t^\top, y_t p_t\right]^\top \in \mathbb{R}^{d_1+1}$ and the last equality holds because $|r(\theta, x, p)| \le u(u_\alpha + u_\beta u)$ and $w_T \ge \max\{1, w_t\}$. Therefore, we have

$$
\begin{aligned}
\sum_{t=1}^T r_{\theta_*}^*(x_t, y_t) - r_{\theta_*}(x_t, y_t, p_t) &\le \max\{2u, 2u(u_\alpha + u_\beta u)\}w_T\sqrt{T}\sqrt{\sum_{t=1}^T\left(\|A_t\|_{V_{t-1}^{-1}}^2 \wedge 1\right)} \\
&\overset{(i)}{\le} \max\{2u, 2u(u_\alpha + u_\beta u)\}w_T\sqrt{T}\sqrt{2(d_1+1)\log\left(\frac{(d_1+1)\lambda + TL^2}{(d_1+1)\lambda}\right)} \\
&\overset{(ii)}{\in} \mathcal{O}(d_1\sqrt{T}\log T).
\end{aligned}
$$

where (i) follows from [17, Lemma 19.4], and we use the same notation $L = \sqrt{x_{\max}^2 + y_{\max}^2 u^2}$ as in the proof of Lemma 6. Moreover, (ii) holds because $w_t \in \mathcal{O}\left(\sqrt{(d_1+1)\log T}\right)$ by setting $\delta = 1/T^2$.

Therefore, we have that the expected regret of Algorithm 1 without checking the condition (17) is bounded as

$$\sum_{t=1}^T \mathbb{E}[r_{\theta_*}^*(x_t, y_t) - r_{\theta_*}(x_t, y_t, p_t)] \in \mathcal{O}(d_1\sqrt{T}\log T) + 2u(u_\alpha + u_\beta u)\sum_{t=1}^T \frac{1}{T^2} \in \mathcal{O}(d_1\sqrt{T}\log T).$$

### E.1.2 Rerget is $\mathcal{O}\left(V^2 T + \frac{d_1 T \log T}{\lambda_{\min}(\hat{\Sigma})}\right)$

By subsection E.1.1, it is trivial if $\lambda_{\min}(\hat{\Sigma}) \leq T^{1/2}$. If $\lambda_{\min}(\hat{\Sigma}) \geq T^{1/2}$, for any $t \geq 1$, suppose $\theta_* \in C_{t-1}$, then from the definition of $(p_t, \tilde{\theta}_t)$, we have

$$
\begin{aligned}
r_{\theta_*}^*(x_t, y_t) - r_{\theta_*}(x_t, y_t, p_t) &= p_{\theta_*}^*(x_t, y_t)(\alpha_*^\top x_t + \beta_* y_t p_{\theta_*}^*(x_t, y_t)) - p_t(\alpha_*^\top x_t + \beta_* y_t p_t) \\
&= -(\beta_* y_t)(p_{\tilde{\theta}_t}^*(x_t, y_t) - p_{\theta_*}^*(x_t, y_t))^2 \\
&\overset{(i)}{\leq} \frac{u_\beta(y_{\max}^2 u_\alpha^2 + x_{\max}^2 u_\beta^2)\|\tilde{\theta}_t - \theta_*\|^2}{4 l_\beta^4} \\
&\leq \frac{u_\beta(y_{\max}^2 u_\alpha^2 + x_{\max}^2 u_\beta^2)\hat{w}_{t,n}^2}{4 l_\beta^4} \overset{(ii)}{\in} \mathcal{O}\left(V^2 + \frac{d_1 \log T}{\lambda_{\min}(\hat{\Sigma})}\right),
\end{aligned} \tag{19}
$$

where (i) holds by following Lemma 4 and (ii) holds because $\hat{w}_{t,n} \in \mathcal{O}\left(V + \frac{\sqrt{d_1 \log T}}{\sqrt{\lambda_{\min}(\hat{\Sigma})}}\right)$ by setting $\delta = 1/T^2$. Therefore, we have that the expected regret of Algorithm 1 without checking the condition (17) is bounded as

$$
\begin{aligned}
\sum_{t=1}^T \mathbb{E}[r^*(\theta_*, x_t) - r(\theta_*, x_t, p_t)] &\in \mathcal{O}\left(V^2 T + \frac{d_1 T \log T}{\lambda_{\min}(\hat{\Sigma})}\right) + 2u(u_\alpha + u_\beta u) \sum_{t=1}^T \frac{1}{T^2} \\
&\in \mathcal{O}\left(V^2 T \log T + \frac{d_1 T \log T}{\lambda_{\min}(\hat{\Sigma})}\right).
\end{aligned}
$$

### E.1.3 Regret is $\mathcal{O}\left(\frac{\lambda_{\max}(\hat{\Sigma})V^2 T \log T + d_1 T \log^2(T)}{\lambda_{\min}(\hat{\Sigma}) + (N \wedge T)\delta^2}\right)$

By subsection E.1.1, it is trivial if $\lambda_{\min}(\hat{\Sigma}) + (N \wedge T)\delta^2 \lesssim T^{1/2} \log T$. By subsection E.1.2, it is trivial if $\lambda_{\min}(\hat{\Sigma}) \gtrsim (N \wedge T)\delta^2$. Therefore, if $\lambda_{\min}(\hat{\Sigma}) + (N \wedge T)\delta^2 \gtrsim T^{1/2} \log T$ and $\lambda_{\min}(\hat{\Sigma}) \lesssim (N \wedge T)\delta^2$, we have $\lambda_{\min}(\hat{\Sigma}) \lesssim N\delta^2$ and $\delta^2 \gtrsim \frac{\log T}{\sqrt{N}}$. Then, by inequality (19) and applying Lemma 9, for any $t \geq 1$, suppose $\theta_* \in C_{t-1}$, we have

$$
\begin{aligned}
\sum_{t=1}^T \mathbb{E}[r_{\theta_*}^*(x_t, y_t) - r_{\theta_*}(x_t, y_t, p_t)] &\in \mathcal{O}\left(\sum_{t=1}^T \frac{\eta_T^2}{\lambda_{\min}(\hat{\Sigma}) + (N \wedge t)\delta^2}\right) + 2u(u_\alpha + u_\beta u) \sum_{t=1}^T \frac{1}{T^2} \\
&\in \mathcal{O}\left(\sum_{t=1}^T \frac{\lambda_{\max}(\hat{\Sigma})V^2 + d_1 \log T}{\lambda_{\min}(\hat{\Sigma}) + (N \wedge t)\delta^2}\right) + 2u(u_\alpha + u_\beta u) \sum_{t=1}^T \frac{1}{T^2},
\end{aligned}
$$

where the last inequality holds because $\eta_T^2 \in \mathcal{O}(w_{T,N}^2) \in \mathcal{O}(\lambda_{\max}(\hat{\Sigma})V^2 + d_1 \log T)$. If $T \leq N$, we have

$$
\begin{aligned}
\mathcal{O}\left(\sum_{t=1}^T \frac{\lambda_{\max}(\hat{\Sigma})V^2 + d_1 \log T}{\lambda_{\min}(\hat{\Sigma}) + (N \wedge t)\delta^2}\right) &\in \mathcal{O}\left(\sum_{t=1}^T \frac{\lambda_{\max}(\hat{\Sigma})V^2 + d_1 \log T}{t\delta^2}\right) \\
&\in \mathcal{O}\left(\sum_{t=1}^T \frac{\lambda_{\max}(\hat{\Sigma})V^2 \log T + d_1 \log^2(T)}{\delta^2}\right) \\
&\in \mathcal{O}\left(\sum_{t=1}^T \frac{\lambda_{\max}(\hat{\Sigma})V^2 T \log T + d_1 T \log^2(T)}{(N \wedge T)\delta^2}\right) \\
&\in \mathcal{O}\left(\frac{\lambda_{\max}(\hat{\Sigma})V^2 T \log T + d_1 T \log^2(T)}{\lambda_{\min}(\hat{\Sigma}) + (N \wedge T)\delta^2}\right).
\end{aligned}
$$

If $T \geq N$, we have

$$
\mathcal{O}\left(\sum_{t=1}^T \frac{\lambda_{\max}(\hat{\Sigma})V^2 + d_1 \log T}{\lambda_{\min}(\hat{\Sigma}) + (N \wedge t)\delta^2}\right) \in \mathcal{O}\left(\sum_{t=1}^N \frac{\lambda_{\max}(\hat{\Sigma})V^2 + d_1 \log T}{\lambda_{\min}(\hat{\Sigma}) + t\delta^2}\right) + \mathcal{O}\left(\sum_{t=N+1}^T \frac{\lambda_{\max}(\hat{\Sigma})V^2 + d_1 \log T}{\lambda_{\min}(\hat{\Sigma}) + N\delta^2}\right)
$$

$$\in \mathcal{O}\left(\sum_{t=1}^{N} \frac{\lambda_{\max}(\hat{\Sigma})V^2 + d_1 \log T}{t\delta^2}\right) + \mathcal{O}\left(\frac{\lambda_{\max}(\hat{\Sigma})V^2 T + d_1 T \log T}{\lambda_{\min}(\hat{\Sigma}) + N\delta^2}\right)$$

$$\in \mathcal{O}\left(\frac{\lambda_{\max}(\hat{\Sigma})V^2 \log(N) + d_1 \log T \log(N)}{\delta^2}\right) + \mathcal{O}\left(\frac{\lambda_{\max}(\hat{\Sigma})V^2 T + d_1 T \log T}{\lambda_{\min}(\hat{\Sigma}) + N\delta^2}\right)$$

$$\in \mathcal{O}\left(\frac{\lambda_{\max}(\hat{\Sigma})V^2 T \log T + d_1 T \log^2(T)}{(N \wedge T)\delta^2}\right),$$

thereby completing the proof of Lemma 13.

# F   Proof of Theorem 2

For simplicity, we provide the proof under the assumption that

$$\epsilon_t \overset{\text{i.i.d.}}{\sim} \mathcal{N}(0,1) \quad \text{and} \quad \hat{\epsilon}_n \overset{\text{i.i.d.}}{\sim} \mathcal{N}(0,1).$$

The proof proceeds in two principal steps F.1 and F.2. In the first step F.1, we will show that for any policy $\pi \in \Pi$, we have

$$\sup_{(\theta'_*, \theta_*) \in \mathcal{J}} R^{\pi}_{\theta'_*, \theta_*}(T) \in \Omega\left(\sqrt{T} \wedge \max\{\frac{T}{\delta^{-2} + V^{-2}}, \frac{T}{\delta^{-2} + (N \wedge T)\delta^2 + \lambda_{\min}(\hat{\Sigma})}\}\right).$$

In the second step F.2, we will show that for any admissible policy $\pi \in \Pi^{\circ}$, if either of the following conditions holds: 1) $V^2 \in \Omega(T^{-1/2})$, or 2) $\lambda_{\min}(\hat{\Sigma}) \in \mathcal{O}(\sqrt{T})$ and $\delta^2 \in \mathcal{O}(T^{-1/2})$, we have

$$\sup_{(\theta'_*, \theta_*) \in \mathcal{J}} R^{\pi}_{\theta'_*, \theta_*}(T) \in \tilde{\Omega}(\sqrt{T}).$$

Therefore, if $\delta^2 \lesssim \max\{V^2, \frac{1}{\lambda_{\min}(\hat{\Sigma})}\} \lesssim T^{-1/2}$, we have

$$\sup_{(\theta'_*, \theta_*) \in \mathcal{J}} R^{\pi}_{\theta'_*, \theta_*}(T) \in \Omega\left(\sqrt{T} \wedge \max\{\frac{T}{\delta^{-2} + V^{-2}}, \frac{T}{\delta^{-2} + \lambda_{\min}(\hat{\Sigma})}\}\right)$$

$$\in \Omega(\delta^2 T),$$

where the first inequality holds because $(n \wedge T)\delta^2 \le T\delta^2 \lesssim \sqrt{T}$.

If $\delta^2 \gtrsim T^{-1/2}$ and $V^2 \lesssim T^{-1/2}$, we have

$$\sup_{(\theta'_*, \theta_*) \in \mathcal{J}} R^{\pi}_{\theta'_*, \theta_*}(T) \in \Omega\left(\sqrt{T} \wedge \max\{\frac{T}{V^{-2}}, \frac{T}{(N \wedge T)\delta^2 + \lambda_{\min}(\hat{\Sigma})}\}\right)$$

$$\in \Omega\left(\sqrt{T} \wedge V^2 T + \frac{T}{(N \wedge T)\delta^2 + \lambda_{\min}(\hat{\Sigma})}\right).$$

If $\max\{V^2, \frac{1}{\lambda_{\min}(\hat{\Sigma})}\} \lesssim \delta^2 \lesssim T^{-1/2}$, we have

$$\sup_{(\theta'_*, \theta_*) \in \mathcal{J}} R^{\pi}_{\theta'_*, \theta_*}(T) \in \Omega\left(\sqrt{T} \wedge \max\{\frac{T}{V^{-2}}, \frac{T}{\lambda_{\min}(\hat{\Sigma})}\}\right)$$

$$\Omega\left(\sqrt{T} \wedge V^2 T + \frac{T}{\lambda_{\min}(\hat{\Sigma})}\right) \in \Omega\left(\sqrt{T} \wedge V^2 T + \frac{T}{(N \wedge T)\delta^2 + \lambda_{\min}(\hat{\Sigma})}\right).$$

Therefore, combining all the cases analyzed above completes the proof of Theorem 2.

## F.1   Details for step 1

Let $H_t = (\hat{\varepsilon}_1, \ldots, \hat{\varepsilon}_N, \varepsilon_1, \ldots, \varepsilon_{t-1}, x_1, \ldots, x_{t-1}, y_1, \ldots, y_{t-1})$ denotes the history before time $t-1$. We define $w = (\theta'_*, \theta_*) \in \mathcal{J}$. Then, we first apply the multivariate van Trees inequality to provide part of the lower bound in Theorem 2. Given $w \in \mathcal{J}$, $H_t$ has the Fisher information matrix:

$$\mathcal{I}^{\pi}_t(H_t) = \mathbb{E}^{\pi}_{\theta'_*, \theta_*} \mathcal{I}_t(H_t),$$

where

$$\mathcal{I}_t(H_t) = \begin{bmatrix} \sum_{i=1}^{N} \hat{x}_i \hat{x}_i^\top & \sum_{i=1}^{N} \hat{x}_i \hat{p}_i \hat{y}_i & 0 & 0 \\ \sum_{i=1}^{N} \hat{p}_i \hat{y}_i \hat{x}_i^\top & \sum_{i=1}^{N} \hat{p}_i^2 \hat{y}_i^2 & 0 & 0 \\ 0 & 0 & \sum_{i=1}^{t-1} x_i x_i^\top & \sum_{i=1}^{t-1} x_i p_i y_i \\ 0 & 0 & \sum_{i=1}^{t-1} p_i y_i x_i^\top & \sum_{i=1}^{t-1} p_i^2 y_i^2 \end{bmatrix}.$$

Given a prior distribution $q(\cdot)$ for $w$ on a subspace $W_1 \subseteq \mathcal{J}$, which we shall specify later, by applying the multivariate van Trees inequality, we obtain

$$\sup_{(\theta'_*, \theta_*) \in \mathcal{J}} R_{\theta'_*, \theta_*}^\pi(T) \geq \sup_{w \in W_1} R_{\theta'_*, \theta_*}^\pi(T) \geq l_\beta \sum_{t=1}^{T} \mathbb{E}_q \mathbb{E}_{\theta'_*, \theta_*}^\pi [(p_t - p_\theta^*(x_t, y_t))^2]$$

$$= l_\beta \sum_{t=1}^{T} \mathbb{E}_{x_t, y_t} \mathbb{E}_q \mathbb{E}_{\theta'_*, \theta_*}^\pi [(p_t - p_{\theta_*}^*(x_t, y_t))^2]$$

$$\geq l_\beta \sum_{t=1}^{T} \frac{\mathbb{E}_{x_t} \left[ (\mathbb{E}_q[C(w)^\top \frac{\partial p_{\theta_*}^*(x_t, y_t)}{\partial w}])^2 \right]}{\mathcal{I}(q) + \mathbb{E}_q \mathbb{E}_{\theta'_*, \theta_*}^\pi [C(w)^\top \mathcal{I}_t(H_t) C(w)]},$$

where $\mathcal{I}(q) = \int_{W_1} \left( \sum_{j=1}^{2d_1+2} \sum_{k=1}^{2d_1+2} \frac{\partial}{\partial w_j} (C_j(w) q(w)) \frac{\partial}{\partial w_k} (C_k(w) q(w)) \right) \frac{1}{q(w)} dw$ and $C(\cdot) : \mathbb{R}^{2d_1+2} \to \mathbb{R}^{2d_1+2}$ is a function of $w$ that are waiting to be specified later. In what follows, we will specify the subspace $W_1$, prior $q$ and functions $C(\cdot)$ to achieve part of the desired lower bound.

We consider the subspace $W_1$ defined as follows:

$$\{(\theta'_*, \theta_*) \in \mathbb{R}^{2d_1+2} \mid \theta_* - \epsilon V \leq \theta'_* \leq \theta_* + \epsilon V \quad \text{and} \quad \bar{\theta} - \epsilon \delta \leq \theta_* \leq \bar{\theta} + \epsilon \delta\}$$

where there always exist $\bar{\theta}$ and $\epsilon$ such that $W_1 \subseteq \mathcal{J}$. We choose the prior $q(\cdot)$ defined on the $W_1$ as follows:

$$q(\theta'_*, \theta_*) = \frac{1}{\epsilon^{2d_1+2} V^{d_1+1} \delta^{d_1+1}} \cos^2 \left( \frac{\pi (\beta'_* - \beta_*)}{2 \epsilon V} \right) \cos^2 \left( \frac{\pi (\beta_* - \bar{\beta})}{2 \epsilon \delta} \right)$$

$$\cdot \prod_{i=1}^{d_1} \cos^2 \left( \frac{\pi (\alpha'_{*,i} - \alpha_{*,i})}{2 \epsilon V} \right) \cos^2 \left( \frac{\pi (\alpha_{*,i} - \bar{\alpha}_i)}{2 \epsilon \delta} \right).$$

In the following, we provide lower bounds by choosing 3 different functions $C(\cdot)$.

**Step 1.1:** If we choose $C(w) = (0, 0, \alpha_*, 2\beta_*)$, then the following calculation applies:

$$\mathbb{E}_q \mathbb{E}_{\theta'_*, \theta_*}^\pi [C(w)^\top \mathcal{I}_t(H_t) C(w)] = \sum_{j=1}^{t-1} \mathbb{E}_q \mathbb{E}_{\theta'_*, \theta_*}^\pi [(\alpha_*^\top x_j + 2\beta_* y_j p_j)^2]$$

$$\leq 4u_\beta^2 \sum_{j=1}^{t-1} \mathbb{E}_q \mathbb{E}_{\theta'_*, \theta_*}^\pi [(p_j - p_{\theta_*}^*(x_j, y_j))^2];$$

$$\mathbb{E}_{x_t, y_t} \left[ (\mathbb{E}_q[C(w)^\top \frac{\partial p_{\theta_*}^*(x_t, y_t)}{\partial w}])^2 \right] = \mathbb{E}_{x_t, y_t} \left[ (\mathbb{E}_q[\frac{\alpha_*^\top x_t}{2\beta_* y_t}])^2 \right] \in \Omega(1);$$

$$\mathcal{I}(q) \in \mathcal{O}((1 + \frac{1}{\delta} + \frac{1}{V})^2).$$

The above three inequalities imply that

$$\sum_{t=1}^{T} \mathbb{E}_q \mathbb{E}_{\theta'_*, \theta_*}^\pi [(p_t - p_{\theta_*}^*(x_t, y_t))^2] \in \Omega \left( \sum_{t=1}^{T} \frac{1}{\mathcal{O}((1 + \frac{1}{\delta} + \frac{1}{V})^2) + \sum_{j=1}^{t-1} \mathbb{E}_q \mathbb{E}_{\theta'_*, \theta_*}^\pi [(p_j - p_{\theta_*}^*(x_j, y_j))^2]} \right)$$

$$\in \Omega \left( \frac{T}{\delta^{-2} + V^{-2} + \sum_{t=1}^{T} \mathbb{E}_q \mathbb{E}_{\theta'_*, \theta_*}^\pi [(p_t - p_{\theta_*}^*(x_t, y_t))^2]} \right).$$

Then, by the fact ([31, EC.18]) that

$$x^2 + bx + c \geq 0 \text{ for } b > 0, c < 0, x \geq 0 \text{ implies } x \geq \frac{1}{\sqrt{2}+1} \min\left\{\sqrt{|c|}, \frac{2|c|}{b}\right\}, \quad (20)$$

we have

$$\sup_{(\theta'_*, \theta_*) \in \mathcal{J}} R^\pi_{\theta'_*, \theta_*}(T) \in \Omega\left(\sqrt{T} \wedge \frac{T}{\delta^{-2} + V^{-2}}\right). \quad (21)$$

**Step 1.2:** If we choose $C(w) = (\alpha_*, 2\beta_*, \alpha_*, 2\beta_*)$, then the following calculation applies:

$$\mathbb{E}_q \mathbb{E}^\pi_{\theta'_*, \theta_*}[C(w)^\top \mathcal{I}_t(H_t) C(w)]$$

$$= \sum_{i=1}^N (\alpha_*^\top \hat{x}_i + 2\beta_* \hat{y}_i \hat{p}_i)^2 + \sum_{j=1}^{t-1} \mathbb{E}_q \mathbb{E}^\pi_{\theta'_*, \theta_*}[(\alpha_*^\top x_j + 2\beta_* y_j p_j)^2]$$

$$\leq 4u_\beta^2 \sum_{i=1}^N (p^*_{\theta_*}(\hat{x}_i, \hat{y}_i) - \hat{p}_i)^2 + 4u_\beta^2 \sum_{j=1}^{t-1} \mathbb{E}_q \mathbb{E}^\pi_{\theta'_*, \theta_*}[(p_j - p^*_{\theta_*}(x_j, y_j))^2]$$

$$\leq 8u_\beta^2 \sum_{i=1}^N (p^*_{\theta_*}(\hat{x}_i, \hat{y}_i) - \hat{p}(\hat{x}_i, \hat{y}_i))^2 + 8u_\beta^2 \sum_{i=1}^N (\hat{p}(\hat{x}_i, \hat{y}_i) - \hat{p}_i)^2 + 4u_\beta^2 \sum_{j=1}^{t-1} \mathbb{E}_q \mathbb{E}^\pi_{\theta'_*, \theta_*}[(p_j - p^*_{\theta_*}(x_j, y_j))^2]$$

$$\lesssim N\delta^2 + \lambda_{\min}(\hat{\Sigma}) + \sum_{j=1}^{t-1} \mathbb{E}_q \mathbb{E}^\pi_{\theta'_*, \theta_*}[(p_j - p^*_{\theta_*}(x_j, y_j))^2],$$

where the last inequality holds by Lemma 10. We also have

$$\mathbb{E}_{x_t, y_t}\left[\left(\mathbb{E}_q[C(w)^\top \frac{\partial p^*_{\theta_*}(x_t, y_t)}{\partial w}]\right)^2\right] = \mathbb{E}_{x_t, y_t}\left[\left(\mathbb{E}_q[\frac{\alpha_*^\top x_t}{2\beta_* y_t}]\right)^2\right] \in \Omega(1) \quad \text{and} \quad \mathcal{I}(q) \in \mathcal{O}((1 + \frac{1}{\delta})^2).$$

The above inequalities imply that

$$\sum_{t=1}^T \mathbb{E}_q \mathbb{E}^\pi_{\theta'_*, \theta_*}[(p_t - p^*_{\theta_*}(x_t, y_t))^2]$$

$$\in \Omega\left(\sum_{t=1}^T \frac{1}{\mathcal{O}((1 + \frac{1}{\delta})^2) + N\delta^2 + \lambda_{\min}(\hat{\Sigma}) + \sum_{j=1}^{t-1} \mathbb{E}_q \mathbb{E}^\pi_{\theta'_*, \theta_*}[(p_j - p^*_{\theta_*}(x_j, y_j))^2]}\right)$$

$$\in \Omega\left(\frac{T}{\delta^{-2} + N\delta^2 + \lambda_{\min}(\hat{\Sigma}) + \sum_{t=1}^T \mathbb{E}_q \mathbb{E}^\pi_{\theta'_*, \theta_*}[(p_t - p^*_{\theta_*}(x_t, y_t))^2]}\right).$$

Therefore, by the fact (20), we have

$$\sup_{(\theta'_*, \theta_*) \in \mathcal{J}} R^\pi_{\theta'_*, \theta_*}(T) \in \Omega\left(\sqrt{T} \wedge \frac{T}{\delta^{-2} + N\delta^2 + \lambda_{\min}(\hat{\Sigma})}\right). \quad (22)$$

**Step 1.3:** If we choose $C(w) = (-\hat{A}, 1, -\hat{A}, 1)$, then the following calculation applies:

$$\mathbb{E}_q \mathbb{E}^\pi_{\theta'_*, \theta_*}[C(w)^\top \mathcal{I}_t(H_t) C(w)] = \sum_{i=1}^N (-\hat{A}^\top \hat{x}_i + \hat{y}_i \hat{p}_i)^2 + \sum_{j=1}^{t-1} \mathbb{E}_q \mathbb{E}^\pi_{\theta'_*, \theta_*}[(-\hat{A}^\top x_j + y_j p_j)^2]$$

$$\leq y_{\max}^2 \sum_{i=1}^N (\hat{p}(\hat{x}_i, \hat{y}_i) - \hat{p}_i)^2 + y_{\max}^2 \sum_{j=1}^{t-1} \mathbb{E}_q \mathbb{E}^\pi_{\theta'_*, \theta_*}[(\hat{p}(x_j, y_j) - p_j)^2]$$

$$\lesssim \lambda_{\min}(\hat{\Sigma}) + (t-1)\delta^2 + \sum_{j=1}^{t-1} \mathbb{E}_q \mathbb{E}^\pi_{\theta'_*, \theta_*}[(p_j - p^*_{\theta_*}(x_j, y_j))^2],$$

where the last inequality holds by Lemma 10. We also have

$$\mathbb{E}_{x_t,y_t}\left[(\mathbb{E}_q[C(w)^\top \frac{\partial p_{\theta_*}^*(x_t,y_t)}{\partial w}])^2\right] = \mathbb{E}_{x_t,y_t}\left[(\mathbb{E}_q[\frac{\hat{A}^\top x_t}{2\beta_* y_t} + \frac{\alpha_*^\top x_t}{2\beta_*^2 y_t}])^2\right] \in \Omega(1) \quad \text{and} \quad \mathcal{I}(q) \in \mathcal{O}((1+\frac{1}{\delta})^2).$$

The above inequalities imply that

$$\sum_{t=1}^{T} \mathbb{E}_q \mathbb{E}_{\theta_*',\theta_*}^\pi[(p_t - p_{\theta_*}^*(x_t,y_t))^2] \in \Omega\left(\frac{T}{\delta^{-2} + T\delta^2 + \lambda_{\min}(\hat{\Sigma}) + \sum_{t=1}^{T} \mathbb{E}_q \mathbb{E}_{\theta_*',\theta_*}^\pi[(p_t - p_{\theta_*}^*(x_t,y_t))^2]}\right).$$

Therefore, by the fact (20), we have

$$\sup_{(\theta_*',\theta_*)\in\mathcal{J}} R_{\theta_*',\theta_*}^\pi(T) \in \Omega\left(\sqrt{T} \wedge \frac{T}{\delta^{-2} + T\delta^2 + \lambda_{\min}(\hat{\Sigma})}\right). \tag{23}$$

Combining the lower bounds in (22) and (23), we obtain

$$\sup_{(\theta_*',\theta_*)\in\mathcal{J}} R_{\theta_*',\theta_*}^\pi(T) \in \Omega\left(\sqrt{T} \wedge \frac{T}{\delta^{-2} + (N \wedge T)\delta^2 + \lambda_{\min}(\hat{\Sigma})}\right). \tag{24}$$

## F.2 Details for step 2

In this step, we show that if either of the following conditions holds:

1. $V \in \Omega(T^{-1/4})$, or
2. $\lambda_{\min}(\hat{\Sigma}) \in \mathcal{O}(\sqrt{T})$ and $\delta^2 \in \mathcal{O}(T^{-1/2})$,

then for any admissible policy $\pi \in \Pi^o$, there exists $(\theta_*',\theta_*) \in \Theta^\dagger \times \Theta^\dagger$ satisfying $\|\theta - \hat{\theta}\| \leq V$ and $\mathbb{E}_{x,y}[(\hat{p}(x,y) - p_{\theta_*}^*(x,y))^2] \in \Theta(\delta^2)$ such that

$$R_{\theta_*',\theta_*}^\pi(T) \in \Omega\left(\frac{\sqrt{T}}{(\log T)^{\lambda_0}}\right).$$

We first define two vectors of offline demand parameters and two vectors of online demand parameters as follows:

$$\theta_1' = (\alpha_1',\beta_1'), \quad \theta_1 = (\alpha_1,\beta_1), \quad \theta_2' = (\alpha_2',\beta_2'), \quad \theta_2 = (\alpha_2,\beta_2).$$

We consider $P_1^\pi$ and $P_2^\pi$ to be the probability measures induced by a common policy $\pi$, with two different sets of demand parameters $(\theta_1',\theta_1)$ and $(\theta_2',\theta_2)$, respectively. Formally, for each $i = 1,2$,

$$P_i^\pi\left(\hat{D}_1,\ldots\hat{D}_N,x_1,\ldots,x_T,y_1,\ldots,y_T,D_1,\ldots,D_T\right)$$

$$=P_X(x_1,\ldots,x_T)P_Y(y_1,\ldots,y_T)\prod_{i=1}^{N}\phi\big(\hat{D}_i - (\hat{\alpha}^\top \hat{x}_i + \hat{\beta}\hat{y}_i\hat{p}_i)\big) \cdot \prod_{t=1}^{T}\phi\big(D_t - (\alpha^\top x_t + \beta y_t p_t)\big),$$

where $P_X(x_1,\ldots,x_T)P_Y(y_1,\ldots,y_T)$ denotes the probability measure of online features and $\phi(x) = \frac{1}{\sqrt{2\pi}}e^{\frac{-x^2}{2}}$ is the density function of the standard normal distribution. Therefore, we have

$$\text{KL}(P_1^\pi, P_2^\pi) = \frac{1}{2}\sum_{i=1}^{N}\left[(\alpha_1' - \alpha_2')^\top \hat{x}_i + (\beta_1' - \beta_2')\hat{y}_i\hat{p}_i\right]^2 + \frac{1}{2}\sum_{t=1}^{T}\mathbb{E}_{\theta_1',\theta_1}^\pi\left[\left((\alpha_1 - \alpha_2)^\top x_t + (\beta_1 - \beta_2)y_t p_t\right)^2\right]. \tag{25}$$

**Step 2.1:** When $V \in \Omega(T^{-1/4})$, we set

$$\theta_1' = \theta_2', \quad \alpha_2 = (1-\Delta)\alpha_1, \quad \beta_2 = (1-2\Delta)\beta_1,$$

where $\Delta < 1/2$ and $(\theta_1',\theta_1)$ satisfies $\|\theta_1' - \theta_1\| \leq V$ and $\mathbb{E}_{x,y}[(\hat{p}(x,y) - p_{\theta_1}^*(x,y))^2] \in \Theta(\delta^2)$. Then, by equation (25), we have

$$\text{KL}(P_1^\pi, P_2^\pi) = \frac{\Delta^2}{2}\sum_{t=1}^{T}\mathbb{E}_{\theta_1',\theta_1}^\pi\left[\left(\alpha_1^\top x_t + 2\beta_1 y_t p_t\right)^2\right]$$

$$\leq 2\Delta^2 u_\beta^2 \sum_{t=1}^{T} \mathbb{E}_{\theta_1',\theta_1}^{\pi}\left[\left(p_t - p_{\theta_1}^*(x_t, y_t)\right)^2\right]. \tag{26}$$

Then, we define two sequences of intervals $I_{1,t}$ and $I_{2,t}$ for all $t \in [T]$ as follows:

$$I_{i,t} = \left[p_{\theta_i}^*(x_t, y_t) - \frac{\Delta l}{4(1-2\Delta)}, p_{\theta_i}^*(x_t, y_t) + \frac{\Delta l}{4(1-2\Delta)}\right], \forall i \in [2].$$

For any $(x, y) \in \mathcal{X} \times \mathcal{Y}$, we have

$$|p_{\theta_1}^*(x, y) - p_{\theta_2}^*(x, y)| = \frac{\Delta}{1-2\Delta} p_{\theta_1}^*(x, y) \geq \frac{\Delta l}{1-2\Delta},$$

which implies $I_{1,t} \cap I_{2,t} = \emptyset$ for every $t \in [T]$. Then, we have

$$R_{\theta_1',\theta_1}^{\pi}(T) + R_{\theta_2',\theta_2}^{\pi}(T) \geq l_\beta \left(\sum_{t=1}^{T} \mathbb{E}_{\theta_1',\theta_1}^{\pi}\left[\left(p_t - p_{\theta_1}^*(x_t, y_t)\right)^2\right] + \sum_{t=1}^{T} \mathbb{E}_{\theta_2',\theta_2}^{\pi}\left[\left(p_t - p_{\theta_2}^*(x_t, y_t)\right)^2\right]\right)$$

$$\geq \frac{\Delta^2 l^2 l_\beta}{16(1-2\Delta)^2} \sum_{t=1}^{T} \left(P_1^{\pi}\left(p_t \notin I_{1,t}\right) + P_2^{\pi}\left(p_t \notin I_{2,t}\right)\right)$$

$$\geq \frac{\Delta^2 l^2 l_\beta}{16(1-2\Delta)^2} \sum_{t=1}^{T} \left(P_1^{\pi}\left(p_t \notin I_{1,t}\right) + P_2^{\pi}\left(p_t \in I_{1,t}\right)\right)$$

$$\geq \frac{\Delta^2 l^2 l_\beta}{32(1-2\Delta)^2} T \exp(-\mathrm{KL}\left(P_1^{\pi}, P_2^{\pi}\right)). \tag{27}$$

By the definition of admissible policy, we have

$$R_{\theta_1',\theta_1}^{\pi}(T) + R_{\theta_2',\theta_2}^{\pi}(T) \leq 2K_0\sqrt{T}(\log T)^{\lambda_0}. \tag{28}$$

Therefore, combining inequalities (26), (27) and (28), we have

$$R_{\theta_1',\theta_1}^{\pi}(T) \geq l_\beta \sum_{t=1}^{T} \mathbb{E}_{\theta_1',\theta_1}^{\pi}\left[\left(p_t - p_{\theta_1}^*(x_t, y_t)\right)^2\right] \geq \frac{l_\beta}{2\Delta^2 u_\beta^2}\mathrm{KL}\left(P_1^{\pi}, P_2^{\pi}\right)$$

$$\geq \frac{l_\beta}{2\Delta^2 u_\beta^2} \log\left(\frac{\sqrt{T}\Delta^2 l^2 l_\beta}{64K_0(1-2\Delta)^2(\log T)^{\lambda_0}}\right).$$

Then, by setting $\Delta^2 \in \Theta(\frac{(\log T)^{\lambda_0}}{\sqrt{T}})$ such that $\frac{\sqrt{T}\Delta^2 l^2 l_\beta}{64K_0(1-2\Delta)^2(\log T)^{\lambda_0}} > 1$, we have $R_{\theta_1',\theta_1}^{\pi}(T) \in \Omega(\frac{\sqrt{T}}{(\log T)^{\lambda_0}})$

**Step 2.2:** When $V \in \mathcal{O}(T^{-1/4})$, $\lambda_{\min}(\hat{\Sigma}) \in \mathcal{O}(\sqrt{T})$ and $\delta^2 \leq \frac{l^2}{2}T^{-1/2}$, we set

$$\theta_1' = \theta_1, \quad \theta_2' = \theta_2, \quad \alpha_1 - \alpha_2 = -\hat{A}(\beta_1 - \beta_2) \quad \text{and} \quad \beta_2 = (1-\Delta)\beta_1,$$

and $(\theta_1', \theta_1)$ satisfies $\mathbb{E}_{x,y}[(\hat{p}(x, y) - p_{\theta_1}^*(x, y))^2] \in \Theta(\delta^2)$. Then, by equation (25), we have

$$\mathrm{KL}\left(P_1^{\pi}, P_{2'}^{\pi}\right) \leq \frac{1}{2}\sum_{i=1}^{N}\left[(\alpha_1 - \alpha_2)^{\top}\hat{x}_i + (\beta_1 - \beta_2)\hat{y}_i\hat{p}_i\right]^2 + \frac{1}{2}\sum_{t=1}^{T}\mathbb{E}_{\theta_1',\theta_1}^{\pi}\left[\left((\alpha_1 - \alpha_2)^{\top}x_t + (\beta_1 - \beta_2)y_tp_t\right)^2\right]$$

$$\leq \frac{\Delta^2 y_{\max}^2\beta_{\max}^2}{2}\sum_{i=1}^{N}(\hat{p}_i - \hat{p}(\hat{x}_i, \hat{y}_i))^2 + \frac{\Delta^2 y_{\max}^2\beta_{\max}^2}{2}\sum_{t=1}^{T}\mathbb{E}_{\theta_1',\theta_1}^{\pi}\left[(\hat{p}(x_t, y_t) - p_t)^2\right]$$

$$\lesssim \Delta^2\left(\lambda_{\min}(\hat{\Sigma}) + T\delta^2 + \sum_{t=1}^{T}\mathbb{E}_{\theta_1',\theta_1}^{\pi}\left[(\hat{p}(x_t, y_t) - p_t)^2\right]\right). \tag{29}$$

Then, we define two sequences of price function classes $I_{1,t}$ and $I_{2,t}$ for all $t \in [T]$ as follows:

$$I_{i,t} = \{p_t : \mathcal{X} \times \mathcal{Y} \to [l, u], \ \mathbb{E}_{x,y}[(p_t(x, y) - p_{\theta_i}^*(x, y))^2] \leq \frac{\Delta^2 l^2}{32(1-\Delta)^2}\}, \forall i \in [2].$$

For any $(x, y) \in \mathcal{X} \times \mathcal{Y}$, we have

$$
\begin{aligned}
\mathbb{E}_{x,y}[(p_{\theta_1}^*(x, y) - p_{\theta_2}^*(x, y))^2] &= \mathbb{E}_{x,y}[(\frac{\alpha_1^\top x}{2\beta_1 y} - \frac{\alpha_2^\top x}{2\beta_2 y})^2] \\
&= \mathbb{E}_{x,y}[(\frac{\alpha_1^\top x}{2\beta_1 y} - \frac{(\alpha_1 + \hat{A}\Delta\beta_1)^\top x}{2(1-\Delta)\beta_1 y})^2] \\
&= \frac{\Delta^2}{4(1-\Delta)^2} \cdot \mathbb{E}_{x,y}[(\frac{\alpha_1^\top x + \beta_1 \hat{A}^\top x}{\beta_1 y})^2] \\
&= \frac{\Delta^2}{4(1-\Delta)^2} \cdot \mathbb{E}_{x,y}[(2p_{\theta_1}^*(x, y) - \hat{p}(x, y))^2] \\
&\geq \frac{\Delta^2}{4(1-\Delta)^2} \cdot \left( \mathbb{E}_{x,y}[p_{\theta_1}^*(x, y)^2] - \frac{1}{2}\mathbb{E}_{x,y}[(p_{\theta_1}^*(x, y) - \hat{p}(x, y))^2] \right) \\
&\geq \frac{\Delta^2}{4(1-\Delta)^2}(l^2 - \delta^2) \geq \frac{\Delta^2 l^2}{8(1-\Delta)^2},
\end{aligned}
$$

where the last inequality holds because $\delta^2 \leq \frac{l^2}{2}T^{-1/2} \leq \frac{l^2}{2}$. which implies $I_{1,t} \cap I_{2,t} = \emptyset$ for every $t \in [T]$. Then, we have

$$
\begin{aligned}
&R_{\theta_1', \theta_1}^\pi(T) + R_{\theta_2', \theta_2}^\pi(T) \\
&\geq l_\beta \left( \sum_{t=1}^T \mathbb{E}_{\theta_1', \theta_1}^\pi \left[ (p_t - p_{\theta_1}^*(x_t, y_t))^2 \right] + \sum_{t=1}^T \mathbb{E}_{\theta_2', \theta_2}^\pi \left[ (p_t - p_{\theta_2}^*(x_t, y_t))^2 \right] \right) \\
&\geq l_\beta \left( \sum_{t=1}^T \mathbb{E}_{\theta_1', \theta_1, \mathcal{F}_{t-1}}^\pi \left[ \mathbb{E}_{x,y} \left[ (p_t(x, y) - p_{\theta_1}^*(x, y))^2 \right] \right] + \sum_{t=1}^T \mathbb{E}_{\theta_2', \theta_2, \mathcal{F}_{t-1}}^\pi \left[ \mathbb{E}_{x,y} \left[ (p_t(x, y) - p_{\theta_2}^*(x, y))^2 \right] \right] \right) \\
&\geq \frac{\Delta^2 l^2 l_\beta}{32(1-\Delta)^2} \left( \sum_{t=1}^T \left( P_1^\pi \left( p_t \notin I_{1,t} \right) + P_2^\pi \left( p_t \notin I_{2,t} \right) \right) \right) \\
&\geq \frac{\Delta^2 l^2 l_\beta}{32(1-\Delta)^2} \left( \sum_{t=1}^T \left( P_1^\pi \left( p_t \notin I_{1,t} \right) + P_2^\pi \left( p_t \in I_{1,t} \right) \right) \right) \\
&\geq \frac{\Delta^2 l^2 l_\beta}{64(1-\Delta)^2} T \exp(-\mathrm{KL}\left( P_1^\pi, P_2^\pi \right)). \quad (30)
\end{aligned}
$$

Therefore, combining inequalities (29), (30) and (28), we have

$$
R_{\theta_1', \theta_1}^\pi(T) \geq l_\beta \sum_{t=1}^T \mathbb{E}_{\theta_1', \theta_1}^\pi \left[ (p_t - p_{\theta_1}^*(x_t, y_t))^2 \right] \gtrsim \frac{1}{\Delta^2}\mathrm{KL}\left( P_1^\pi, P_2^\pi \right) - T\delta^2 - \lambda_{\min}(\hat{\Sigma})
$$

$$
\geq \frac{1}{\Delta^2} \log \left( \frac{\sqrt{T}\Delta^2 l^2 l_\beta}{128 K_0 (1-\Delta)^2 (\log T)^{\lambda_0}} \right) - T\delta^2 - \lambda_{\min}(\hat{\Sigma}).
$$

Then, because $\delta^2 \leq \frac{l^2}{2}T^{-1/2}$ and $\lambda_{\min}(\hat{\Sigma})$, we can always set $\Delta^2 \in \Theta(\frac{(\log T)^{\lambda_0}}{\sqrt{T}})$ such that $\frac{\sqrt{T}\Delta^2 l^2 l_\beta}{64 K_0 (1-2\Delta)^2 (\log T)^{\lambda_0}} > 1$ and $R_{\theta_1', \theta_1}^\pi(T) \in \Omega(\frac{\sqrt{T}}{(\log T)^{\lambda_0}})$.

# G   Proof of Theorem 4

The proof of Theorem 4 is similar to that of Theorem 2. Hence, we only highlight the key differences here. For simplicity, we provide the proof under the assumption that

$$
\epsilon_t \overset{\text{i.i.d.}}{\sim} \mathcal{N}(0, 1) \quad \text{and} \quad \hat{\epsilon}_n \overset{\text{i.i.d.}}{\sim} \mathcal{N}(0, 1).
$$

Let $H_t = (\hat{\varepsilon}_1, \ldots, \hat{\varepsilon}_N, \varepsilon_1, \ldots, \varepsilon_{t-1}, x_1, \ldots, x_{t-1}, y_1, \ldots, y_{t-1})$ denotes the history before time $t - 1$. We define $w = (\theta_*', \theta_*) \in \bar{\mathcal{J}}$. Then, we first apply the multivariate van Trees inequality to provide part of the lower bound in Theorem 4. Given $w \in \bar{\mathcal{J}}$, $H_t$ has the Fisher information matrix:

$$
\mathcal{I}_t^\pi(H_t) = \mathbb{E}_{\theta_*', \theta_*}^\pi \mathcal{I}_t(H_t),
$$

where

$$\mathcal{I}_t(H_t) = \begin{bmatrix} \sum_{i=1}^N \hat{x}_i \hat{x}_i^\top & \sum_{i=1}^N \hat{x}_i \hat{p}_i \hat{y}_i^\top & 0 & 0 \\ \sum_{i=1}^N \hat{y}_i \hat{p}_i \hat{x}_i^\top & \sum_{i=1}^N \hat{y}_i \hat{p}_i^2 \hat{y}_i^\top & 0 & 0 \\ 0 & 0 & \sum_{i=1}^{t-1} x_i x_i^\top & \sum_{i=1}^{t-1} x_i p_i y_i^\top \\ 0 & 0 & \sum_{i=1}^{t-1} y_i p_i x_i^\top & \sum_{i=1}^{t-1} y_i p_i^2 y_i^\top \end{bmatrix}.$$

Given a prior distribution $q(\cdot)$ for $w$ on a subspace $W_1 \subseteq \bar{\mathcal{J}}$, which we shall specify later, by applying the multivariate van Trees inequality, we obtain

$$\sup_{(\theta_*', \theta_*) \in \bar{\mathcal{J}}} R_{\theta_*', \theta_*}^\pi(T) \geq \sup_{w \in W_1} R_{\theta_*', \theta_*}^\pi(T) \geq l_\beta \sum_{t=1}^T \frac{\mathbb{E}_{x_t}\left[(\mathbb{E}_q[C(w)^\top \frac{\partial p_{\theta_*}^*(x_t, y_t)}{\partial w}])^2\right]}{\mathcal{I}(q) + \mathbb{E}_q \mathbb{E}_{\theta_*', \theta_*}^\pi [C(w)^\top \mathcal{I}_t(H_t) C(w)]},$$

where $\mathcal{I}(q) = \int_{W_1} \left( \sum_{j=1}^{2d_1+2d_2} \sum_{k=1}^{2d_1+2d_2} \frac{\partial}{\partial w_j}(C_j(w)q(w)) \frac{\partial}{\partial w_k}(C_k(w)q(w)) \right) \frac{1}{q(w)} dw$ and $C(\cdot) : \mathbb{R}^{2d_1+2d_2} \to \mathbb{R}^{2d_1+2d_2}$ is a function of $w$ that are waiting to be specified later. In what follows, we provide lower bounds by specifing two different the subspaces $W_1$, priors $q$ and functions $C(\cdot)$ to achieve part of the desired lower bound.

**Step 1:** We consider the subspace $W_1$ defined as follows:

$$\{(\theta_*', \theta_*) \in \mathbb{R}^{2d_1+2d_2} \mid \theta_* - \epsilon V \leq \theta_*' \leq \theta_* + \epsilon V \quad \text{and} \quad \bar{\theta} - \epsilon V \leq \theta_* \leq \bar{\theta} + \epsilon V\}$$

where there always exist $\bar{\theta}$ and $\epsilon$ such that $W_1 \subseteq \bar{\mathcal{J}}$. We choose the prior $q(\cdot)$ defined on the $W_1$ as follows:

$$q(\theta_*', \theta_*) = \frac{1}{\epsilon^{2d_1+2d_2} V^{2d_1+2d_2}} \prod_{i=1}^{d_1} \cos^2\left(\frac{\pi(\alpha_{*,i}' - \alpha_{*,i})}{2\epsilon V}\right) \cos^2\left(\frac{\pi(\alpha_{*,i} - \bar{\alpha}_i)}{2\epsilon V}\right)$$
$$\cdot \prod_{j=1}^{d_2} \cos^2\left(\frac{\pi(\beta_{*,j}' - \beta_{*,j})}{2\epsilon V}\right) \cos^2\left(\frac{\pi(\beta_{*,j} - \bar{\beta}_j)}{2\epsilon V}\right).$$

We choose $C(w) = (0, 0, \alpha_*, 2\beta_*)$, then the following calculation applies:

$$\mathbb{E}_q \mathbb{E}_{\theta_*', \theta_*}^\pi [C(w)^\top \mathcal{I}_t(H_t) C(w)] = \sum_{j=1}^{t-1} \mathbb{E}_q \mathbb{E}_{\theta_*', \theta_*}^\pi [(\alpha_*^\top x_j + 2\beta_*^\top y_j p_j)^2]$$
$$\leq 4 u_\beta^2 \sum_{j=1}^{t-1} \mathbb{E}_q \mathbb{E}_{\theta_*', \theta_*}^\pi [(p_j - p_{\theta_*}^*(x_j, y_j))^2];$$
$$\mathbb{E}_{x_t, y_t}\left[(\mathbb{E}_q[C(w)^\top \frac{\partial p_{\theta_*}^*(x_t, y_t)}{\partial w}])^2\right] = \mathbb{E}_{x_t, y_t}\left[(\mathbb{E}_q[\frac{\alpha_*^\top x_t}{2\beta_*^\top y_t}])^2\right] \in \Omega(1);$$
$$\mathcal{I}(q) \in \mathcal{O}((1 + \frac{1}{V})^2).$$

The above three inequalities imply that

$$\sum_{t=1}^T \mathbb{E}_q \mathbb{E}_{\theta_*', \theta_*}^\pi [(p_t - p_{\theta_*}^*(x_t, y_t))^2] \in \Omega\left(\sum_{t=1}^T \frac{1}{\mathcal{O}((1 + \frac{1}{V})^2) + \sum_{j=1}^{t-1} \mathbb{E}_q \mathbb{E}_{\theta_*', \theta_*}^\pi [(p_j - p_{\theta_*}^*(x_j, y_j))^2]}\right)$$
$$\in \Omega\left(\frac{T}{V^{-2} + \sum_{t=1}^T \mathbb{E}_q \mathbb{E}_{\theta_*', \theta_*}^\pi [(p_t - p_{\theta_*}^*(x_t, y_t))^2]}\right),$$

which implies

$$\sup_{(\theta_*', \theta_*) \in \bar{\mathcal{J}}} R_{\theta_*', \theta_*}^\pi(T) \in \Omega\left(\sqrt{T} \wedge V^2 T\right).$$

**Step 2:** Let $\tilde{\theta} = (\tilde{\alpha}, \tilde{\beta})$ be the eigenvector of $\hat{\Sigma}$ corresponding to the eigenvalue $\lambda_{\min}(\hat{\Sigma})$, normalized so that $\|\tilde{\theta}\|^2 = \|\alpha_*\|^2 + 4\|\beta_*\|^2$. We define

$$\delta^2 := \mathbb{E}_{x,y}[((\alpha_* - \tilde{\alpha})^\top x)^2] + u^2 \mathbb{E}_{x,y}[((2\beta_* - \tilde{\beta})^\top y)^2].$$

Because we have no restriction on $\delta^2$, we focus on the case that $\frac{1}{\lambda_{\min}(\hat{\Sigma})} \lesssim \delta^2 \lesssim T^{-1/2}$ and consider the subspace $W_1$ defined as follows:

$$\{(\theta'_*, \theta_*) \in \mathbb{R}^{2d_1 + 2d_2} \mid \theta_* - \epsilon V \le \theta'_* \le \theta_* + \epsilon V \quad \text{and} \quad \bar{\theta} - \epsilon\delta \le \theta_* \le \bar{\theta} + \epsilon\delta\},$$

where there always exist $\bar{\theta}$ and $\epsilon$ such that $W_1 \subseteq \bar{\mathcal{J}}$ and $\frac{1}{\lambda_{\min}(\hat{\Sigma})} \lesssim \delta^2 \lesssim T^{-1/2}$. We choose the prior $q(\cdot)$ defined on the $W_1$ as follows:

$$q(\theta'_*, \theta_*) = \frac{1}{\epsilon^{2d_1 + 2d_2} V^{d_1 + d_2} \delta^{d_1 + d_2}} \prod_{i=1}^{d_1} \cos^2\left(\frac{\pi(\hat{\alpha}_i - \alpha_i)}{2\epsilon V}\right) \cos^2\left(\frac{\pi(\alpha_i - \bar{\alpha}_i)}{2\epsilon\delta}\right)$$

$$\cdot \prod_{i=1}^{d_2} \cos^2\left(\frac{\pi(\hat{\beta}_i - \beta_i)}{2\epsilon V}\right) \cos^2\left(\frac{\pi(\beta_i - \bar{\beta}_i)}{2\epsilon\delta}\right).$$

We choose $C(w) = (\tilde{\alpha}, \tilde{\beta}, \tilde{\alpha}, \tilde{\beta})$, then the following calculation applies:

$$\mathbb{E}_q \mathbb{E}^\pi_{\theta'_*, \theta_*}[C(w)^\top \mathcal{I}_t(H_t) C(w)]$$

$$= \lambda_{\min}(\hat{\Sigma}) + \sum_{j=1}^{t-1} \mathbb{E}_q \mathbb{E}^\pi_{\theta'_*, \theta_*}[(\tilde{\alpha}^\top x_j + \tilde{\beta}^\top y_j p_j)^2]$$

$$\le \lambda_{\min}(\hat{\Sigma}) + 4T\mathbb{E}_{x,y}[((\alpha_* - \tilde{\alpha})^\top x)^2] + 4u^2 T\mathbb{E}_{x,y}[((2\beta_* - \tilde{\beta})^\top y)^2] + 2\sum_{j=1}^{t-1} \mathbb{E}_q \mathbb{E}^\pi_{\theta'_*, \theta_*}[(\alpha_*^\top x_j + 2\beta_*^\top y_j p_j)^2]$$

$$\le \lambda_{\min}(\hat{\Sigma}) + 4T\delta^2 + 8u_\beta^2 \sum_{t=1}^{T} \mathbb{E}_q \mathbb{E}^\pi_{\theta'_*, \theta_*}[(p_t - p^*_{\theta_*}(x_t, y_t))^2];$$

$$\mathbb{E}_{x_t, y_t}\left[(\mathbb{E}_q[C(w)^\top \frac{\partial p^*_{\theta_*}(x_t, y_t)}{\partial w}])^2\right] \in \Omega(1); \quad \text{and} \quad \mathcal{I}(q) \in \mathcal{O}((1 + \frac{1}{\delta})^2).$$

The above three inequalities imply that

$$\sum_{t=1}^{T} \mathbb{E}_q \mathbb{E}^\pi_{\theta'_*, \theta_*}[(p_t - p^*_{\theta_*}(x_t, y_t))^2]$$

$$\in \Omega\left(\sum_{t=1}^{T} \frac{1}{\mathcal{O}((1 + \frac{1}{\delta})^2) + \lambda_{\min}(\hat{\Sigma}) + t\delta^2 + \sum_{j=1}^{t-1} \mathbb{E}_q \mathbb{E}^\pi_{\theta'_*, \theta_*}[(p_j - p^*_{\theta_*}(x_j, y_j))^2]}\right)$$

$$\in \Omega\left(\frac{T}{\delta^{-2} + \lambda_{\min}(\hat{\Sigma}) + T\delta^2 + \sum_{t=1}^{T} \mathbb{E}_q \mathbb{E}^\pi_{\theta'_*, \theta_*}[(p_t - p^*_{\theta_*}(x_t, y_t))^2]}\right),$$

which implies

$$\sup_{(\theta'_*, \theta_*) \in \bar{\mathcal{J}}} R^\pi_{\theta'_*, \theta_*}(T) \in \Omega\left(\sqrt{T} \wedge \frac{T}{\delta^{-2} + \lambda_{\min}(\hat{\Sigma}) + T\delta^2}\right) \in \Omega\left(\sqrt{T} \wedge \frac{T}{\lambda_{\min}(\hat{\Sigma})}\right),$$

where the last inequality holds because $\frac{1}{\lambda_{\min}(\hat{\Sigma})} \lesssim \delta^2 \lesssim T^{-1/2}$. Hence, combining the two lower bounds obtained in the above steps, we have

$$\sup_{(\theta'_*, \theta_*) \in \bar{\mathcal{J}}} R^\pi_{\theta'_*, \theta_*}(T) \in \Omega\left(\sqrt{T} \wedge (V^2 T + \frac{T}{\lambda_{\min}(\hat{\Sigma})})\right),$$

thereby completing the proof of Theorem 4.

# H    Proof of Theorem 5

We begin by presenting Lemma 14, which establishes how closely the empirical bias $\|\hat{\theta}'_* - \hat{\theta}_*\|$ approximates the exact bias $\|\theta'_* - \theta_*\|$ and defer the proof of Lemma 14 to the end of this section.

**Lemma 14.** *Let $\{(\hat{D}_n, \hat{x}_n, \hat{y}_n, \hat{p}_n)\}_{n=1}^N$ denote the offline data and $\{(D_t, x_t, y_t, p_t)\}_{t=1}^{T'}$ the observations collected during the test phase of Algorithm 3. Under Assumption 1, if $T' = \Omega(\log T)$, then with probability at least $1 - \frac{2\epsilon}{3}$, we have*

$$|\|\theta'_* - \theta_*\| - \|\hat{\theta}'_* - \hat{\theta}_*\|| \le f \quad and$$

$$f := \frac{\lambda\sqrt{\alpha_{\max}^2 + \beta_{\max}^2}}{\lambda + \lambda_{\min}(\hat{\Sigma})} + \frac{R\sqrt{d_1 + d_2} + R\sqrt{2\log(3/\epsilon)}}{\sqrt{\lambda + \lambda_{\min}(\hat{\Sigma})}} \tag{31}$$

$$+ \frac{\lambda\sqrt{\alpha_{\max}^2 + \beta_{\max}^2}}{\lambda + \lambda_{\min}(\Sigma_{T'})} + \frac{R\sqrt{d_1 + d_2} + R\sqrt{2\log(3/\epsilon)}}{\sqrt{\lambda + \lambda_{\min}(\Sigma_{T'})}},$$

*where $\Sigma_{T'} := \sum_{t=1}^{T'} \begin{bmatrix} x_t \\ y_t p_t \end{bmatrix} \begin{bmatrix} x_t^\top & p_t y_t^\top \end{bmatrix}$. With probability at least $1 - \exp(-\frac{T'(u-l)^2\lambda_{\min}(\mathbb{E}[yy^\top])}{32L^2})$, we have*

$$\lambda_{\min}(\Sigma_{T'}) \ge \frac{T'(u-l)^2\lambda_{\min}(\mathbb{E}[yy^\top])}{8}.$$

Because $0 < \alpha < 1/2 < \beta$, with probability at least $1 - \exp(-\frac{T'(u-l)^2\lambda_{\min}(\mathbb{E}[yy^\top])}{32L^2})$, the quantity $f$ introduced in Lemma 14 satisfies

$$f \in \Theta\left(\frac{\sqrt{d_1 + d_2} + \sqrt{\log T}}{\sqrt{T^\alpha}}\right). \tag{32}$$

Fix $\alpha \in (0, \frac{1}{2})$ and set $T' = T^\alpha$. We now prove Theorem 5 by consider the following three cases.

**Case 1:** $V_{\text{true}} > 3f$.   In this case, by triangle inequality and Lemma 14, with probability at least $1 - \frac{2}{3T^2} - \exp(-\frac{T'(u-l)^2\lambda_{\min}(\mathbb{E}[yy^\top])}{32L^2})$,

$$\|\hat{\theta}'_* - \hat{\theta}_*\| \ge V_{\text{true}} - |V_{\text{true}} - \|\hat{\theta}'_* - \hat{\theta}_*\|| > 2f.$$

Therefore, we have

$$R_{\theta'_*, \theta_*}^\pi(T) \lesssim T^\alpha + (d_1 + d_2)\sqrt{T}\log T \lesssim (d_1 + d_2)\sqrt{T}\log T.$$

**Case 2:** $V_{\text{true}} \le f$.   In this case, by triangle inequality and Lemma 14, with probability at least $1 - \frac{2}{3T^2} - \exp(-\frac{T'(u-l)^2\lambda_{\min}(\mathbb{E}[yy^\top])}{32L^2})$,

$$\|\hat{\theta}'_* - \hat{\theta}_*\| \le V_{\text{true}} + |V_{\text{true}} - \|\hat{\theta}'_* - \hat{\theta}_*\|| \le 2f.$$

Therefore, by inequality (19), we have

$$R_{\theta'_*, \theta_*}^\pi(T) \lesssim T^\alpha + (T - T^\alpha)\mathbb{E}[\|\hat{\theta}'_* - \theta_*\|^2]$$

$$\lesssim T^\alpha + T\mathbb{E}[\|\hat{\theta}'_* - \theta'_*\|^2] + V_{\text{true}}^2 T$$

$$\lesssim T^\alpha + V_{\text{true}}^2 T + \frac{T(d_1 + d_2 + \log T)}{\lambda_{\min}(\Sigma)},$$

where the last inequality holds by following inequality (34).

**Case 3:** $f < V_{\text{true}} \le 3f$.   In this case, by inequality (32), we have

$$R_{\theta'_*, \theta_*}^\pi(T) \lesssim \max\left\{(T^\alpha + V_{\text{true}}^2 T + \frac{T(d_1 + d_2 + \log T)}{\lambda_{\min}(\Sigma)}), (d_1 + d_2)\sqrt{T}\log T\right\}$$

$$\lesssim T^\alpha + V_{\text{true}}^2 T + \frac{T(d_1 + d_2 + \log T)}{\lambda_{\min}(\Sigma)}.$$

Combining the three cases above, we obtain that for any $\alpha \in (0, \frac{1}{2})$,

$$R_{\theta'_*, \theta_*}^\pi(T) \in \begin{cases} \mathcal{O}\left((d_1 + d_2)\sqrt{T}\log T\right), & \text{if } V_{\text{true}}^2 \gtrsim T^{-\alpha}; \\ \mathcal{O}\left(T^\alpha + V_{\text{true}}^2 T + \frac{T(d_1 + d_2 + \log T)}{\lambda_{\min}(\Sigma)}\right), & \text{otherwise.} \end{cases}$$

When $\beta \geq 1$, $\frac{T}{\lambda_{\min}(\Sigma)} \in \mathcal{O}(1)$ and

$$R_{\theta'_*, \theta_*}^\pi(T) \in \begin{cases} \mathcal{O}\left((d_1 + d_2)\sqrt{T}\log T\right), & \text{if } V_{\text{true}}^2 \gtrsim T^{-\alpha}; \\ \tilde{\mathcal{O}}\left(T^\alpha + V_{\text{true}}^2 T\right), & \text{otherwise.} \end{cases}$$

Hence no single value of $\alpha \in (0, \frac{1}{2})$ is uniformly optimal: a smaller $\alpha$ yields lower regret when $V_{\text{true}}^2 \leq T^{\alpha-1}$, but increases the worst-case regret to $T^{1-\alpha}$ when $V_{\text{true}}^2 = \Theta(T^{-\alpha})$.

For $\beta \in (\frac{1}{2}, 1)$ and any $\alpha \in (0, 1 - \beta]$, we have

$$R_{\theta'_*, \theta_*}^\pi(T) \in \begin{cases} \mathcal{O}\left((d_1 + d_2)\sqrt{T}\log T\right), & \text{if } V_{\text{true}}^2 \gtrsim T^{-\alpha}; \\ \tilde{\mathcal{O}}\left(V_{\text{true}}^2 T + \frac{T(d_1 + d_2 + \log T)}{\lambda_{\min}(\Sigma)}\right), & \text{otherwise.} \end{cases}$$

In this regime, choosing $\alpha = 1 - \beta$ is optimal: for any $V_{\text{true}}$, the resulting regret is no greater than that obtained with any $\alpha \in (0, 1 - \beta)$.

For $\beta \in (\frac{1}{2}, 1)$ and $\alpha \in [1 - \beta, \frac{1}{2})$, no single choice is strictly preferred, for the same trade-off discussed in the $\beta > 1$ case, and we have

$$R_{\theta'_*, \theta_*}^\pi(T) \in \begin{cases} \mathcal{O}\left((d_1 + d_2)\sqrt{T}\log T\right), & \text{if } V_{\text{true}}^2 \gtrsim T^{-\alpha}; \\ \tilde{\mathcal{O}}\left(T^\alpha + V_{\text{true}}^2 T\right), & \text{otherwise.} \end{cases}$$

Therefore, we complete the proof of Theorem 5.

### H.1 Proof of Lemma 14

$$|\|\theta'_* - \theta_*\| - \|\hat{\theta}'_* - \hat{\theta}_*\|| \leq \|\theta'_* - \hat{\theta}'_*\| + \|\theta_* - \hat{\theta}_*\|$$

For the first term $\|\theta'_* - \hat{\theta}'_*\|$, with probability at least $1 - \epsilon/3$,

$$\|\theta'_* - \hat{\theta}'_*\| = \|V_{0,n}^{-1} \sum_{i=1}^n \begin{bmatrix} \hat{x}_i \\ \hat{y}_i \hat{p}_i \end{bmatrix} (\alpha_*'^\top \hat{x}_i + \beta_*'^\top \hat{y}_i \hat{p}_i + \hat{\epsilon}_i) - V_{0,n}^{-1} \sum_{i=1}^n \begin{bmatrix} \hat{x}_i \\ \hat{y}_i \hat{p}_i \end{bmatrix} (\alpha_*'^\top \hat{x}_i + \beta_*'^\top \hat{y}_i \hat{p}_i) - \lambda V_{0,n}^{-1} \theta'_*\|$$

$$\leq \lambda \|V_{0,n}^{-1} \theta'_*\| + \|V_{0,n}^{-1} \sum_{i=1}^n \begin{bmatrix} \hat{x}_i \\ \hat{y}_i \hat{p}_i \end{bmatrix} \hat{\epsilon}_i\|$$

$$\leq \frac{\lambda \|\theta'_*\|}{\lambda + \lambda_{\min}(\hat{\Sigma})} + \frac{R\sqrt{d_1 + d_2} + R\sqrt{2\log(3/\epsilon)}}{\sqrt{\lambda + \lambda_{\min}(\hat{\Sigma})}} \tag{33}$$

$$\leq \frac{\lambda \sqrt{\alpha_{\max}^2 + \beta_{\max}^2}}{\lambda + \lambda_{\min}(\hat{\Sigma})} + \frac{R\sqrt{d_1 + d_2} + R\sqrt{2\log(3/\epsilon)}}{\sqrt{\lambda + \lambda_{\min}(\hat{\Sigma})}}, \tag{34}$$

where the last inequality follows from Lemma 11 and the same argument used in the proof of Lemma 5.

Similarly, for fixed $\{(x_t, y_t, p_t)\}_{t \in [T']}$, with probability at least $1 - \epsilon/3$,

$$\|\theta_* - \hat{\theta}_{T'}\| \leq \frac{\lambda \sqrt{\alpha_{\max}^2 + \beta_{\max}^2}}{\lambda + \lambda_{\min}(\Sigma_{T'})} + \frac{R\sqrt{d_1 + d_2} + R\sqrt{2\log(3/\epsilon)}}{\sqrt{\lambda + \lambda_{\min}(\Sigma_{T'})}},$$

where we define $\Sigma_{T'} := \sum_{t=1}^{T'} \begin{bmatrix} x_t \\ y_t p_t \end{bmatrix} \begin{bmatrix} x_t^\top & p_t y_t^\top \end{bmatrix}$. For every $t \in [T']$, we have

$$\lambda_{\min} \left( \mathbb{E}[p_t^2]\mathbb{E}[y_t y_t^\top] - \mathbb{E}[p_t]^2 \mathbb{E}[y_t x_t^\top]\mathbb{E}[x_t x_t^\top]^{-1}\mathbb{E}[x_t y_t^\top] \right) \overset{(i)}{\geq} (\mathbb{E}[p_t^2] - \mathbb{E}[p_t]^2)\lambda_{\min}(\mathbb{E}[y_t y_t^\top])$$
$$= \frac{(u-l)^2 \lambda_{\min}(\mathbb{E}[yy^\top])}{4}.$$

where (i) holds by following Lemma 3. Also, by the definition of $L$ in Appendix A, for every $t \in [T']$, we have

$$\lambda_{\max}(\mathbb{E}[\begin{bmatrix} x_t \\ y_t p_t \end{bmatrix} \begin{bmatrix} x_t^\top & p_t y_t^\top \end{bmatrix}]) \leq L^2,$$

Then, by following the matrix Chernoff inequaility [27, Theorem 5.1.1], with probability at least $1 - \exp(-\frac{T'(u-l)^2 \lambda_{\min}(\mathbb{E}[yy^\top])}{32L^2})$,

$$\lambda_{\min}(\Sigma_{T'}) \geq \frac{T'(u-l)^2 \lambda_{\min}(\mathbb{E}[yy^\top])}{8},$$

thereby completing the proof of Lemma 14.

# I   Stochastic linear bandit with (biased) offline data

In this section, we outline that how our design of GCO3 and its regret–upper-bound analysis extend seamlessly to the stochastic linear bandit with biased offline data.

Firstly, we specify the model of stochastic linear bandit with (biased) offline data problem.

**Online model.** Consider a learner who make decisions over a time horizon of $T$ periods. In round $t$, the learner is given the decision set $\mathcal{A}_t \subset \mathbb{R}^d$, from which it chooses an action $a_t \in \mathcal{A}_t$ and receives reward

$$x_t = \theta_*^\top a_t + \eta_t,$$

where $\theta_* \in \mathbb{R}^d$ denotes the unknown online parameter vector and $\{\eta_t\}_{t \geq 1}$ is an sequence of independent random noise with zero mean and $R-$subgaussian.

**Offline data model.** The learner have access to a pre-existing offline dataset prior to the online learning process. Let this dataset consist of $N$ samples $\{\hat{a}_n, \hat{x}_n\}_{n \in [N]}$. For each $n \in [N]$, the reward realization $\hat{x}_n$ under the historical action $\hat{a}_n$ is generated according to the linear model

$$\hat{x}_n = \theta_*'^\top \hat{a}_n + \hat{\eta}_n,$$

where $\theta_*'$ is the unknown offline parameter vector $\{\hat{\eta}_n\}_{n \in [N]}$ is an sequence of independent random noise with zero mean and $R-$subgaussian. We use $\hat{\Sigma} = \sum_{n=1}^N \hat{a}_n \hat{a}_n^\top$ to denote the offline Gram matrix.

**Policies and performance metrics.** We consider the design and analysis of policies for a decision maker (DM) that does not know the true $\theta_*$. At the time $t$, the DM proposes the action $a_t \in \mathcal{A}_t$ as an output of a policy function $\pi_t$ that takes all the historical information by time $t-1$ and the current feature $\mathcal{A}_t$ as input arguments. That is,

$$a_t = \pi_t(\{(\hat{a}_n, \hat{x}_n)\}_{n \in [N]}, \{(a_s, x_s)\}_{s \in [t-1]}, \mathcal{A}_t).$$

We denote $\Pi$ as the set of all such policies $\pi = (\pi_1, \pi_2, \dots)$. The set $\Pi$ includes all policies that are feasible for the DM to execute. For any policy $\pi \in \Pi$, the regret of $\pi$, denoted by $R_{\theta_*', \theta_*}^\pi(T)$, is defined as the difference between the optimal expected reward generated by the clairvoyant policy that knows the exact value of $\theta_*$ and the expected reward generated by pricing policy $\pi$, i.e.,

$$R_{\theta_*', \theta_*}^\pi(T) = \mathbb{E}\Big[ \sum_{t=1}^T \max_{a \in \mathcal{A}_t} \langle \theta_*, a \rangle - \sum_{t=1}^T x_t \Big] = \mathbb{E}\Big[ \sum_{t=1}^T \langle \theta_*, a_t^* \rangle - \sum_{t=1}^T x_t \Big],$$

where we define $a_t^* := \arg\max_{a \in \mathcal{A}_t} \theta_*^\top a$. The expectation is taken with respect to the both offline and online random fluctuations $\{\hat{\epsilon}_n\}_{n \in [N]}$ and $\{\epsilon_t\}_{t \in [T]}$.

We assume the learner knows a bias bound $V$ satisfying $V \geq \|\theta_*' - \theta_*\|$ and impose the standard condition that all online and offline parameter vectors—as well as all actions—are uniformly bounded.

Our algorithm $\pi$ constructs at each round $t$ the same confidence set as in Algorithm 2,

$$\mathcal{C}_t = \left\{ \theta \in \mathbb{R}^d : \|\theta - \hat{\theta}_{t,N}\| \leq \hat{w}_{t,N}, \|\theta - \hat{\theta}_t\|_{\Sigma_t} \leq w_t \right\}$$

and chooses the action $a_t$ via the optimistic rule

$$(a_t, \tilde{\theta}_t) = \arg\max_{a \in \mathcal{A}_t} \mathrm{UCB}_t(a) \quad \text{and} \quad \mathrm{UCB}_t(a) = \max_{\theta \in \mathcal{C}_{t-1}} \theta^\top a.$$

By Lemma 12, we have $\theta_* \in \mathcal{C}_t$ with high probability; hence

$$\langle \theta_*, a_t^* \rangle \leq \mathrm{UCB}_t(a_t^*) \leq \mathrm{UCB}_t(a_t).$$

Applying [17, Eq. 19.10], we obtain

$$r_t := \langle \theta_*, a_t^* - a_t \rangle \leq \mathrm{UCB}_t(a_t) - \langle \theta_*, a_t \rangle \leq 2 \min\{\underbrace{w_t(1 \wedge \|a_t\|_{V_{t-1}^{-1}})}_{T_1}, \underbrace{\hat{w}_{t,N}\|a_t\|}_{T_2}\}.$$

Hence, summing over the $T_1$ terms and invoking [17, Corollary 19.3], we obtain

$$R_{\theta_*', \theta_*}^\pi(T) \in \tilde{\mathcal{O}}(d\sqrt{T}).$$

On the other hand, Lemma 5 gives $\hat{w}_{t,N} \in \tilde{\mathcal{O}}(V + \frac{\sqrt{d}}{\sqrt{\lambda_{\min}(\hat{\Sigma})}})$. Summing these $T_2$ terms we obtain

$$R_{\theta_*', \theta_*}^\pi(T) \in \tilde{\mathcal{O}}(VT + \frac{\sqrt{d}T}{\sqrt{\lambda_{\min}(\hat{\Sigma})}}).$$

Therefore, we have

$$R_{\theta_*', \theta_*}^\pi(T) \in \tilde{\mathcal{O}}\big(d\sqrt{T} \wedge (VT + \frac{\sqrt{d}T}{\sqrt{\lambda_{\min}(\hat{\Sigma})}})\big). \tag{35}$$

Some remarks about the result (35) are in order. For general contextual online pricing with biased offline data, we obtain the regret bound $\tilde{\mathcal{O}}\big(d\sqrt{T} \wedge (V^2T + \frac{dT}{\lambda_{\min}(\hat{\Sigma})})\big)$. Note that the factor $(V^2 + \frac{d}{\lambda_{\min}(\hat{\Sigma})})$ is the *square* of $(V + \frac{\sqrt{d}}{\sqrt{\lambda_{\min}(\hat{\Sigma})}})$, in contrast to the stochastic linear bandit. This difference stems from the special structure of the pricing problem, in which the regret can be bounded by the *second* moment $\|\tilde{\theta} - \theta_*\|^2$; see inequality (19). Finally, up to a $\sqrt{d}$ factor, the bound in (35) matches the minimax-optimal regret for the $K$-armed bandit with biased offline data established in [10].

