# OpenReview forum: "Contextual Online Pricing with (Biased) Offline Data"
_NeurIPS.cc/2025/Conference — NeurIPS 2025 poster_

### Official Review · Reviewer_4evj · 2025-06-18

**Clarity:** 1
**Significance:** 2
**Originality:** 1
**Rating:** 3
**Confidence:** 4

**Summary:**

This paper studies the problem of dynamic pricing in the presence of offline data, possibly coming from a different distribution. The model is the following.

The learner has access to $N$ offline observations $(D_n, x_n, y_n, p_n)_{n\leq N}$, where $p_n$ is the price corresponding to the $n$-th (offline) transaction, $x_n$ and $y_n$ are contexts describing this transaction, and $D_n$ is the demand level for this sale at the price $p_n$, given by
$$D_n = x_n^{\top}\alpha' + y_n^{\top}\beta' + \epsilon'_n,$$
where $\epsilon'_n$ is a noise term. Then, at each round $t\leq T$, the learner observes a context $x_t, y_t$, posts a price $p_t$, and receives a reward
$$p_tD_t = p_t(x_t^{\top}\alpha + y_t^{\top}\beta + \epsilon_t).$$

The authors propose a mechanism to leverage the previous observations and mimic the offline strategy to maximize the expected cumulative reward. They derive upper bounds on the regret, depending on problem-dependent constants, on the distance between the offline and online parameters $\alpha', \beta'$ and $\alpha, \beta$, and on the gap between the optimal strategy and the best linear approximation to the offline strategy. They also obtain lower bounds on the regret for this problem and present simulations showcasing their method.

**Questions:**

Adequately addressed by the authors.

**Ethical Concerns:**

["NO or VERY MINOR ethics concerns only"]

**Final Justification:**

I remain unconvinced by the overall contributions of the paper, which appear rather incremental relative to Bu et al. (2020). In addition, the lack of clarity in both the presentation and the proofs significantly reduces the paper’s impact, making it difficult to extract meaningful conclusions, understand the core challenges of the problem, or extend the proposed proof techniques. Therefore, I still recommend rejection of this paper.

**Limitations:**

One of the limitations of this work is that the model considered differs significantly from the classical model of dynamic pricing with covariates, which is not properly discussed in the paper.

**Quality:**

2

**Strengths And Weaknesses:**

**Strengths**

This paper studies the well-motivated problem of leveraging prior informations in dynamic pricing problem. This is, in my opinion, a relevant problem as classical bandit technic rely on a lot of exploration, which is problematic in practise.

**Weaknesses**

- *Clarity*: I find the paper’s clarity to be below the standards for NeurIPS. For example, the ideas behind the algorithm are not presented clearly. Among other issues, the fact that $\widehat{p}$ mimics the pricing strategy $p_n$ instead of estimating the parameter $\theta'$ is not discussed. The authors present their algorithm as addressing an exploration-exploitation trade-off, when in fact the constraints that the price should be upper and lower bounded, combined with the randomness of the context, imply that all prices yield approximately the same amount of information. Thus, the problem reduces to estimating the parameters of a linear regression model from offline and online data and so all rounds correspond to "exploitation rounds". Additionally, the presentation of the theoretical results would benefit from greater clarity; key steps in the proofs are difficult to follow and lack structure.  Moreover, the current version of the paper lacks rigor in certain aspects—for instance, the algorithm does not explicitly define the parameter estimators or the widths of the confidence ellipsoids. These crucial details are only provided in the proof of a lemma in the appendix, which undermines the accessibility and transparency of the method.

- *Rigor*: The authors have addressed my concern regarding an error in the proof of their lower bound.

- *Significance*: The authors study a problem very different from the one classically used in the literature, without justifying it, which may indicate that the contributions are limited. I strongly feel that the discussion of related literature is insufficient. From my (admittedly limited) knowledge of dynamic pricing and dynamic programming, the demand model used in this paper appears to diverge significantly from the classical framework, which typically assumes that each buyer has a private valuation for the good, and demand is binary—equal to 1 if the posted price is below the valuation, and 0 otherwise. This standard model has been extensively studied, and it would be valuable for the authors to more thoroughly engage with this line of work. In particular, it would be helpful to clarify how their setting compares with or generalizes these models. The current treatment, which mainly cites the outdated reference [10]—itself focused primarily on the deterministic case—is not sufficient in my opinion.

- *Originality*: The technical improvement with regards to the previous work [6] in the one dimensional elasticity setting seems incremental at best as the authors study the same model under very similar assumptions, only assuming that the online and offline parameters may be different.In the multi-dimensional elasticity setting, the techniques appear to be standard, and the results are largely expected. Therefore, the technical contributions appear to be of limited significance.

---

> ### Author Rebuttal · Authors · 2025-07-28
>
> We thank the reviewer for recognizing our paper’s contribution and strengths, and for the constructive comments. We provide our detailed responses below.
>
> In what follows, we use [1] [2] etc. to refer to papers cited in our submission, and [a] [b] etc. for new references with bibliographic information given at the end of this response.
>
> **Comment 1: The ideas behind the algorithm are not presented clearly and the algorithm does not explicitly define the parameter estimators or the widths of the confidence ellipsoids in the main paper.**
>
> We thank the reviewer for the suggestion. We do outline the key ideas behind Algorithms 1–3 before presenting their regret bounds in Sections 3 and 4. The intuition for the parameter estimators and confidence‑ellipsoid widths is explained in the main paper; the full mathematical details are deferred to Appendix A due to space constraints. In the revised paper, we will expand the algorithmic details and provide additional discussions to further improve the clarity.
>
> **Comment 2: The fact that $\widehat{p}$ mimics the pricing strategy $p_n$ instead of estimating the parameter $\theta'$ is not discussed.**
>
> We thank the reviewer for this question.  When the offline data are used solely to estimate $\theta_*'$ and only the bias bound $V$ is known, the squared estimation error of the online parameter $\theta_*$ cannot, in order, improve beyond  $V^2+ 1/\lambda_{\min}(\hat \Sigma)$, which yields the worst-case regret bound discussed in the general price-elasticity setting.
>
> In the scalar price-elasticity case—a key contribution of our work—the regret can be further **tightened** through the instance-dependent parameter $\delta^{2}$, which measures how far the empirical pricing rule $\hat p$ departs from the (unknown) optimal policy.  If we understand correctly, the reviewer’s $p_n$ refers to the offline prices $\hat p_n$.  Our $\hat p$ mimics these prices in a **structured** way: it is the best-fit linear rule $\hat p(x,y)=\hat A^\top x/y$ over the offline sample, and, when no context is present, reduces to their empirical mean.  Crucially, $\hat{p}$ is **not** obtained by first estimating $\theta_*'$ and then selecting the corresponding optimal pricing strategy.
>
>
> Then, The effect of $\delta^{2}$ (see Lines 184–206) is two-fold:
>
> 1. If $\delta^{2}$ is **large**, allowing online price strategies $p_t$ to diverge from $\widehat{p}$ accelerates both exploration and exploitation.
> 2. If $\delta^{2}$ is **small**, the online phase should simply adopt $\widehat{p}$.
>
> Thus, $\widehat{p}$ guides the online strategy either to follow it or to deviate from it. Because $\delta^{2}$ is unknown, Algorithm 1 first estimates it (Line 1) and then decides whether it is small enough to justify using $\widehat{p}$ exclusively in the online phase. We will add further intuition about the structure of $\hat{p}$ in the revised paper.
>
> **Comment 3: There may be a mistake in the proof/statement of the lower bound (Theorem 2)... This would imply a total regret that is at most (poly-)logarithmic in $T$. This error was already present in [6] and [28].**
>
> We thank the reviewer for this question. We want to point out that our proof and statement of Theorem 2 is **correct**. First, we would like to emphasize that **dynamic pricing with a linear demand model is not a purely exploitative task.** Prior work \[a] shows that even in the simplest non‑contextual, one‑dimensional setting, $
> D = \alpha + \beta \cdot p + \varepsilon,
> $ the regret is lower‑bounded by $\Omega(\sqrt{T})$.
>
> The key insight is that prices do **not** provide the same amount of information. In dynamic pricing, the instantaneous regret satisfies $$r_t \lesssim (p_t - p^*)^{2} \overset{(1)}{\lesssim} ||\theta_{\*}- \hat{\theta}_t ||^2.$$
>
> On one hand, if $\lVert\theta_*-\hat\theta_t\rVert = O(t^{-1/2})$, achieving an instantaneous regret of $O(t^{-1})$ requires that inequality (1) holds—i.e., $p_t$ must be the optimal price with respect to $\hat\theta_t$, which implies **no**  exploration should be incorporated. On the other hand, achieving the estimation rate $\lVert\hat\theta_t-\theta_*\rVert = O(t^{-1/2})$ demands at least $\Theta(t)$ exploratory rounds, and effective exploration must use **at least two** prices separated by a fixed gap, incurring $\Theta(t)$ regret (as in the ILS‑d algorithm \[a]). Hence, one cannot **simultaneously** maintain a $t^{-1/2}$ estimation error **and** a $t^{-1}$ instantaneous regret for all $1 \leq t \leq T$. Prior work on the estimation–regret trade‑off—see, e.g., \[b]—likewise proves that no algorithm can achieve both the optimal $\sqrt{T}$ regret and the optimal $1/\sqrt{T}$ estimation error within $T$ rounds, highlighting the fundamental exploration–exploitation tension in dynamic pricing with a linear demand model.
>
>
> **Comment 4: The presentation of the theoretical results would benefit from greater clarity; key steps in the proofs are difficult to follow and lack structure.**
>
> We thank the reviewer for the suggestion. In the revised paper, we will expand the discussion accompanying each theoretical result and restructure the proofs.
>
> **Comment 5: The paper tackles a pricing model that deviates from the classical binary‑demand framework without explaining why, raising doubts about the scope of its contributions. The literature review is sparse—relying mainly on the outdated reference \[10]. The authors should clarify how their setting relates to, or generalizes, these binary‑demand framework.**
>
> We thank the reviewer for raising this point. We would like to point out that linear‑demand models are long‑standing classics \[a, e, f] and remain an active research area, as studied by recent work\[2, 4, b, c, d]. Prior work \[e] also demonstrates that many non‑linear demand functions can be closely approximated by a linear specification. In the revision, we will expand our discussion to clarify how our framework relates to the binary‑demand literature and highlight the connections between the two approaches. For a detailed account of our contributions, please see our response to Comment 6.
>
> **Comment 6: The technical improvement with regards to the previous work [6] in the one dimensional elasticity setting seems incremental at best as the authors study the same model under very similar assumptions, only assuming that the online and offline parameters may be different.In the multi-dimensional elasticity setting, the techniques appear to be standard, and the results are largely expected. Therefore, the technical contributions appear to be of limited significance.**
>
> We would like to highlight three principal originalities of this work:
>
> 1. **Instance‑dependent parameter $\delta^{2}$.** We introduce the first instance‑dependent quantity, $\delta^{2}$, for contextual online pricing with offline data—whether biased or unbiased—without making any assumptions about how the offline data are generated. This parameter enables us to derive the first matching instance‑dependent lower and upper regret bounds, which are substantially tighter than existing worst‑case guarantees. The definition of $\delta^{2}$ appears on page 5 (immediately before Section 3.1), and the techniques used to obtain the matching bounds are detailed in Section 3.1 and Appendix C.
>
> 2. **Extension to stochastic linear bandits.** Although this paper centers on contextual online pricing, the techniques and results in Sections 3–4.1 readily extend to stochastic linear bandits with adversarially chosen action sets, as noted in lines 261–263 and detailed in Appendix I.
>
> 3. **Robustness without discrepancy information.** To the best of our knowledge, this is the first work to establish theoretical robustness guarantees for online learning with biased offline data when no information is available about the discrepancy between the online environment and the offline sample (see Section 4.2). Although we focus on contextual pricing, our analysis provides a clear blueprint for extending these results to a wider range of online‑learning problems.
>
>
>
> While we draw inspiration from earlier studies on contextual online pricing with unbiased offline data [6, 28] and K‑armed bandits with biased offline data [9], the proofs required to achieve the three principal originalities above demand several novel techniques.
>
>
> References:
>
> [a] Keskin, N Bora and Zeevi, Assaf. Dynamic pricing with an unknown demand model: Asymptotically optimal semi-myopic policies. 2014.
>
> [b] Simchi-Levi, David and Wang, Chonghuan. Pricing experimental design: causal effect, expected revenue and tail risk. 2023.
>
> [c] Li, Xiaocheng and Zheng, Zeyu. Dynamic pricing with external information and inventory constraint. 2024.
>
> [d] Ao, Ruicheng and Jiang, Jiashuo and Simchi-Levi, David. Learning to Price with Resource Constraints: From Full Information to Machine-Learned Prices. 2025
>
> [e] Besbes, Omar and Zeevi, Assaf. On the (surprising) sufficiency of linear models for dynamic pricing with demand learning. 2015.
>
> [f] Qiang, Sheng and Bayati, Mohsen. Dynamic pricing with demand covariates. 2016

---

> ### Comment · Reviewer_4evj · 2025-08-04
> **Response to comment 3**
>
> I remain unconvinced by the authors’ response to Comment 3 and will maintain my score unless they provide a substantially more compelling rebuttal.
>
> Without previous observations, the estimation error $\Vert \theta - \hat{\theta} \Vert $ decreases as (up to log factors and multiplicative constants) the square root of the inverse of the minimum eigenvalue of the design matrix $\Sigma_t$ (see, e.g., the proof of Lemma 5). Here, we have $$\Sigma_t  = \sum_{s\leq t} z_s z_s^{\top}  = \sum_{s\leq t} z'_s z'_s^{\top} +  \sum_{s\leq t} z''_s z''_s^{\top} \geq \sum_{s\leq t} z'_s z'_s^{\top}$$
> where $z_s = (x_s^{\top}, p_s y_s^{\top})^{\top}$, $z'_s = (x_s^{\top}, l y_s^{\top})^{\top}$, and $z''_s = (0, (p_s - l) y_s^{\top})^{\top}$ (here the inequality should be understood in the sense $A \geq B$ if $A - B$ is semi-definite non-negative, and it holds because $p_s \geq l$). Thus, the estimation error $\Vert \theta - \hat{\theta}_t \Vert $ decreases faster than (up to log factors and multiplicative constants) the square root of the inverse of the minimum eigenvalue of the design matrix $ \geq \sum_{s\leq t} z'_s z'_s^{\top}$, thus decreasing faster than (up to log factors and numerical constants) $1/\sqrt{t}$ (because of the assumption that $x_t$ and $y_t$ are i.i.d. with positive covariance matrices).
>
> Thus, contrary to the author's claim, no matter the strategy (as long as it posts price above $l$), the estimation error $\Vert \theta - \hat{\theta}_t \Vert $ is of order $1/\sqrt{t}$ (here again forgetting multiplicative constants and log factors).
>
> I strongly recommend that the authors address this point using rigorous mathematical proofs rather than informal intuition—for instance, by explicitly explaining the logical connection between the two equations immediately following Step 1.1. I am aware that this observation implies the lower bound in [b] is incorrect.

---

> > ### Author Response · Authors · 2025-08-05
> >
> > We thank the reviewer for the follow-up comments and the detailed explanation of why you believe the regret could be $O(\log T)$. Unfortunately, two key steps in that argument are **invalid**.
> >
> > 1. The claim
> > $$
> > \sum_{s \le t} z_s z_s^{\top}\succeq\sum_{s \le t} z'_s {z'_s}^{\top}
> > $$
> > does not hold in general.  A single-round counter-example suffices: Take $t=1$ and choose $x=1,y=1,l=1,p=2$.
> > Then, $zz^{\top} - z'z'^{\top}
> > =\begin{bmatrix} 0 & 1 \\\ 1 & 3 \end{bmatrix}$, whose determinant is $-1<0$; Hence, the matrix is not even positive-semidefinite.
> >
> >
> > 2.  Even if one had $
> > \sum_{s \le t} z_s z_s^{\top}\succeq\sum_{s \le t} z'_s {z'_s}^{\top}
> > $, the matrix
> >
> > $$\sum\_{s\le t} z'_s {z'_s}^{\top}$$
> >
> > can still have a **zero** minimum eigenvalue almost surely for every $t$.  Our Assumption 1 states only that $\{(x_t,y_t)\}_{t\ge1}$ are i.i.d. and that $\mathbb E[x_t x_t^{\top}]$ and $\mathbb E[y_t y_t^{\top}]$ are positive-definite; However, it does **not** assume $x_t$ and $y_t$ are independent.
> >
> > * Suppose a non-zero vector $(a,b)$ satisfies $x_t^{\top}a + y_t^{\top}b \equiv 0$ a.s.
> >   Setting $v\coloneqq(a,b/l)$, one has
> >   $$
> >   v^{\top}\Bigl(\sum_{s\le t} z'_s {z'_s}^{\top}\Bigr)v = 0,
> >   $$
> >
> > which means the minimum eigenvalue is a.s. zero for every $t$. In other words, the desired $\lambda_{\min}\bigl(\sum_{s\le t} z'_s z'^{\top}_s\bigr)\gtrsim\sqrt{t}$ bound would not hold.
> >
> > * A concrete instance is the one-dimensional, context-free case with $x_t=y_t=1$ (i.e., the demand model is $D=\alpha + \beta p + \varepsilon$).  Choosing $v=(1,-1/l)$ yields the same degeneracy.
> > Simply put, because the correlation structure between $x_t$ and $y_t$ is unknown, **a single price is insufficient for exploration**. Additionally, we would like to highlight that when the demand is $D=\alpha + \beta p + \varepsilon$, Theorem 1 of [a] already establishes a $\sqrt{T}$ lower  bound.
> >
> > We will also emphasize that the two equations immediately following Step 1.1 in our draft are **correct**:
> >
> > * **First equality**  $\mathbb{E}[C(w)^{\top}I_t(H_t)  C(w)] =\sum_{j = 1}^{t-1}\mathbb{E}[(\alpha_*^\top x_j + 2\beta_* y_j p_j)^2]$:
> >
> > A direct substitution of the definitions of $C(w)=(0,0,\alpha_*,2\beta_*)$ and
> >
> > $$I_t(H_t)=\left[\begin{array}{cccc}
> > \sum_{i=1}^N \hat{x}_i\hat{x}_i^\top &  \sum\_{i=1}^N \hat{x}_i\hat{p}_i\hat{y}_i & 0 & 0\\\\
> > \sum\_{i=1}^N \hat{p}_i\hat{y}_i\hat{x}_i^\top  &  \sum\_{i=1}^N \hat{p}_i^2\hat{y}_i^2 & 0 & 0\\\\
> > 0 &  0 & \sum\_{i=1}^{t-1} x_ix_i^\top & \sum\_{i=1}^{t-1} x_ip_iy_i  \\\\
> > 0 &  0 & \sum\_{i=1}^{t-1} p_iy_ix_i^\top   & \sum\_{i=1}^{t-1} p_i^2y_i^2
> > \end{array}\right]$$
> >
> > * **Second inequality** $\sum_{j = 1}^{t-1}\mathbb{E}[(\alpha_*^\top x_j + 2\beta_* y_j p_j)^2] \leq 4u_\beta^2\sum_{j = 1}^{t-1}\mathbb{E}[(p_j - p^*_{\theta_\*}(x_j,y_j))^2]$:
> >
> > After multiplying and dividing  $(2\beta_* y_j)^2$ to each summand in the LHS, we have $\sum_{j = 1}^{t-1}\mathbb{E}[(\alpha_*^\top x_j + 2\beta_* y_j p_j)^2] = \sum_{j = 1}^{t-1}\mathbb{E}[(2\beta_* y_j)^2(p_j + \frac{\alpha_*^\top x_j}{2\beta_* y_j})^2]$.
> >
> > Under Assumption 1, $$
> >      p_{\theta_*}^{\*}(x_j,y_j)= -\frac{\alpha_*^{\top}x_j}{2\beta_* y_j},
> >    \qquad
> >      (2\beta_* y_j)^2 \le 4u_\beta^2,
> >    $$ which yields the claimed bound.
> >
> > References:
> >
> > [a] Keskin, N Bora and Zeevi, Assaf. Dynamic pricing with an unknown demand model: Asymptotically optimal semi-myopic policies. 2014.

---

> > > ### Comment · Reviewer_4evj · 2025-08-05
> > > **Response acknowledgment**
> > >
> > > I thank the authors for their response, which adresses my concerns regarding the correctness of the proof of the lower bound.

---

### Official Review · Reviewer_oYic · 2025-06-20

**Clarity:** 2
**Significance:** 3
**Originality:** 3
**Rating:** 3
**Confidence:** 2

**Summary:**

In this paper, the authors studied the problem of contextual online pricing with offline data which can be biased. Existing work showed that when offline data are unbiased in the sense of coming from the same distribution as the online data, incorporating the offline data to online policy can outperform pure online learning techniques. However, distribution shifts may exist, and it is common for the offline data to be different from the online data. As a result, it would be desirable to study the effect of potentially biased offline data for online learning. In this paper, the authors proposed an Optimism-in-the-Face-of-Uncertainty policy which is shown to be minimax-optimal. Both the scalar price elasticity and the general setting have been considered. Both known and unknown bias bounds have been investigated as well.

**Questions:**

1.	In the case where V is assumed to be known, how sensitive is the algorithm to inaccuracies in V? How strongly does it rely on the precise specification of V?
2.	For the regret bounds for the general case, why N does not appear in the bounds?
3.	Although linear demand model was previously used in the literature and is the focus for this paper, it is unlikely to hold in practice. Some discussion on nonlinear models and the corresponding implications would be beneficial.
4.	For RCO3 in the numerical study, it appears that the benefit is relatively small for small V yet the damage on regret can be very substantial for larger V. Overall, despite the authors claim robustness, it seems safer to use UCB instead.

**Ethical Concerns:**

["NO or VERY MINOR ethics concerns only"]

**Final Justification:**

The authors have addressed most of my comments in the rebuttal. However, my concern regarding the linear demand model was not adequately addressed. I am maintaining my current rating of 3.

**Limitations:**

The current paper does not have clear discussions on the limitations and potential negative societal impact of their work. More discussions on the feasibility of known V and how to examine the usability of offline data are necessary. The paper does not study real applications.

**Quality:**

2

**Strengths And Weaknesses:**

Online pricing with the offline data is an interesting and worthwhile problem to study. This paper investigates the potential data shifts in terms of biased offline data compared with online data. The regret bounds shed some lights on the impact of offline data for online learning with respect to various factors such as bias bound etc.
Although theoretically insightful, the majority of the paper considers the case of known V. However, in practice, V is likely unknown. Despite there are some developments for the case of unknown V, it is important and necessary for authors to add discussions and explanations on when V can be known in practice. Also, the numerical studies are somewhat limited to two simple simulated settings.

---

> ### Author Rebuttal · Authors · 2025-07-28
>
> We thank the reviewer for recognizing our paper’s contribution and strengths, and for the constructive comments. We provide our detailed responses below.
>
> In what follows, we use [1] [2] etc. to refer to papers cited in our submission, and [a] [b] etc. for new references with bibliographic information given at the end of this response.
>
> **Comment 1: Add discussions and explanations on when V can be known in practice**
>
> We thank the reviewer for raising this point. We emphasize that $V$ is only a **valid upper bound** on the bias $\lVert\theta_*-\theta_*'\rVert$; specifically, $V \ge \lVert\theta_*-\theta_*'\rVert$. It is **not** intended as an exact estimate, i.e., we do not assume $V = \lVert\theta_*-\theta_*'\rVert$. In practice, $V$ can be estimated with robust machine‑learning methods \[5] or through cross‑validation \[8], as also argued in prior work on K‑armed bandits with biased offline data \[9]. Setting $V=\infty$ trivially guarantees validity and recovers the same performance as classical algorithms that ignore offline data (see Sections 3–4.1). However, when a finite upper bound that scales favourably with the time horizon $T$ is available, the regret guarantees in Sections 3–4.1 become sharper. We will expand the discussion that estimating a valid upper bound $V$ is practically feasible.
>
> **Comment 2: In the case where V is assumed to be known, how sensitive is the algorithm to inaccuracies in V? How strongly does it rely on the precise specification of V?**
>
> We thank the reviewer for the question. As explained in Comment 1, our results in Sections 3–4.1 require only that $V$ be a **valid upper bound** on the true bias $\lVert\theta_*-\theta_*'\rVert$; such a upper bound can be estimated with robust ML methods or cross‑validation. Theorems 1 and 3 show that regret depends on $V$ and the dependence is solely through how $V$ scales with the time horizon $T$. For example, if $\lVert\theta_*-\theta_*'\rVert = T^{-5/16}$ and we use the upper bound $V = 1.1\lVert\theta_*-\theta_*'\rVert>\lVert\theta_*-\theta_*'\rVert$, then $V$ still scales as $T^{-5/16}$, and our theorems guarantee regret is strictly better than classical algorithms that ignore offline data. Numerical experiments corroborate this: with a tight estimate ($V = 1.1\lVert\theta_*-\theta_*'\rVert$) we achieve lower regret; and even with a loose estimate ($V = 10\lVert\theta_*-\theta_*'\rVert$) the regret does not exceed that of a purely online baseline.
>
> We acknowledge that robust ML methods or cross‑validation can, with small probability, produce an *invalid* estimate of $V$ (i.e., $V < \lVert\theta_*-\theta_*'\rVert$). In that event, our confidence intervals no longer hold and the regret can become linear in theory. From a practical standpoint, however, it is still valuable to understand how regret behaves when $V$ is underestimated. Accordingly, our revised paper will include additional experiments that test the algorithm’s performance under such misspecifications.
>
> **Comment 3: Although linear demand model was previously used in the literature and is the focus for this paper, it is unlikely to hold in practice. Some discussion on nonlinear models and the corresponding implications would be beneficial.**
>
> We thank the reviewer for raising this point. Our linear‑model framework extends naturally to a generalized linear model with a known link function $g$: $D=g\bigl(\alpha^{\top}x + (\beta^{\top}y)p\bigr) + \varepsilon,$ provided that (i) $g$ is differentiable and strictly increasing, and (ii) the minimizer of $
> \bar D(p)=g\bigl(\alpha^{\top}x+ (\beta^{\top}y)p\bigr)
> $ lies in the interior of the feasible price interval $[l,u]$. These assumptions are standard; see [2,b].
> The only change required is to replace the least‑squares estimate with a maximum quasi‑likelihood objective. Because $g$ is differentiable and strictly increasing, it is locally linear, we are confident that our core regret analysis framework can be extended to this setting and achieve comparable results; see [c] for a similar argument and [2] for the full derivation. We will add the discussion in the revised manuscript.
>
> **Comment 4: For RCO3 in the numerical study, it appears that the benefit is relatively small for small V yet the damage on regret can be very substantial for larger V. Overall, despite the authors claim robustness, it seems safer to use UCB instead.**
>
> We appreciate the reviewer’s observation. We do **not** claim that RCO3 always outperforms UCB. In fact, our impossibility result shows that, without any knowledge of the mismatch between offline and online data distributions, no algorithm can always outperform a purely online baseline such as UCB.
>
> The robustness section therefore has a different goal: it introduces an algorithm that, *when an upper bias bound $V$ is unavailable*, (i) achieves lower regret than a pure‑online method when the true bias is small, and (ii) still guarantees sublinear regret when the bias is large. Empirically, Figure 2(c) shows that the percentage drop in regret for small bias is comparable to the percentage increase for large bias, even though the absolute loss in the latter case is higher. Theoretically, these robustness results complement Sections 3–4.1—which assume $V$ is known—by providing the first robustness guarantee for contextual online pricing with biased offline data and paving the way for analogous guarantees in broader online‑learning settings that leverage such data.
>
>
> **Comment 5: The current paper does not have clear discussions on the limitations and potential negative societal impact of their work. More discussions on the feasibility of known V and how to examine the usability of offline data are necessary.**
>
> We appreciate the reviewer’s suggestion. For additional discussion on how a valid upper bound $V$ can be obtained, please see our responses to Comments 1 and 2.
>
> **Usability of offline data**
>
> * **If a valid bound $V$ is available:** We should always incorporate the offline data and use Algorithms 1 or 2. As indicated by Theorems 1 and 3, these algorithms never perform worse than a pure‑online baseline and can outperform it when $V$ is small and the offline data are informative (Sections 3–4.1).
>
> * **If no valid bound $V$ is available:** We can adopt Algorithm 3, selecting the length of the testing phase within the choice set in Theorem 5, according to the decision maker’s risk tolerance. A shorter test phase lowers regret when the true bias is small but increases the worst‑case regret, and vice versa. A special case is $\alpha = 1/2$, where the decision maker does not accept regret exceeding $\sqrt{T}$; in this scenario, Algorithm 3 matches the performance of a purely online algorithm.
>
> We will clarify these guidelines and expand the discussion in the revised manuscript.
>
> For discussions on potential negative societal impact, our work is primarily theoretical and focuses on the design and analysis of an online learning algorithm for pricing with biased offline data, with the goal of achieving low regret. As such, we abstract away from specific application domains, and do not make assumptions about user demographics or other sensitive attributes. We also want to emphasize that our algorithm does not rely on access to sensitive features.
>
> That said, we agree that pricing mechanisms, even when designed with performance guarantees, can have real-world implications on fairness and equity. In future work, it would be valuable to study how such algorithms perform in settings where users belong to different groups, and to examine whether the algorithm's decisions introduce or exacerbate disparities across these groups. We will add a discussion in the revision to acknowledge these concerns and point out directions for fairness-aware extensions of our work.
>
> **Comment 6: The paper does not study real applications.**
>
> We thank the reviewer for raising this point. We agree that incorporating real-world data is an important next step. The *CPRM-12-001: On-Line Auto Lending* dataset from Columbia University’s Center for Pricing and Revenue Management—widely used in personalized lending-rate studies \[2, d]—is a natural choice for our online component. We will perform the experiment with  this (online) dataset and synthetic (offline) data generated by large language models (LLMs), which can closely approximate real-world behaviors.
>
> We also would like to emphasize that the primary contribution of this paper is a comprehensive theoretical understanding of contextual online pricing with biased offline data. For a detailed summary of these theoretical contributions and their originality, please refer to our response to Reviewer 4jmU’s Comment 2.
>
> References:
>
> [a] Lei, Yanzhe and Miao, Sentao and Momot, Ruslan. Privacy-preserving personalized revenue management. 2024.
>
> [b] Den Boer, Arnoud V and Zwart, Bert. Simultaneously learning and optimizing using controlled variance pricing. 2014.
>
> [c] Keskin, N Bora and Zeevi, Assaf. Dynamic pricing with an unknown demand model: Asymptotically optimal semi-myopic policies. 2014.

---

> > ### Comment · Reviewer_oYic · 2025-08-02
> >
> > Thank you for your responses in addressing my concerns. Including clarifications of these issues and real data applications in the revision will help strengthen the paper. Regarding my concern on linear demand model, the authors mentioned the extension to a generalized linear model with a known link function with g. I trust the results can be extended. However, the assumption of a known link function doesn’t really address general nonlinear models.

---

> ### Author Response · Authors · 2025-08-05
>
> We appreciate the reviewer’s follow-up comments and agree that broadening our discussion of nonlinear demand models will strengthen the paper. Notably, prior work [a] shows that—under a suitable regularity condition on the unknown nonlinear link function (Assumption 1 in [a])—a semi-myopic policy based on a linear model converges to the optimal policy for the true nonlinear model and achieves minimax-optimal regret. This finding further underscores the importance of studying linear models in online pricing. We will incorporate these insights and their implications into the revised manuscript.
>
>
>
> References:
>
> [a] Besbes, Omar and Zeevi, Assaf. On the (surprising) sufficiency of linear models for dynamic pricing with demand learning. 2015.

---

### Official Review · Reviewer_TipE · 2025-07-02

**Clarity:** 1
**Significance:** 3
**Originality:** 2
**Rating:** 4
**Confidence:** 2

**Summary:**

This paper studies contextual online pricing where a firm must set prices over time while learning customer demand parameters from observed sales. In addition to online data, the firm has access to offline data that may be biased, meaning it comes from a different distribution than the online environment. The authors propose CO3 algorithm for scalar price elasticity, GCO3 algorithm for general elasticity, and RCO3 algorithm for settings where the bias level is unknown. They derive instance-dependent and minimax-optimal regret bounds that explicitly capture the effects of offline data bias, offline data dispersion, and online learning horizon. The paper also establishes matching lower bounds and impossibility results, showing when and how offline data can improve over purely online approaches. The theoretical results are supported by simulation experiments illustrating the performance of the proposed algorithms relative to standard baselines.

**Questions:**

1) Could the authors clarify the computational cost of their algorithms?

2) Could the authors provide guidance on choosing the length of the test phase in RCO3 and its sensitivity to this choice?

3) The paper assumes i.i.d. contexts over time. Could the authors comment on how their regret guarantees might extend or degrade under temporal dependence or adversarially chosen contexts?

**Ethical Concerns:**

["NO or VERY MINOR ethics concerns only"]

**Final Justification:**

After reviewing the authors’ rebuttal and comparing their work with Bu et al. (2020), I find that the paper offers meaningful technical novelty. While some assumptions and parameters could be better motivated and the presentation could more clearly highlight the significance of the contributions, I keep my score at 4

**Limitations:**

The paper does not address potential negative societal impacts, such as how biased offline data or aggressive pricing strategies could unintentionally amplify unfair pricing or discrimination.

**Paper Formatting Concerns:**

No formatting concerns.

**Quality:**

2

**Strengths And Weaknesses:**

Strengths:

1) The paper provides a rigorous theoretical framework for contextual online pricing with biased offline data.

2) The formulation of the Contextual Bandits for Online Pricing with Offline Data problem and the proposed algorithms are novel.

3) The paper is clearly written and well organized.

4) The paper addresses a practically important problem.

Weaknesses:

1) The paper does not discuss computational aspects of the proposed algorithms.

2) Proposed methods assume the linear demand model is correctly specified, there is no analysis or experimental evaluation of robustness to model misspecification.

3) Proposed methods are not tested on real-world datasets or realistic pricing simulations, where context distributions, noise properties, or demand patterns may not fully match the assumptions.

4) The authors do not explicitly discuss limitations related to real-world deployment.

---

> ### Author Rebuttal · Authors · 2025-07-28
>
> We thank the reviewer for recognizing our paper’s contribution and strengths, and for the constructive comments. We provide our detailed responses below.
>
> In what follows, we use [1] [2] etc. to refer to papers cited in our submission, and [a] [b] etc. for new references with bibliographic information given at the end of this response.
>
> **Comment 1: The paper does not discuss computational aspects of the proposed algorithms.**
>
> We appreciate the reviewer’s question. The computational complexity of our algorithms indeed merits additional discussion. In Algorithms 1 and 2, we update the policy and parameter estimates by repeatedly solving a quadratically constrained quadratic program (QCQP) (see Line 7 of Algorithm 1). Although efficient QCQP solvers exist \[g], we still incur $T$ oracle calls over a horizon of length $T$.
>
> As noted by \[f], fully adaptive, per-round updates hinder parallelism and limit large-scale deployment. In purely online settings, recent work on $K$-armed bandits \[a], linear bandits \[f], and generalized linear bandits \[e] shows that optimal $O(\sqrt{T})$ regret can be achieved with only $O(\log\log T)$ policy updates. Adapting these ideas to our setting—developing regret-optimal methods that require far fewer QCQP calls—is a promising research direction. We will expand on this avenue in the revised manuscript.
>
> **Comment 2: Proposed methods assume the linear demand model is correctly specified, there is no analysis or experimental evaluation of robustness to model misspecification.**
>
> We thank the reviewer for this suggestion. Developing algorithms that are robust to model misspecification—for example, those explored in [b, c]—is a popular research direction. We note, however, that even with unbiased offline data, the effects of model misspecification are not yet fully understood. A promising avenue for future work is to establish theoretical guarantees for online pricing when both unbiased offline data and an unknown (potentially misspecified) demand model are present. Investigation of this problem requires new ideas for algorithm design and theoretical analysis, which is beyond the scope of the current paper. We will leave it  as future work and expand this discussion in the revised manuscript.
>
> **Comment 3: Proposed methods are not tested on real-world datasets or realistic pricing simulations, where context distributions, noise properties, or demand patterns may not fully match the assumptions. The authors do not explicitly discuss limitations related to real-world deployment.**
>
> We thank the reviewer for raising this point. We agree that incorporating real-world data is an important next step. The *CPRM-12-001: On-Line Auto Lending* dataset from Columbia University’s Center for Pricing and Revenue Management—widely used in personalized lending-rate studies \[2, d]—is a natural choice for our online component. We will perform the experiment with  this (online) dataset and synthetic (offline) data generated by large language models (LLMs), which can closely approximate real-world behaviors.
>
> We also wish to emphasize that the primary contribution of this paper is a comprehensive theoretical understanding of contextual online pricing with biased offline data. The assumptions on context distributions, noise, and demand patterns are standard in the literature [2] for deriving theoretical results of online pricing, and our simulations serve to validate the theory.
>
> **Comment 4: Could the authors provide guidance on choosing the length of the test phase in RCO3 and its sensitivity to this choice?**
>
> We thank the reviewer for the question. As Theorem 5 and its discussion show, the test‑phase length is not uniquely optimal; any value within a certain range is admissible. Within this range, a shorter test phase lowers regret when the true bias is small but increases the worst‑case regret. Therefore, the choice ultimately depends on the decision maker’s risk tolerance—balancing a higher worst‑case regret against the potential for better performance when the bias is small. A special case is $\alpha = 1/2$, where the decision maker does not accept regret exceeding $\sqrt{T}$; in this scenario, Algorithm 3 matches the performance of a purely online algorithm. We will expand the manuscript to clarify this trade‑off.
>
> **Comment 5: The paper assumes i.i.d. contexts over time. Could the authors comment on how their regret guarantees might extend or degrade under temporal dependence or adversarially chosen contexts?**
>
> We thank the reviewer for the question. In the scalar price‑elasticity case we assume i.i.d. contexts, which allows us to define the instance‑dependent parameter $\delta^{2}$ and derive the corresponding instance‑dependent regret bound. For worst‑case regret bounds, however, this assumption is unnecessary; our results still hold when contexts are chosen adversarially. Lines 261–263 and Appendix I already show how our techniques extend to stochastic linear bandits with adversarially selected action sets. We will clarify the role of the i.i.d. assumption in the revised manuscript.
>
>
> **Comment 6: The paper does not address potential negative societal impacts, such as how biased offline data or aggressive pricing strategies could unintentionally amplify unfair pricing or discrimination.**
>
> We thank the reviewer for raising this important point. Our work is primarily theoretical and focuses on the design and analysis of an online learning algorithm for pricing with biased offline data, with the goal of achieving low regret. As such, we abstract away from specific application domains, and do not make assumptions about user demographics or other sensitive attributes. We also want to emphasize that our algorithm does not rely on access to sensitive features.
>
> That said, we agree that pricing mechanisms, even when designed with performance guarantees, can have real-world implications on fairness and equity. In future work, it would be valuable to study how such algorithms perform in settings where users belong to different groups, and to examine whether the algorithm's decisions introduce or exacerbate disparities across these groups. We will add a discussion in the revision to acknowledge these concerns and point out directions for fairness-aware extensions of our work.
>
> References:
>
> [a] Perchet, Vianney and Rigollet, Philippe and Chassang, Sylvain and Snowberg, Erik. Batched bandit problems. 2016
>
> [b] Besbes, Omar and Zeevi, Assaf. On the (surprising) sufficiency of linear models for dynamic pricing with demand learning. 2015.
>
> [c] Nambiar, Mila and Simchi-Levi, David and Wang, He. Dynamic learning and pricing with model misspecification. 2019.
>
> [d] Lei, Yanzhe and Miao, Sentao and Momot, Ruslan. Privacy-preserving personalized revenue management. 2024.
>
> [e] Sawarni, Ayush and Das, Nirjhar and Barman, Siddharth and Sinha, Gaurav. Generalized linear bandits with limited adaptivity. 2024.
>
> [f] Ruan, Yufei and Yang, Jiaqi and Zhou, Yuan. Linear bandits with limited adaptivity and learning distributional optimal design. 2021
>
> [g] Nesterov, Yurii and Nemirovskii, Arkadii. Interior-point polynomial algorithms in convex programming. 1994.

---

> > ### Comment · Reviewer_TipE · 2025-08-06
> >
> > Thank you for your rebuttal. I have one additional clarification request: could the authors elaborate on what is technically new in the algorithmic design or proof techniques relative to Bu et al. (2020) in the non-contextual setting? From the current presentation, it appears that the main difference is the extension to the contextual setting without the fixed-policy assumption of [28]. It would be helpful to understand whether there are additional conceptual or technical innovations beyond this generalization.

---

> ### Author Response · Authors · 2025-08-06
>
> We appreciate the reviewer’s follow-up comments. We would like to highlight four main originalities of this work:
>
> 1. **Instance‑dependent parameter $\delta^2$.** We introduce the first instance‑dependent quantity $\delta^2$, for contextual online pricing with offline data—whether biased or unbiased—without making any assumptions about how the offline data are generated. This parameter enables us to derive the first matching instance‑dependent lower and upper regret bounds, which are substantially tighter than existing worst‑case guarantees.
>
> 2. **Biased offline data.** We emphasize that the bias bound $V$ introduces substantial challenges in deriving tight upper and lower bounds—especially when our goal is to obtain instance-dependent results. In particular, correctly figuring out the respective roles of $\delta^{2}$ and $V$ adds an extra layer of difficulty to the analysis.
>
> Regarding the first two points, we emphasize that the paper introduces both a novel algorithmic design and new proof techniques.
> * **Algorithmic design.** We develop a three-ellipsoid confidence-interval construction, coupled with a carefully chosen test period, to obtain a tight upper bound.
> * **Proof techniques.** The above two innovations create substantial challenges in deriving matching upper and lower bounds, far beyond those encountered in the analyses of Bu et al. (2020) and [28].
>
> We would also like to emphasize some completely novel results of this paper:
>
> 3. **Extension to stochastic linear bandits.** Although this paper centers on contextual online pricing, the techniques and results in Sections 3–4.1 readily extend to stochastic linear bandits with **adversarially chosen action sets**, as noted in lines 261–263 and detailed in Appendix I.
>
> 4. **Robustness without discrepancy information.** To the best of our knowledge, this is the first work to establish theoretical robustness guarantees for online learning with biased offline data when no information is available about the discrepancy between the online environment and the offline sample (see Section 4.2). Although we focus on contextual pricing, our analysis provides a clear blueprint for extending these results to a wider range of online‑learning problems.
>
> We will state our contributions more clearly in the revised manuscript.

---

### Official Review · Reviewer_4jmU · 2025-07-03

**Clarity:** 3
**Significance:** 2
**Originality:** 2
**Rating:** 5
**Confidence:** 2

**Summary:**

The work focuses on online pricing in the presence of biased data retrieved from previous offline interactions (not made by a specific algorithm). The authors introduce the setting and the assumptions, and present 3 algorithms to learn online in this setting, exploiting also offline data. One algorithm is for scalar price elasticity, while the others are for general price elasticity. The authors study the regret guarantees of the solutions and provide an experimental validation of their solutions.

**Questions:**

See weaknesses (and minors).

**Ethical Concerns:**

["NO or VERY MINOR ethics concerns only"]

**Final Justification:**

I believe that this is a good work overall and I left my overall judge unchanged. However, I was not aware of the work by Bu et al. (2020), which may reduce the impact of the work, as highlighted by the other Reviewers.

**Limitations:**

Yes. The limitations of the work are properly discussed.

**Paper Formatting Concerns:**

No formatting concerns.

**Quality:**

3

**Strengths And Weaknesses:**

### Strengths
- The paper is well-written and structured. It is visible that the authors invested time in the writing phase to properly present their findings in the limited space available.
- The authors, at every step, discuss what they are presenting, comparing it with the literature.
- The theoretical analysis is formal and well-performed, and all the results are reasonable given the reviewer's knowledge of the related literature. I haven't checked carefully the theoretical analysis, but it seems to be complete.

### Weaknesses
I haven't seen any critical points for this work. However, I have some weaknesses to highlight.
- The authors present all the assumptions in a unique "block" (Assumption 1). This assumption should be better discussed to allow the user to understand it properly.
- The originality of the work is limited.
- The experimental validation is limited and presented in a very concentrated way. However, given the theoretical character of this work, this experimental validation is sufficient.

### Minors

- References are often imprecise. For example:
    [9]  -> ICML,
    [11] -> ICLR,
    [12] -> ECB,
    [23] -> Single author,
    [29] -> ICML.
- In order to understand the algorithms, the reader has to rely heavily on the pseudocode. The presentation of the algorithms, in my opinion, should be expanded in the camera-ready exploiting the additional page.

---

> ### Author Rebuttal · Authors · 2025-07-27
>
> We thank the reviewer for recognizing our paper’s contribution and strengths, and for the constructive comments. We provide our detailed responses below.
>
> **Comment 1: The authors present all the assumptions in a unique "block" (Assumption 1). This assumption should be better discussed to allow the user to understand it properly.**
>
> We thank the reviewer for this helpful suggestion. Assumption 1 collects the standard conditions used in contextual online pricing (see, e.g., \[2, 4, 16]). Because of space constraints, we grouped them into a single block. In the revised paper, we will expand the accompanying discussion to provide additional context for Assumption 1.
>
> **Comment 2: The originality of the work is limited.**
>
> We would like to highlight three main originalities of this work:
>
> 1. **Instance‑dependent parameter $\delta^{2}$.** We introduce the first instance‑dependent quantity, $\delta^{2}$, for contextual online pricing with offline data—whether biased or unbiased—without making any assumptions about how the offline data are generated. This parameter enables us to derive the first matching instance‑dependent lower and upper regret bounds, which are substantially tighter than existing worst‑case guarantees. The definition of $\delta^{2}$ appears on page 5 (immediately before Section 3.1), and the techniques used to obtain the matching bounds are detailed in Section 3.1 and Appendix C.
>
> 2. **Extension to stochastic linear bandits.** Although this paper centers on contextual online pricing, the techniques and results in Sections 3–4.1 readily extend to stochastic linear bandits with adversarially chosen action sets, as noted in lines 261–263 and detailed in Appendix I.
>
> 3. **Robustness without discrepancy information.** To the best of our knowledge, this is the first work to establish theoretical robustness guarantees for online learning with biased offline data when no information is available about the discrepancy between the online environment and the offline sample (see Section 4.2). Although we focus on contextual pricing, our analysis provides a clear blueprint for extending these results to a wider range of online‑learning problems.
>
>
>
> While we draw inspiration from earlier studies on contextual online pricing with unbiased offline data [6, 28] and K‑armed bandits with biased offline data [9], the proofs required to achieve the three main originalities above demand several novel techniques.
>
> **Comment 3: The experimental validation is limited and presented in a very concentrated way. However, given the theoretical character of this work, this experimental validation is sufficient.**
>
> We appreciate the reviewer’s recognition of our theoretical contribution and agree that broader empirical evidence would further strengthen the paper. In the revision, we will extend the numerical study by (i) exploring a wider range of online‑offline scenarios and (ii) examining the sensitivity of our results to the bias bound $V$.
>
> **Comment 4: References are often imprecise. For example: [9] -> ICML, [11] -> ICLR, [12] -> ECB, [23] -> Single author, [29] -> ICML.**
>
> We thank the reviewer for raising this point. We will correct the imprecise references in the revised manuscript.
>
> **Comment 5: In order to understand the algorithms, the reader has to rely heavily on the pseudocode. The presentation of the algorithms, in my opinion, should be expanded in the camera-ready exploiting the additional page.**
>
> We thank the reviewer for the excellent suggestion. We will provide additional details and discussions about our algorithms in the revised paper.

---

> > ### Comment · Reviewer_4jmU · 2025-08-08
> >
> > Thanks for the response, I have no further points to discuss.

---

### Note · Authors · 2025-08-15

We thank all reviewers for their thoughtful feedback and careful engagement, and we thank the area chair for overseeing the process. In light of the reviews, we respectfully emphasize the following points.

**Correctness of the lower-bound proof.**
We have addressed Reviewer’s concerns and clarified the argument; the lower bound is correct.

**Significance of the linear demand model.**
Linear demand is both classical [b,c] and actively studied, with many recent papers advancing this line of work [d,e]. Notably, [a] show that—under a suitable regularity condition on the unknown nonlinear link function (Assumption 1 in [a])—a semi-myopic policy based on a linear model converges to the optimal policy for the true nonlinear model and attains minimax-optimal regret. This further underscores the importance of studying linear models in online pricing.

**Key contributions.**
This paper opens the door for incorporating biased offline data into contextual online learning for pricing, and establishes:

1. the first instance-dependent upper and lower bounds that cover both biased and unbiased offline data;

2. an extension to stochastic linear bandits;

3. the first robustness guarantees when the offline–online discrepancy is unknown.

**Future directions.**
Guided by the reviewers’ comments, we see several promising extensions: (i) improving computational efficiency, (ii) handling model misspecification, and (iii) generalizing to nonlinear demand models. We also hope this work spurs broader study on integrating general offline data into online learning.

References:

[a] Besbes, Omar and Zeevi, Assaf. On the (surprising) sufficiency of linear models for dynamic pricing with demand learning. 2015.

[b] Keskin, N Bora and Zeevi, Assaf. Dynamic pricing with an unknown demand model: Asymptotically optimal semi-myopic policies. 2014.

[c] Qiang, Sheng and Bayati, Mohsen. Dynamic pricing with demand covariates. 2016

[d] Simchi-Levi, David and Wang, Chonghuan. Pricing experimental design: causal effect, expected revenue and tail risk. 2023.

[e] Li, Xiaocheng and Zheng, Zeyu. Dynamic pricing with external information and inventory constraint. 2024.

---

### Decision · Program_Chairs · 2025-09-17

**Decision:**

Accept (poster)

**Comment:**

This paper tackles the important and practical problem of contextual online pricing when an agent has access to a potentially biased offline dataset. The authors propose algorithms that leverage this offline data to improve online performance, providing minimax-optimal and instance-dependent regret bounds. The reviewers are somewhat divided, with scores ranging from 3 to 5. However, a careful reading of the reviews and the author rebuttal process reveals that this is a technically solid paper with meaningful contributions.

All reviewers agree that the problem setting is well-motivated, timely, and significant. The primary strengths of the paper lie in its theoretical contributions. Reviewers 4jmU and TipE respectively described the analysis as "formal and well-performed" and "rigorous," highlighting the paper's technical depth.

The main points of contention were:

**Novelty and Incrementality**: Several reviewers (4evj, 4jmU, oYic) initially questioned the paper's novelty, particularly in relation to prior work by Bu et al. (2020). However, in their rebuttal, the authors provided a detailed breakdown of their novel contributions, including the introduction of a new instance-dependent parameter, new proof techniques for handling biased data, and robustness guarantees. This explanation was sufficiently compelling to convince Reviewer TipE of the paper's "meaningful technical novelty." While other reviewers remained concerned, the authors' defense makes a strong case that the work is more than a simple extension of prior art.

**Technical Correctness**: Reviewer 4evj raised a serious concern about an error in the proof of the lower bound. This point was debated extensively. The authors provided a detailed and rigorous mathematical counter-argument that ultimately convinced the reviewer.

**Clarity and Presentation**: Reviewer 4evj was critical of the paper's clarity. However, this view was not shared by Reviewer 4jmU ("well-written and structured") or Reviewer TipE ("clearly written and well organized"). The authors have also committed to improving the presentation in the final version.

**Model Misspecification and Experiments**: Reviewers TipE and oYic both raised concerns about the potential model misspecification of the linear demand model. This is a legitimate concern. However, linear demand models are indeed quite common in the dynamic pricing literature. The authors also mentioned that linear demand models have certain robustness in the context of decision-making. That being said, the authors should discuss the issue of model misspecification in the paper. Moreover, the authors should expand the numerical studies using real-world datasets, which will substantially strengthen the final paper.

In summary, this paper addresses a significant problem at the intersection of online learning and revenue management. While the initial reviews raised valid concerns, the authors have successfully addressed the most critical of these. The technical core of the paper has withstood scrutiny, particularly regarding the lower-bound proof. The authors have also made a convincing case for the novelty of their contributions. The work is technically solid and provides valuable insights. Therefore, I recommend acceptance. Meanwhile, I suggest the authors further improve their experiments and the presentation and discuss the issue of model misspecification.